# Nanoscale analysis of human G1 and metaphase chromatin in situ

Jon Ken Chen [ID][1,4], Tingsheng Liu [ID][1], Shujun Cai [ID][1], Weimei Ruan [ID][2], Cai Tong Ng [ID][1], Jian Shi [ID][1], Uttam Surana [ID][2,3] & Lu Gan [ID][1,4]✉

## Abstract

**The structure of chromatin at the nucleosome level inside cells is still incompletely understood. Here we present in situ electron cryotomography analyses of chromatin in both G1 and metaphase RPE-1 cells. G1 nucleosomes are concentrated in globular chromatin domains, and metaphase nucleosomes are concentrated in the chromatids. Classification analysis reveals that canonical mononucleosomes, and in some conditions ordered stacked dinucleosomes and mononucleosomes with a disordered gyre-proximal density, are abundant in both cell-cycle states. We do not detect class averages that have more than two stacked nucleosomes or side-by-side dinucleosomes, suggesting that groups of more than two nucleosomes are heterogeneous. Large multi-megadalton structures are abundant in G1 nucleoplasm, but not found in G1 chromatin domains and metaphase chromatin. The macromolecular phenotypes studied here represent a starting point for the comparative analysis of compaction in normal vs. unhealthy human cells, in other cell-cycle states, other organisms, and in vitro chromatin assemblies.**

**Keywords** Chromatin; Mitosis; Nucleosome; Cryo-ET
**Subject Category** Chromatin, Transcription & Genomics

## Introduction

Eukaryotic chromosomes are long nucleoprotein polymers (chromatin) whose basic units are called nucleosomes. The canonical nucleosome is a 10-nm wide, 6-nm thick cylindrical structure that has an eight-histone protein core surrounded by 1.65 left-handed turns (145–147 base pairs) of ordered DNA (Hewish and Burgoyne, 1973; Kornberg, 1974; Luger et al, 1997; Olins and Olins, 1974). Sequential nucleosomes are connected by short stretches of linker DNA. Groups of nucleosomes (oligonucleosomes) are believed to make higher-order interactions. Electron microscopy studies of purified and cellular samples have led to diverse higher-order chromatin models, including the 30-nm fiber (Finch and Klug, 1976), arc-shaped nucleosome stacks (Dubochet and Noll, 1978), liquid-like chromatin (Eltsov et al, 2008; McDowall et al, 1986), zig-zags (Bednar et al, 1998), 5- to 24-nm-thick fibers (Ou et al, 2017), plates (Chicano et al, 2019), 100- to 200-nm periodic structures (Hayashida et al, 2021), and 200-nm-wide slinkies of sequentially stacked nucleosomes (Sedat et al, 2022a; Sedat et al, 2022b). These models have not been adequately tested in situ in the absence of traditional electron-microscopy sample-preparation perturbations.

In metazoans, chromosomes are decompacted during interphase (Huxley and Zubay, 1961) and compacted to individualized chromatids in mitosis (Flemming, 1880). Super-resolution microscopy studies of interphase mammalian cells have detected small irregular clusters of nucleosomes called clutches (Ricci et al, 2015) and 160- to 500-nm-wide chromatin-rich regions called "chromatin domains" (Barth et al, 2020; Fang et al, 2018; Li et al, 2021; Miron et al, 2020; Nagashima et al, 2019; Nozaki et al, 2017; Otterstrom et al, 2019; Xiang et al, 2018; Xu et al, 2018). Chromatin domains have also been indirectly detected by super-resolution microscopy of mitotic cells (Nozaki et al, 2017). Correct chromatin organization is required for important mitotic phenotypes like transcriptional repression and accurate segregation. Recent cryo-EM advancements make it possible to study these rearranged states at the nucleosome level in 3-D.

Electron cryotomography (cryo-ET) is a popular form of cryo-EM that is used to study the in situ 3-D structure of cells in a frozen-hydrated life-like state, without the artifacts associated with the chemical fixation, dehydration, and staining (Ng and Gan, 2020). In situ cryo-ET studies of eukaryotic chromatin have been done in microbial (Cai et al, 2018b; Chen et al, 2016; Gan et al, 2013; Tan et al, 2023; Tan et al, 2025), human (Cai et al, 2018a; Hou et al, 2023), mouse embryonic fibroblasts (MEF) (Wang et al, 2024), and insect cells (Eltsov et al, 2018; Fatmaoui et al, 2022). These studies confirmed that the chromatin of multiple species is irregular in situ (Itoh et al, 2021). Importantly, cryo-ET can reveal nucleosome-level chromatin structure (Cai et al, 2018a; Eltsov et al, 2018), non-chromatin globular multi-megadalton complexes (megacomplexes), and "pockets"—regions that are devoid of megacomplexes and nucleosome-like particles (Cai et al, 2018b).

[1]Department of Biological Sciences and Centre for BioImaging Sciences, National University of Singapore, Singapore 117543, Singapore. [2]Institute of Molecular and Cell Biology and Agency for Science Technology and Research, 61 Biopolis Drive, Singapore 138673, Singapore. [3]Department of Pharmacology, National University of Singapore, Singapore 117543, Singapore. [4]Present address: Department of Molecular Physiology and Biological Physics, University of Virginia, Charlottesville, VA 22903, USA. ✉E-mail: lu@anaphase.org

When groups of macromolecular complexes that have very similar conformations are abundant, i.e., they are ordered, they manifest as unambiguous class averages in 3-D classification and subtomogram averaging analysis. In situ cryo-ET is therefore an essential tool to test different chromatin compaction models and to discover previously undescribed molecular phenotypes.

Here we present cryo-ET analysis of chromatin in cryo-FIB-milled RPE-1 cells, arrested in G1 phase (herein abbreviated to G1) and in metaphase. These cell-cycle states represent the extremes of both chromosome compaction and of transcriptional activity. Our study reveals differences in chromatin structure spanning length scales from ~2 nm to more than 100 nm, from the mononucleosome to the chromatin domain level. Subtomogram classification and averaging analyses reveal canonical nucleosomes and other nucleosome-containing assemblies not seen in previous studies. Remapping shows that G1 nucleosomes pack into domains and that metaphase chromatin does not have internal domain-like structure. These findings reveal structural features that are either shared or that deviate in two vastly different cell-cycle states.

## Results

### Cell cycle synchronization

We used a pharmacological block-and-release protocol to enrich hTERT-immortalized human retinal pigment epithelial (RPE-1) cells in both G1 and metaphase (Scott et al, 2020). First, G1 cells are enriched by arrest with palbociclib. The cells are then released from G1, allowed to grow, and treated with nocodazole to arrest at prometaphase. Finally, the cells are washed free of nocodazole and immediately incubated in MG132 to allow progression to and then arrest at metaphase (Appendix Fig. S1A). This mitotic arrest is reversible (Scott et al, 2020), which we confirmed by following the loss of mitotic markers histone H3 phosphorylated at serine 10 and Cyclin B1 (Appendix Fig. S1B). The cytology of both G1 and metaphase cells and chromosomes were verified by immunofluorescence microscopy (Appendix Fig. S1C,D). Herein, we refer to palbociclib- and MG132-arrested cells as G1 and metaphase cells, respectively.

In addition to morphological differences, an important phenotypic difference between interphase and mitotic chromatin is the repression of transcription in mitosis. To confirm these phenotypes in synchronized RPE-1 cells, we used antibodies specific for the RNA polymerase II (RNAPII) large subunit Rpb1's C-terminal heptad repeats YSPTSPS phosphorylated at S5 (CTD-S5P) and at S2 (CTD-S2P). These epitopes indicate initiating RNAPII and elongating RNAPII, respectively (Harlen and Churchman, 2017; Komarnitsky et al, 2000). As expected, both forms of RNAPII were detected in G1 cell nuclei and were largely excluded from the chromatin of metaphase cells (Fig. 1A). These immunofluorescence results are consistent with the near-global repression of chromatin transcription in metaphase cells.

### Optimization of cryoprotectants for cellular cryo-ET

The visualization of macromolecular structure in a life-like frozen-hydrated state in situ requires that cells be rapid-frozen in way that avoids the formation of crystalline ice (Dubochet et al, 1988). Here

we briefly summarize the logic and extensive control experiments done to reproducibly generate crystalline-ice-free frozen-hydrated RPE-1 cells; see the Appendix discussion for details. We tested both cryosectioning (Ladinsky et al, 2006; Studer et al, 2014) of self-pressurized-frozen samples (Han et al, 2012; Yakovlev and Downing, 2011) and cryo-FIB milling of plunge-frozen samples (Hayles et al, 2007; Marko et al, 2006; Medeiros et al, 2018; Rigort et al, 2010; Schaffer et al, 2015; Villa et al, 2013). Our cryosectioned self-pressurized frozen samples showed diffraction-contrast features (Dubochet et al, 1988) (Appendix Fig. S2A,B), stringy chromatin densities (Appendix Fig. S2C,D), severe cutting artifacts, and grid-attachment issues, all of which preclude the use of cryosections. Nevertheless, the pilot cryosections of metaphase-arrested RPE-1 cells showed ribosome-excluding chromatin, reproducing previous in situ cryo-EM studies (Eltsov et al, 2008; McDowall et al, 1986), which guided the localization of metaphase chromosomes in our subsequent experiments. To reproducibly freeze cells, we instead used plunge-freezing, which we and others have successfully used to study the nuclear periphery (Cai et al, 2018a; Hou et al, 2023; Mahamid et al, 2016). To suppress ice crystal growth, we froze the cells in the presence of cryoprotectants (Bäuerlein et al, 2022; Cai et al, 2022; Creekmore et al, 2024; Glynn et al, 2024; Harapin et al, 2015; Jentoft et al, 2023), of which glycerol and DMSO are popular choices. Timelapse light microscopy showed that cells in DMSO (but not glycerol) showed time- and concentration-dependent deformations (Appendix Fig. S3), meaning that incubations in DMSO are kept as short as possible. Trial tilt series of cryoprotected RPE-1 G1 cells show that that ice-crystal diffraction contrast was undetectable in most samples in 9% DMSO (Appendix Fig. S4).

The presence of cryoprotectants lowers cryo-EM image contrast, prompting us to increase the contrast by imaging with a Volta phase plate (VPP) (Danev et al, 2014; Fukuda et al, 2015). VPP data showed that 9% glycerol treatment resulted in lower contrast than 9% DMSO (Appendix Fig. S5). Because DMSO destabilizes some protein complexes (Chan et al, 2017), we performed control VPP cryo-ET experiments on HeLa oligonucleosomes frozen in either pure storage buffer or storage buffer plus 9% DMSO. Cryotomograms showed the characteristic beads-on-string motif and double-gyre motif corresponding to side and gyre views (Zhou et al, 2019) in both the presence and absence of 9% DMSO (Appendix Fig. S6). Subtomogram classification and averaging analysis (see methods) revealed canonical nucleosome class averages that have short ordered linker DNA in both samples (Appendix Fig. S7).

To further test for potential perturbative effects of cryoprotectant treatment, we performed immunofluorescence to detect the distribution of initiating (Fig. 1B,C) and elongating RNA polymerase II (Fig. EV1), which have CTD-S5P CTD-S2P markers, respectively. These images showed that in both G1- and metaphase-arrested cells, large-scale chromatin structure is not disrupted by 9% glycerol or DMSO. Additional western blot analysis showed that the apoptosis marker cleaved poly (ADP-ribose) polymerase-1 (PARP-1) (Kaufmann, 1989; Kaufmann et al, 1993) is only detected in the positive control that was treated with the apoptosis inducer staurosporine (Bertrand et al, 1994), but not in the negative control or in the cryoprotectant-treated cells (Appendix Fig. S8). Because brief exposure to 9% DMSO preserves nucleosome structure in RPE-1 cells and does not kill them, we used 9% DMSO to cryoprotect most of the samples described in this study.

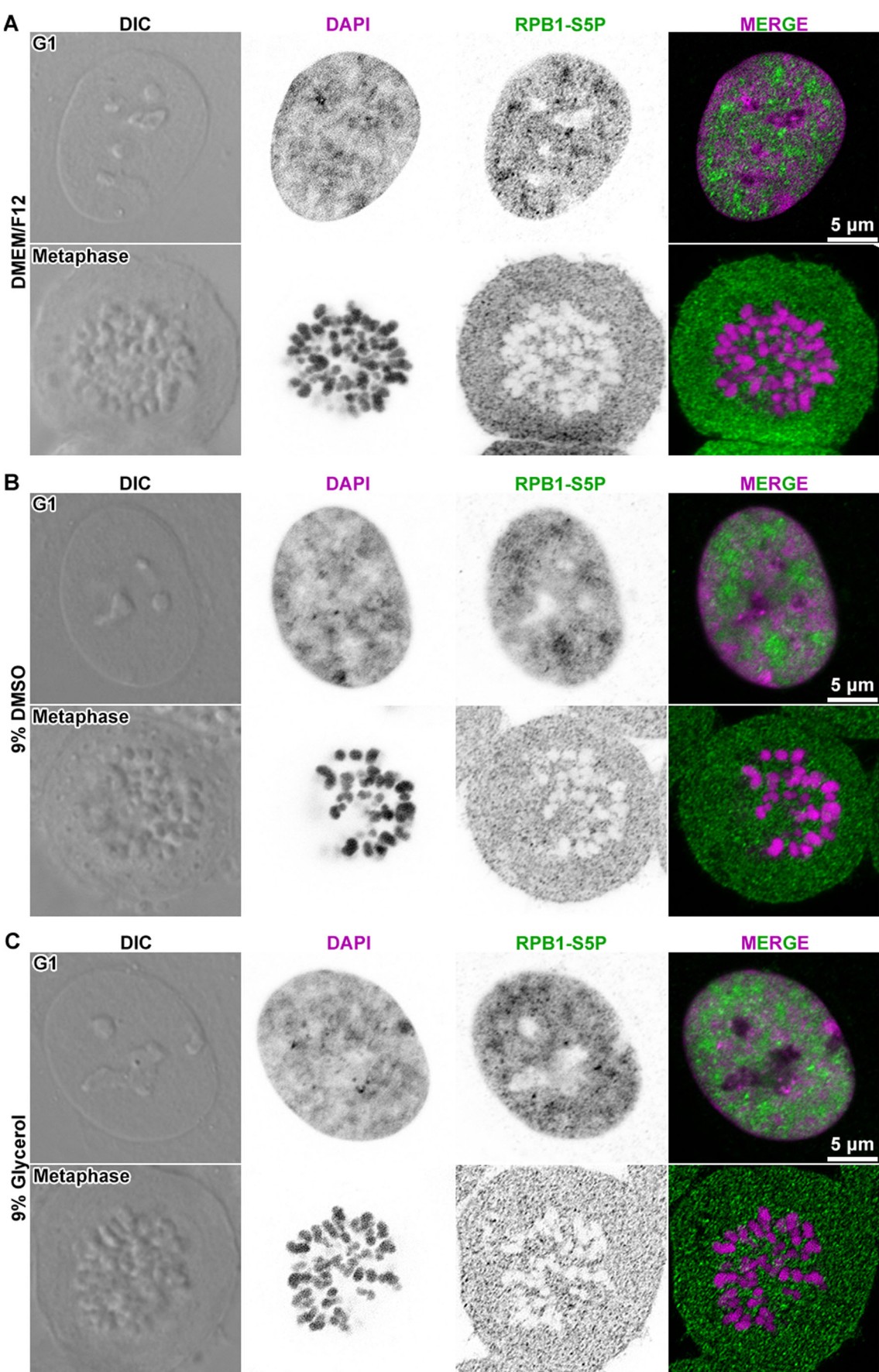

**Figure 1.  Metaphase chromosomes are depleted of RNA polymerase II.**

Differential interference contrast (DIC) images of representative G1 and metaphase cells that were stained to detect DNA (stained with DAPI) and immunofluorescent detection of initiating RNA polymerase II (RNAPII) phosphorylated at serine 5 of the RPB1 subunit's C-terminal tail heptad repeats (RPB1-S5P). (**A**) G1 and metaphase cells were incubated in DMEM/F12 medium for 1 min before fixation. (**B**) G1 and metaphase cells were incubated in DMEM/F12 medium containing 9% DMSO for 1 min before fixation. (**C**) G1 and metaphase cells were incubated in DMEM/F12 medium containing 9% glycerol for 1 min before fixation. In the two middle columns, the DAPI and immunofluorescence signals are shown in inverted contrast for clarity.

## Overview of the G1 nucleus interior

G1-arrested cells were immersed in medium containing 9% DMSO, plunge-frozen, cryo-FIB milled, and then imaged by VPP cryo-ET. In the G1 cell nucleus cryotomographic slices, we observed masses of nucleosome-like densities, both near the nuclear envelope (Fig. 2A) and further in the nuclear interior (Fig. 2B). An additional example cryotomographic slice from another G1 cell is shown in Appendix Fig. S9. To control for the cryolamellae quality, we template matched cytoplasmic ribosomes in G1 cryolamellae (Appendix Fig. S10) and then directly 3-D classified the candidate ribosomes to yield 684 ribosome subtomograms. Note that the low number of ribosomes results from the targeting of nuclei, meaning that cytoplasm occupies small portions of the fields of view. The refined average is at ~33 Å resolution (Appendix Fig. S10B) and has the same handedness as and comparable features to density maps of human ribosome single-particle cryo-EM reconstructions (Khatter et al, 2015) simulated at similar resolutions (Appendix Fig. S10C).

Densities that resemble the nucleosome gyre view are abundant in the locally concentrated groups (chromatin domains; see below) in both perinuclear and intranuclear regions (Fig. 2C,E). In addition to nucleosome-like densities, the G1 nucleus contains conspicuous non-nucleosome features, of which the most abundant are megacomplexes (Fig. 2D). Because the best-characterized nuclear megacomplexes are preribosomes, we template matched them using a spherical reference. We performed both 2-D and 3-D classification, yielding one class average of candidate preribosomes (Fig. EV2A). The preribosome average resembles the 60S complex, which is apparent when the refined structure is oriented to the large subunit of the 80S ribosome (Fig. EV2B). In addition to preribosomes, G1 nuclei have dense bodies with irregular shapes and dimensions more than 40 nm (Fig. 2F); because they are much larger than megacomplexes, herein we refer to them as "dense irregular bodies". The dense irregular bodies do not have any double-gyre motifs, suggesting that they do not contain nucleosomes. They are also not a result of DMSO treatment because we have, in retrospect, observed them in both glycerol-cryoprotected RPE-1 cells and in other cells that were not treated with DMSO (see below).

## G1 chromatin domains have multiple ordered nucleosome species

The mammalian interphase nucleus contains both chromatin-rich and chromatin-poor regions. This uneven chromatin distribution is visible in confocal images of RPE-1 cells (Fig. 1), consistent with recent confocal and super-resolution studies of other cell types (Barth et al, 2020; Cho et al, 2022; Fang et al, 2018; Li et al, 2021; Miron et al, 2020; Nagashima et al, 2019; Nozaki et al, 2017; Otterstrom et al, 2019; Xiang et al, 2018; Xu et al, 2018). The super-

resolution studies have named the 100- to 500-nm-wide compact chromatin-rich regions "chromatin domains". Because the nucleosome-rich regions that we see in G1 cells have dimensions compatible with the chromatin domains seen by super-resolution microscopy, herein we call them "chromatin domains", and for brevity, we refer to the chromatin-poor regions as nucleoplasm.

Next, we performed subtomogram classification and averaging analysis of nucleosomes in the G1 chromatin domains, which revealed canonical nucleosome class averages (Appendix Fig. S11A) like we saw in HeLa (Cai et al, 2018a) and what others have seen in T-lymphoblast cells (Hou et al, 2023), and mouse cells (Wang et al, 2024). For a negative control, we repeated this analysis in the cytoplasm, but did not find any nucleosome class averages as expected (Appendix Fig. S11B). A modified template-matching and classification strategy (see Appendix discussion) revealed 4 canonical-nucleosome class averages (Appendix Fig. S12A), of which two had an extra density near one face. Further classification using a double-cylinder reference and larger mask (Appendix Fig. S12B) revealed two nucleosome class averages that each had an extra density near the face. The subtomograms belonging to these two classes were pooled and subjected to a second round of 3-D classification, using the larger mask and one of the nucleosome class averages as the reference (Appendix Fig. S12C). This second round of classification revealed a stacked dinucleosome class average and six class averages that have an ordered mononucleosome plus an extra, disordered density. In the stacked dinucleosome, both nucleosome densities have unambiguous features of mononucleosomes with DNA gyres well resolved; herein we refer to this class as ordered stacked dinucleosomes. Here, the term "ordered" implies that there are sufficient numbers of canonical nucleosome pairs that are oriented and positioned with very low heterogeneity, resulting in a class average with two unambiguous canonical nucleosome densities. Stacked dinucleosomes have a broader range of inter-nucleosome orientations and positions would be too heterogeneous to resolve as a separate, unambiguous class average.

To potentially detect more subtomograms that contain a canonical nucleosome, we repeated the first round of 3-D classification with the larger mask and using the stacked dinucleosome class average as the reference (Appendix Fig. S12D). This modified classification strategy revealed three types of class averages: (1) canonical nucleosomes, (2) a stacked dinucleosome, and (3) a canonical nucleosome with a large noisy density near the DNA gyres (Appendix Fig. S12D). The subtomograms of each class average type were subjected to a third round of classification, resulting in seven classes from group 1, three classes from group 2, and five classes from group 3 (Appendix Fig. S12E). Group 1's class averages primarily differ in the length of the linker DNA and the noisy densities near the nucleosome face, so we refined each class average separately. Group 2's classes differ primarily in their noise

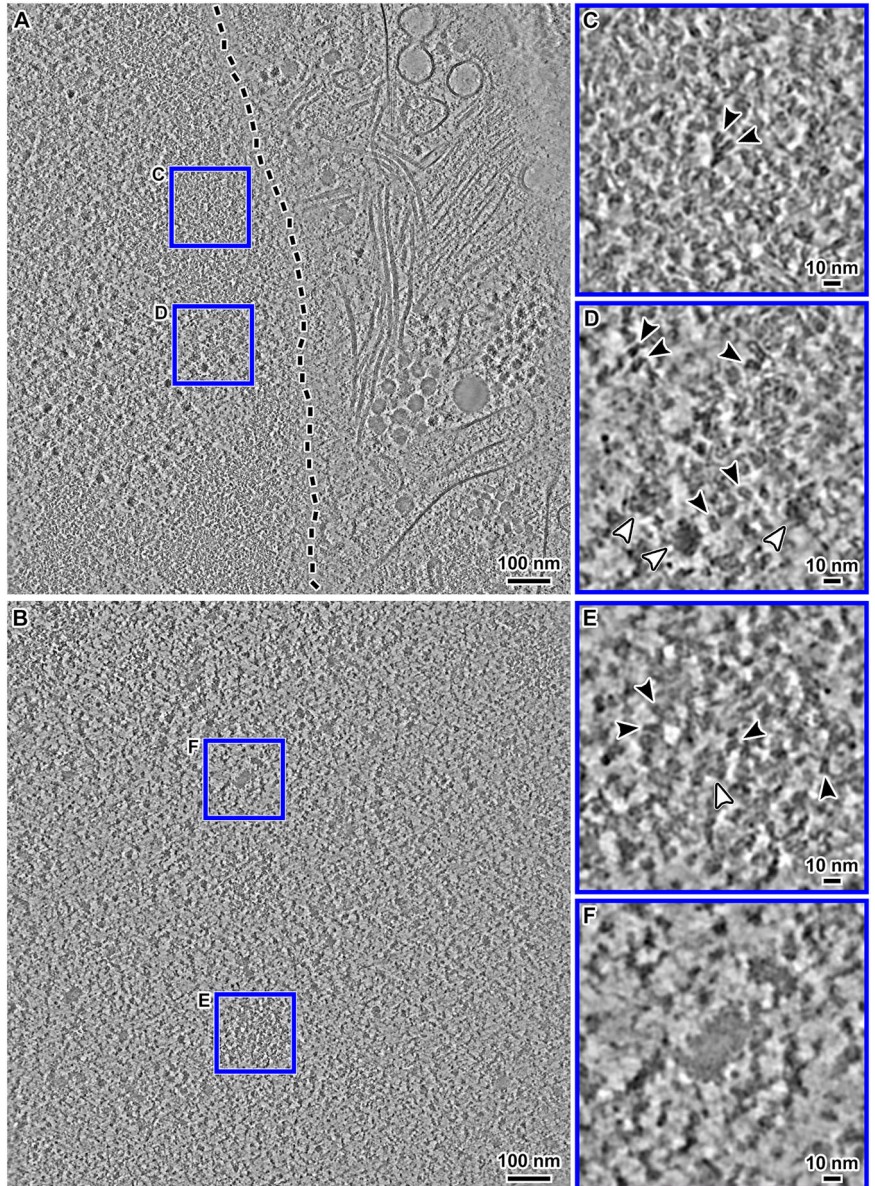

**Figure 2. Cryo-ET of G1 chromatin in situ.**

(A) Cryotomographic slice (10 nm) of the perinuclear region. The nuclear envelope is oblique to the milling direction and is therefore not visible. Its approximate center is indicated by the dashed line. (B) Cryotomographic slice (10 nm) of a region deeper inside the nucleus. (C) Enlargement (4-fold) of one region boxed in (A), showing a perinuclear chromatin domain. (D) Enlargement of a megacomplex-rich nucleoplasm region boxed in (A). (E) Enlargement of the chromatin domain boxed in (B). (F) Enlargement (4-fold) of a "dense irregular body" in (B). In all insets, nucleosomes are indicated by black arrowheads and megacomplexes are indicated by white arrowheads. These images are from cryotomograms denoised by CryoCARE. Non-denoised versions are shown in Appendix Fig. S32.

densities, so we pooled them for refinement. Group 3's class averages have gyre-proximal densities at variable positions, so we refined each class separately. The 13 refined class averages have resolutions ranging from 24 to 34 Å (Appendix Fig. S13, Movie EV1).

Group 1's seven refined mononucleosome class averages differ by the length of ordered linker DNA (Fig. 3A). Furthermore, three of these class averages have asymmetry in the lengths of the two linker DNAs. Note that the difference in the subtomogram averages' linker DNA lengths may result either from the absolute

length of the individual nucleosomes' linkers or from the heterogeneity of the linker DNA structure (see Discussion). When rendered at a lower contour level, the classes have a heterogeneous extra density near one of the faces (Appendix Fig. S13A), indicating that another complex is present, though at a lower occupancy (more details below). Remapping analysis in the later section suggests that some of gyre-proximal densities have contributions from other nucleosomes.

The two nucleosomes of the ordered stacked dinucleosome class average (Group 2) differ in the lengths of ordered linker DNA

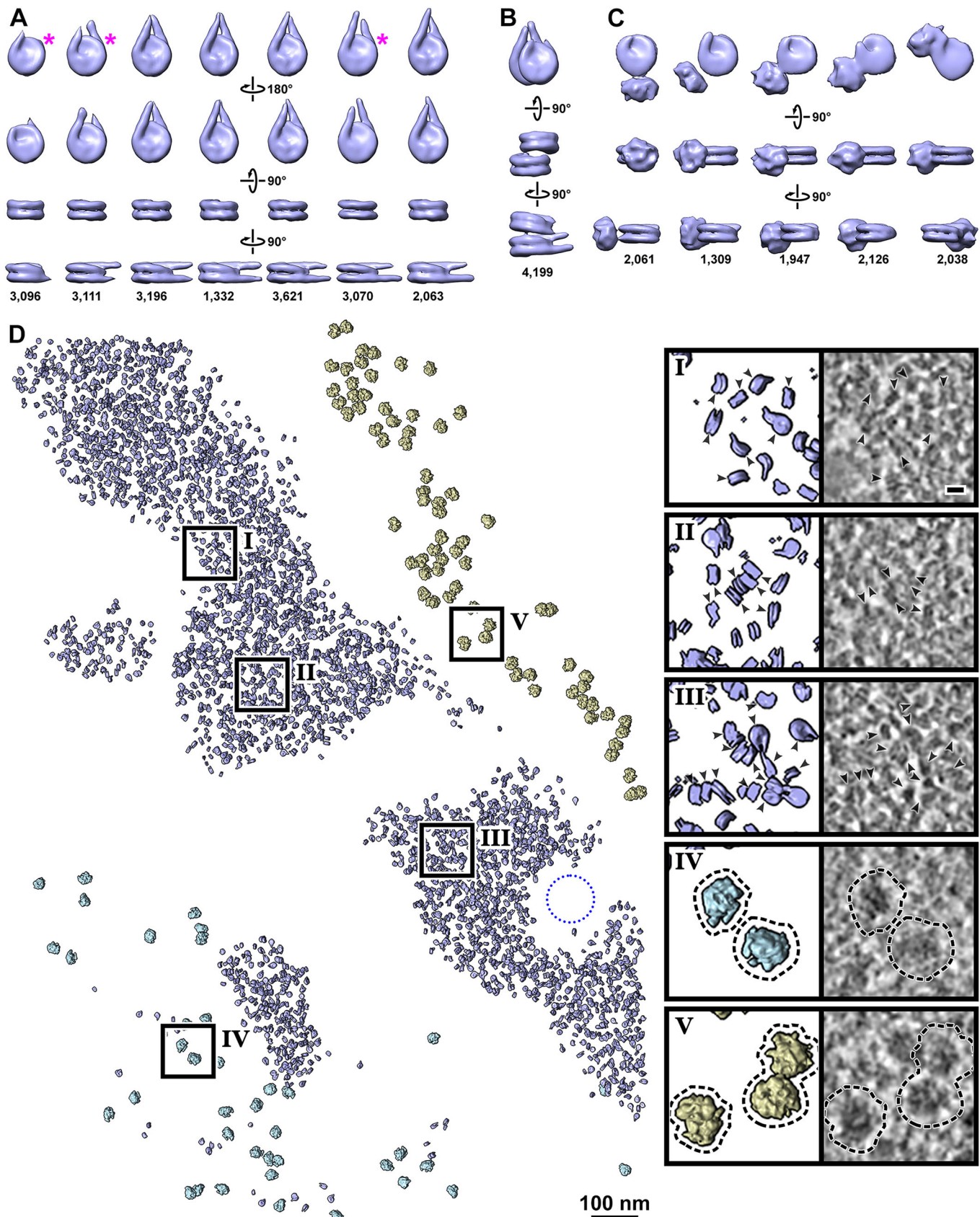

**Figure 3. Subtomogram analysis of G1 nucleosomes.**

(A) Subtomogram averages of mononucleosomes in G1 cells, shown in the (top to bottom) disc, disc, gyre, and side views, respectively. They are ordered (left to right) from those having the shortest to the longest ordered linker DNA. Classes marked with a magenta asterisk have noticeably asymmetric linker DNA length. (B) Subtomogram average of ordered stacked dinucleosomes. (C) Subtomogram averages of nucleosomes with disordered gyre-proximal densities; to show this extra density, these averages are rendered at a lower contour level than those of (A) and (B). Some of the images in (A–C) are reproduced from Appendix Fig. S13; the structures are also shown in Movie EV1. At the bottom of (A–C), the numbers of subtomograms that belong to each class is shown. (D) Remapped model of nucleosomes (blue) in the chromatin domains, preribosomes (light blue), and cytoplasmic ribosomes (yellow). The remapped model includes nucleosome classes shown in (A–C). To facilitate the visualization of the nucleosomes, the proximal densities are hidden in the remapped model by a combination of higher contour level and the UCSF Chimera "hide dust" tool. The remapped model spans the entire thickness of the lamella (~90 nm). The left insets show 3-fold enlargements of the regions boxed on the left and right inset shows a cryotomographic slice (10 nm) of the same remapped region. In the insets, the black arrowheads indicate nucleosomes, and the black dashed outlines indicate either preribosomes or ribosomes. The circular gap in the lower-right group of nucleosomes is volume occupied by a nuclear pore complex (approximate position indicated by a 100-nm dotted circle). See also Movie EV2. The cryotomographic slices in (D) are from cryotomograms denoised by Cryo-CARE. This figure is reproduced with non-denoised cryotomographic slices in Appendix Fig. S33.

(Figs. 3B and S15B). As with the Group 1 mononucleosomes, the linker DNA density could appear shorter in one of the dinucleosome average's members due to either short linker DNA or variable linker DNA conformation in the contributing nucleosomes (see Discussion). Earlier single-particle cryo-EM analysis of recombinant oligonucleosome arrays revealed dinucleosome stacking (Dombrowski et al, 2022; Garcia-Saez et al, 2018; Song et al, 2014). Both the tetranucleosome and twisted-fiber arrays show two types of dinucleosome stacking: closely packed dinucleosomes (Type I interface) and less-closely packed nucleosomes (Type II interface). Following the nomenclature in these studies, the G1 stacked dinucleosome structures we observe have the Type II interface (Fig. 3B).

Nucleosomes with a gyre-proximal density (Group 3) are conformationally variable (Fig. 3C; Appendix Fig. S13C). Each class average has a canonical nucleosome—albeit with lower-resolution features than in the mononucleosome and stacked dinucleosome classes. These class averages refined lower resolutions than the other groups (30–34 Å vs. low ~20's Å resolution; Appendix Fig. S13D–F). Compared to the other groups, the disc views of this group are undersampled (Appendix Fig. S13G), meaning that the members of this class are more abundant than the numbers detected here. When rendered at a lower contour level, an additional noisy density becomes visible proximal to the DNA gyres (Appendix Fig. S13C, lower row). Even though the additional densities appear noisy, they are not uniformly distributed around the gyre. Instead, their centers of mass are at four different superhelical locations (seven when the nucleosome's pseudo-twofold symmetry is considered); see the analysis with docked nucleosome crystal structure below.

### Other ordered multiple-nucleosome motifs are not detected in G1

We also attempted to detect ordered sets of nucleosomes packed in other arrangements. One such arrangement is the side-by-side dinucleosome. The original mask (240 Å sphere) is big enough to accommodate a side-by-side dinucleosome. However, the isotropic shape of the mask can enclose other nearby particles, which, if heterogeneous, would interfere with both alignment and classification. We therefore designed a mask that better conforms to the longer aspect ratio needed to accommodate two nucleosomes packed side-by-side. We then performed another round of 3-D classification on the subtomograms from Group 3 using the new

mask and reference. However, none of the resultant class averages revealed two canonical nucleosome densities (Fig. EV3A).

The presence of ordered stack dinucleosomes raised the possibility of stacks with more than two ordered nucleosomes. To test if ordered stacking extends beyond two nucleosomes in G1 chromatin domains, we modified our classification workflow to accommodate either three or four stacked nucleosomes (Fig. EV3B). We used stacked tubular masks instead of a spherical one and we used the stacked dinucleosome class average as the reference for an additional round of 3-D classification on the subtomograms from Group 2. No ordered tri- or tetranucleosome stacks emerged from 3-D classification. A few classes have noisy densities above/below the upper-most/bottom-most ordered nucleosome, which may correspond to an additional heterogeneously oriented/positioned nucleosome or other macromolecular complex. Because the ordered stacked trinucleosome and tetranucleosome would be part of stacks with even more ordered nucleosomes (pentanucleosomes, hexanucleosomes, etc.), class averages with larger ordered stacked nucleosome assemblies would also be absent.

### Ordered stacked dinucleosomes not detected in glycerol-treated cells and have a different structure in non-cryoprotected cells

To control for the effects of DMSO treatment in situ, we also performed cryo-ET on G1 cells that were cryoprotected in 9% glycerol (Appendix Fig. S14A,B). The nucleosomes in these cells were also enriched in the domains (Appendix Fig. S14C–E), but the DNA gyres were less-defined than in the DMSO-treated cells. Stacked nucleosomes (Appendix Fig. S14C–E) and dense irregular bodies were also found in the nucleoplasm (Appendix Fig. S14F). By visual inspection, we could not find any notable differences between the glycerol- and DMSO-cryoprotectant nuclei cryotomogram densities.

To determine if glycerol and DMSO treatment lead to differences in chromatin structure that is too fine to directly visualize, we performed 3-D classification and averaging analysis. Template-matching and 3-D classification analysis using the same strategy as for DMSO-treated cells produced class averages of mononucleosomes and nucleosomes with gyre-proximal densities (Appendix Fig. S15). No ordered stacked dinucleosome classes were found. Group 1's class averages refined to 29–34 Å resolution while group 3's class averages refined to 31 Å resolution (Appendix Fig. S16).

We also performed Volta cryo-ET analysis of G1 cells that were not treated with any cryoprotectants. Note that these cells were damaged by crystalline ice, whose presence could be detected in the tilt-series diffraction-contrast features exemplified in our DMSO concentration screen (Appendix Fig. S4). Volta cryotomographic slices of perinuclear regions had greater contrast (Appendix Fig. S17A,B) than similar regions imaged in either DMSO- or glycerol-cryoprotected cells. Enlargements show that the density difference between nucleosomes and nucleoplasm is more pronounced (Appendix Fig. S17C–E) than in cells cryoprotected with DMSO or glycerol. The nucleoplasm between the macromolecular complexes appears emptier than in both the DMSO- and glycerol-cryoprotected cells. This empty-nucleoplasm phenotype, like the string-like chromatin phenotype of the cryosections, may have resulted from the aggregation artifact that occurs in the absence of vitreous freezing (Dubochet and Sartori Blanc, 2001; Kellenberger, 1987).

To characterize the changes at higher resolution, we did classification analysis using the workflow described above (Appendix Fig. S18A–D), but only detected convincing Group 1 (mononucleosome) class averages. We did not detect any convincing group 2 (ordered stacked dinucleosome) and group 3 (nucleosomes with gyre-proximal densities) class averages. Group 1's class averages refined to 29–56 Å resolution (Appendix Fig. S19); note that two of the class averages (2 and 9) have only a few hundred subtomograms and are therefore at much-lower resolution. Given the lower contrast cryotomograms of the glycerol-cryoprotected cells and the ice-crystal artifacts in the non-cryoprotected cells, the analysis below reports only on the DMSO-treated cells, with caveats where appropriate.

## G1 chromatin domains are irregular and have few megacomplexes and pockets

To visualize the overall shape of these chromatin-rich regions, we used the EMAN2 convolutional neural network tool (Chen et al, 2017; Tang et al, 2007) to train and then automatically segment the chromatin domains (Fig. EV4). Segmentations were done at both the nuclear periphery and the nuclear interior (more examples in Appendix Fig. S20). Because the cryolamellae are thinner than the chromatin domains, the segmentation samples a random slice through each domain. Nevertheless, all chromatin domain segmentations have irregular shapes, suggesting that the domains are themselves irregular because random sections through platonic bodies like spheres and ovoids would have circular and oval profiles, respectively.

To visualize the nucleosomes' in situ 3-D distribution, we remapped them into the volume of the original cryotomogram along with the preribosomes and (cytoplasmic) ribosomes (Fig. 3D, Movie EV2). Consistent with the appearance of cryotomographic slices, the nucleosomes were not mixed with the preribosomes (megacomplexes). Instead, they are concentrated in regions that span several hundred nm as expected of chromatin domains. To determine if nucleosomes belonging to each group had a biased distribution, we remapped the mononucleosomes, ordered stacked dinucleosomes, or nucleosomes with gyre-proximal densities into three separate maps (Appendix Fig. S21), but did not find any bias or clustering of any of these classes. Furthermore, separate remaps of each class (7 for mononucleosomes, 1 for stacked dinucleosomes,

5 for nucleosomes with gyre-proximal densities) did not reveal any biased distributions (Appendix Fig. S22). The remapped model shows that the chromatin domains have irregular shapes, like we expected based on the segmentations. Stacks with four nucleosomes are also present in the remapped model (Fig. 3D, inset II), but are not ordered because their numbers were too low to detect as a class average with multiple unambiguous nucleosome densities; see similar analysis of metaphase chromatin below.

Our previous cryo-ET study of fission yeast mitotic chromatin brought our attention to two non-chromatin features—megacomplexes and regions devoid of macromolecular complexes, which we termed pockets (Cai et al, 2018b). After inspecting 21 cryotomograms, we found only one candidate intra-domain megacomplex. In these cryotomograms, we did not find any unambiguous pockets such as the ones seen in metaphase chromatin (see below). In summary, G1 chromatin domains have few non-chromatin features that we have previously seen in compacted fission yeast chromatin.

## Canonical nucleosome classes were not obtained from the nucleoplasm

Mammalian chromatin domains are thought to be connected by short stretches of chromatin "linker filaments" that extend through the nucleoplasm (Li et al, 2021; Miron et al, 2020). The composition of these linker filaments is unknown: they could be mostly naked DNA, oligonucleosome chains, or compacted chromatin fibers. In our cellular cryotomograms, we are unable to see naked DNA, but we can locate canonical nucleosomes by class averaging—if they are abundant. We therefore performed template matching followed by 3-D classification on subtomograms that were outside of the chromatin domains, i.e., within the nucleoplasm and some cytoplasm. The nucleoplasm is more constitutionally heterogeneous than the chromatin domains, so we used 200 classes, of which 157 remained non-empty after the run completed. Only one canonical-nucleosome class average was found (Appendix Fig. S23), but remapping showed that they came from the periphery of the chromatin domains due to imperfect seeding of the initial search points. None of the other class averages bear resemblance to a nucleosome. Therefore, either the linker filament nucleosomes are too rare to manifest as a nucleoplasmic class average in our dataset or the linker-filament nucleosomes have structures that we do not yet recognize.

## Overview of RPE-1 metaphase chromatin

We next prepared DMSO-cryoprotected cryo-FIB-milled metaphase-arrested cells and then recorded VPP cryo-ET data. Unlike G1 cells, in which the chromatin is separated from cytoplasm by a high-contrast easy-to-recognize nuclear envelope, metaphase chromosomes are in contact with the cytoplasm due to mitotic nuclear-envelope breakdown. This difference complicates metaphase-chromosome targeting for cryo-ET. Based on clues from our pilot cryotomograms of cryosections, we found that in the lower-magnification survey images of cryolamellae, cytoplasm regions of cryolamellae have granular and linear densities from ribosomes, membranes, and cytoskeleton; in contrast metaphase chromatin has more uniform densities (Appendix Fig. S24). These visual cues greatly facilitated target selection. In cryotomographic

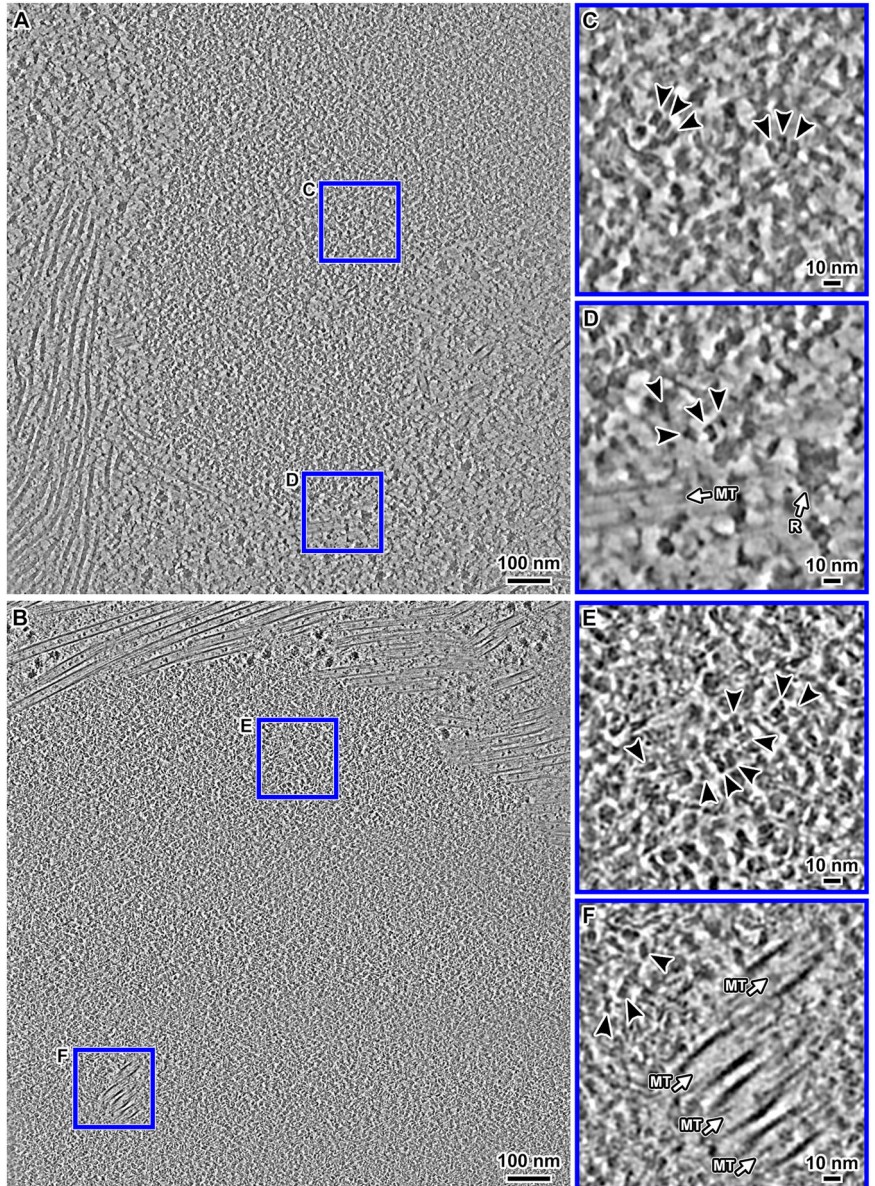

**Figure 4. Cryo-ET of metaphase chromatin in situ.**

(A, B) Cryotomographic slice (10 nm) overview of portions of two metaphase chromosomes. (C, D) Enlargements (4-fold) of two regions boxed in (A). (E, F) Enlargements (4-fold) of two regions boxed in (B). (E) An enlargement of a region of the chromosome boundary. (F) An enlargement of microtubules embedded in the chromosome. In all the insets, the nucleosome-like densities (double-gyre motifs and top-views) are indicated by arrowheads. Note that there are stacked nucleosomes in (C) and (E). Microtubules (MT) and ribosomes (R) are also indicated in the insets. These images are from cryotomograms denoised by Cryo-CARE. Non-denoised versions are shown in Appendix Fig. S34.

slices, metaphase chromatin appears as large regions enriched with nucleosome-like particles and without any cytoplasmic constituents such as ribosomes, bundles of ~10-nm-diameter filaments (Fig. 4A), and microtubule bundles (Fig. 4B). Another metaphase cell example is shown in Appendix Fig. S25.

Ribosomes are located near the surface of the compacted chromosomes (Fig. 4A), consistent with previous analysis of both HeLa cell cryosections (Eltsov et al, 2008), U2OS cryolamellae (Zhao and Jensen, 2022), our cryosections (Appendix Fig. S2), and isolated metaphase chromosomes (Nishino et al, 2012). Spindle microtubules that are inserted into metaphase chromosomes were previously observed in U2OS cells (Kamasaki et al, 2013) and are also seen in the RPE-1 chromosomes (Fig. 4B). Nucleosomes are also irregularly packed in the metaphase chromosome surface and interior, without any large higher-order chromatin structures like 30-nm fibers (Scheffer et al, 2011). In RPE-1 metaphase chromosomes, pockets are present (Appendix Fig. S25), albeit rarer than in G1. However, neither megacomplexes nor dense irregular bodies were seen inside any of the metaphase chromatin regions in our dataset.

## Mononucleosomes and stacked dinucleosomes in metaphase chromatin

The most conspicuous macromolecular complexes in metaphase chromatin are nucleosome-like particles (Fig. 4C–F). We performed template matching, subtomogram classification, and averaging of metaphase nucleosomes using the same image-processing parameters and workflow as for G1 chromatin (Appendix Fig. S26A–E). The individually refined mononucleosome class averages of group 1 have linker DNA of different lengths (Fig. 5A). Unlike in G1 chromatin, we did not find a class average that resembles the nucleosome core particle, which has barely detectable linker DNA. Some of the metaphase class averages with shorter linker length appear to be less well-defined, suggesting that they are more heterogeneous. We also observed a mononucleosome class with asymmetric linker DNA lengths. As with the G1 mononucleosome averages, the differences in metaphase mononucleosome linker DNA lengths could arise from multiple factors (see Discussion).

Metaphase chromatin also has numerous densities that resemble stacked nucleosomes (Fig. 4C–F). Accordingly, subtomogram analysis revealed ordered stacked metaphase dinucleosome class averages (Appendix Fig. S26C–E; Fig. 5B). We also found class averages of nucleosomes with a noisy density proximal to the gyres (Appendix Fig. S27C; Fig. 5C). We refined these class averages to 26 Å (mononucleosomes and stacked dinucleosomes) and 32–35 Å (nucleosomes with gyre-proximal densities) (Appendix Fig. S27, Movie EV3). When rendered at a lower contour level, the metaphase stacked dinucleosome and some of the mononucleosome class averages show an extra density near the nucleosome face (Appendix Fig. S27A,B, lower rows), like what is seen in G1 (more details below).

We performed the modified search for ordered side-by-side dinucleosomes like we did for G1 chromatin and did not detect any candidates in the metaphase chromatin either (Fig. EV5A). Likewise, using the same modified strategy as for G1 chromatin, we attempted—and failed—to detect ordered stacked trinucleosomes in metaphase chromatin (Fig. EV5B). In summary, ordered stacked dinucleosomes, and nucleosomes with gyre-proximal densities are abundant in both G1 and metaphase cells, but ordered structures with two side-by-side or more than two stacked nucleosomes are not.

Next, we remapped the refined class averages of all three nucleosome groups and the ribosomes (Fig. 5D). The remapped ribosomes localized exclusively outside the chromosome, consistent with the absence of megacomplexes inside. Remapping revealed a few sets of stacks with more than two nucleosomes (Fig. 5D, insets I–IV), suggesting that stacked tri- and tetranucleosomes are present (Fig. 4D,E, Movie EV4). These tri- and tetranucleosome stacks are too few to manifest as class averages with unambiguous nucleosomes. Therefore, nucleosome stacking can extend beyond two nucleosomes in situ in both G1 and metaphase, albeit with less order.

As with the G1 cryotomograms, we also remapped the mononucleosomes, ordered stacked dinucleosomes, and nucleosomes with gyre-proximal densities as three separate groups (Appendix Fig. S28), but did not find any distribution bias. Likewise, we remapped each class (8 for mononucleosomes, 1 for ordered stacked dinucleosomes, and 4 for nucleosomes with gyre-proximal densities) and did not find biased distributions of these class averages either (Appendix Fig. S29).

## Detailed remapping analysis of nucleosome packing

At least one class average from each group has an extra density, which is visible at a sufficiently low contour level. To test if these densities are from other nucleosomes, we remapped all the class averages at low contour levels and then shaded them in different colors (Fig. 6). This visualization scheme allowed us to see if any of the 'extra' densities overlapped with the nucleosome densities of neighboring class averages. In both G1 (Fig. 6A) and metaphase (Fig. 6B), we found (1) examples where the extra density partially overlapped a nearby remapped nucleosome (black arrowheads), (2) the extra density did not overlap with a remapped nucleosome but did partially overlap with a nucleosome-like density in the tomographic slice (gray arrowheads), and (3) examples of no overlap with either a remapped nucleosome or a nucleosome-like density in the cryotomographic slice (white arrowhead). In the latter case, there was some overlap with unknown structures, suggesting that non-nucleosome complexes may also contribute to the extra densities. In summary, both nucleosomes and non-nucleosomes contribute to the heterogeneous extra densities in our class averages.

## Comparative analysis of G1 and metaphase nucleosomes in situ

The organization of RPE-1 nucleosomes can be explored in greater detail by comparison with crystal structures. To simplify the analysis, we assigned the group 1 mononucleosome class averages into only two classes, based on either short or long linker DNA lengths. These two classes were then refined (Appendix Fig. S30), separately for G1 and metaphase nucleosomes. Next, we docked nucleosome and chromatosome crystal structures as rigid bodies in the refined class averages from both G1 (Fig. 7A) and metaphase (Fig. 7B). The main source of variation is the length of ordered linker DNA in the mononucleosome and the stacked dinucleosome structures. Both G1 and metaphase stacked dinucleosomes have a center-to-center spacing of ~6.8 nm. Comparison of the nucleosomes with gyre-proximal densities shows that the centers of mass of these densities populate the following approximate superhelical locations: 3.8, 4.8, 5.5, and 6 in G1 chromatin and 3.4, 3.9, 4.2, and 5 in metaphase chromatin. For nucleosomes with gyre-proximal densities, metaphase nucleosomes do not show interactions near the DNA entry/exit site whereas G1 nucleosomes do (superhelical location 6).

To assess how the populations of the ordered nucleosome-containing subtomograms differ in G1 and metaphase, we calculated the ratios of class averages of stacked dinucleosomes plus nucleosomes with gyre-proximal densities relative to the canonical mononucleosome class averages (all linker DNA lengths) in each cell-cycle state (Table 1). Note that this analysis is limited because (1) the total number of canonical vs. non-canonical nucleosomes is unknown and (2) only ordered stacked dinucleosomes are considered here. We found modest differences in the class-average populations: compared to G1 cells, metaphase cells have ~23% more stacked dinucleosomes and ~24% fewer

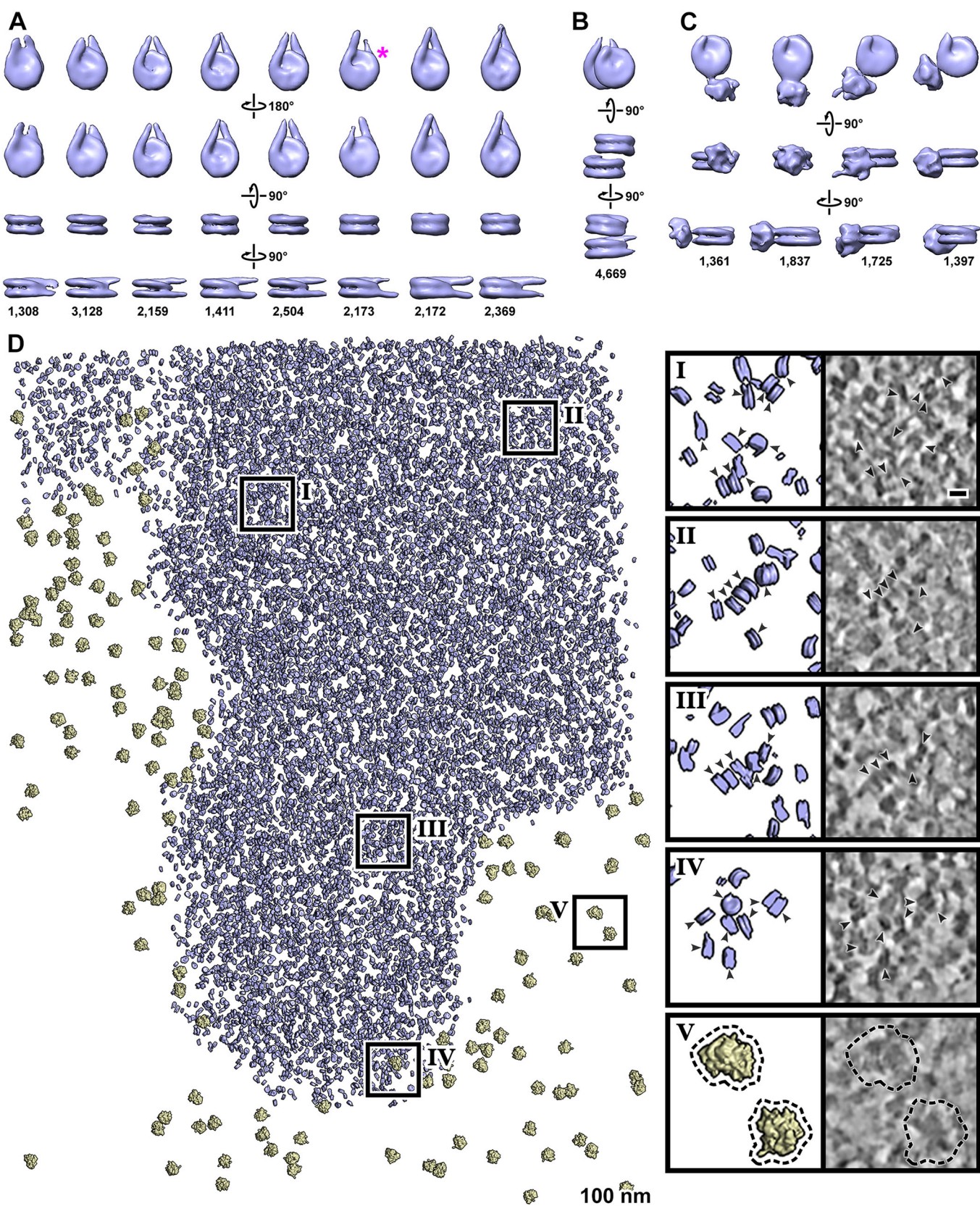

**Figure 5. Subtomogram analysis of metaphase nucleosomes.**

(A) Subtomogram averages of mononucleosomes in metaphase cells, shown in the (top to bottom) disc, disc, gyre, and side views, respectively. They are ordered (left to right) from those having the shortest to the longest ordered linker DNA. The class marked with a magenta asterisk has noticeably asymmetric linker DNA length. (B) Subtomogram average of ordered stacked dinucleosomes. (C) Subtomogram averages of nucleosomes with disordered gyre-proximal densities; to show this extra density, these averages are rendered at a lower contour level than those of (A) and (B). Some of the images in (A–C) are reproduced from Appendix Fig. S27; the structures are also shown in Movie EV3. At the bottom of (A–C), the numbers of subtomograms that belong to each class is shown. (D) Remapped model of nucleosomes (blue) and ribosomes (yellow). The remapped model includes nucleosome classes shown in (A–C). Nucleosomes with gyre-proximal densities are not rendered in the remapped model. The remapped volume contains all Z slices of the reconstructed lamella (~140 nm thick). The left insets show 3-fold enlargements of the regions boxed on the left and the right insets show cryotomographic slices (10 nm) of the same remapped region. In the insets, the black arrowheads indicate nucleosomes, and the black dashed outlines indicate ribosomes. See also Movie EV4. The cryotomographic slices in (D) are from cryotomograms denoised by Cryo-CARE. This figure is reproduced with non-denoised cryotomographic slices in Appendix Fig. S35.

nucleosomes with gyre-proximal densities relative to mononucleosomes (Table 1; raw values are in Tables S6 and S7).

# Discussion

## In situ nucleosome structure and its role in higher-order structure

RPE-1 cells have canonical nucleosome class averages that differ in the length of ordered linker-DNA densities. The class averages that have long ordered linker DNA in a crossed configuration resemble the chromatosome, which is a nucleosome with an additional 10 to 25 bp of ordered linker DNA (Simpson, 1978; Zhou et al, 2015). A crystal structure of the chromatosome with 25 extra base pairs of ordered DNA per arm (Garcia-Saez et al, 2018) could be rigid-body docked into the subtomogram average of the nucleosome classes that have long linker DNA, suggesting that their in situ structure resembles the in vitro one. The averages that have shorter linker densities can arise from nucleosomes that have shorter linker DNA, nucleosomes that have longer but structurally variable linker DNA, or a combination of both. A mononucleosome class average with ultra-short linker DNA was detected in G1 cells but not in metaphase cells, suggesting that linker DNA ordering may be a consequence of chromatin compaction. The class averages of nucleosomes with gyre-proximal densities all have ultra-short linker DNA, in both G1 and metaphase; these averages could result from nucleosomes that have ultra-short linker DNA or those that have more DNA structural heterogeneity. Nucleosome class averages that have short and long ordered linker DNA were also seen in HeLa, T-lymphoblast, and budding yeast cells (Cai et al, 2018a; Hou et al, 2023; Tan et al, 2023), but not in MEFS (Wang et al, 2024); note that canonical nucleosome class averages (regardless of linker DNA length) are much rarer in yeast.

Nucleosome linker DNA orientation and length variability may both contribute to the nucleosome class averages that have short linker-DNA densities. Previous work on isolated chromosomes and oligonucleosomes in vitro produced tomograms of unambiguous nucleosomes (Beel et al, 2021; Zhang et al, 2023; Zhang et al, 2022). While it is possible to directly visualize DNA in purified chromatin, which contains nucleosome and DNA densities, such a task is not feasible in situ, which contains many non-chromatin densities. Denoising may facilitate the visualization of chromatin densities, but may introduce errors if the training data is biased, for example with simulated densities. Further improvements in image quality,

sample preparation, and denoising algorithms may facilitate the annotation and measurement of DNA in unaveraged subtomograms in situ.

Because we did not enforce C2 symmetry in our subtomogram analysis, the mononucleosome class averages show linker-DNA length asymmetry. This asymmetry is also evident in the in situ class averages of HeLa and budding yeast cells (Cai et al, 2018a; Tan et al, 2023). Nucleosomes with asymmetric linker DNA structure have been seen in cryo-EM studies of nucleosomes with histones and DNA from cellular lysates (Arimura et al, 2021); reconstituted oligonucleosomes within small condensates (Zhang et al, 2022); digested native oligonucleosomes (Jentink et al, 2023); decompacted isolated mitotic chromosomes (Beel et al, 2021); and reconstituted chromatosomes (Bednar et al, 2017; Wang et al, 2021b; Zhou et al, 2021). In contrast, nucleosomes with symmetric linker DNA have been found in vitro in oligonucleosome arrays assembled from the Widom "601" sequence and that have more than two ordered nucleosomes (Dombrowski et al, 2022; Ekundayo et al, 2017; Garcia-Saez et al, 2018; Jentink et al, 2023; Lewis et al, 2021; Schalch et al, 2005; Song et al, 2014; Zhou et al, 2022). Taking into account the previous cryo-EM studies, our in situ data, and mesoscale simulations (Collepardo-Guevara and Schlick, 2014), we suggest that in situ linker DNA conformational variability may suppress the formation of ordered oligonucleosome structures that have more than two nucleosomes.

## Type II stacked dinucleosomes are abundant in DMSO-cryoprotected cells

DMSO-cryoprotected G1 and metaphase cells have ordered stacked dinucleosomes, in which DNA gyres and some linker DNA are resolved in both nucleosomes within the same class average. The in situ ordered stacked nucleosomes resemble those studied in vitro that interact via a Type II interface, (Dombrowski et al, 2022; Garcia-Saez et al, 2018; Song et al, 2014), in which the H4 N-terminal residues from one nucleosome contact the H2A-H2B acidic patch of its neighbor. The absence of symmetry enforcement allowed us to see that in the ordered stacked dinucleosome, the two nucleosomes have a ~10 bp difference in ordered linker DNA. We did not detect ordered trinucleosome stacks, meaning that the orientation and/or position of the third nucleosome is variable. Furthermore, while stacks with three nucleosomes are seen in cryotomographic slices (Fig. 4) and stacks with up to four non-ordered nucleosomes were seen in remapped models (Figs. 3D, 5D, and 6), such arrangements are rare compared to ordered stacked dinucleosomes. The rarity of ordering beyond two nucleosomes is

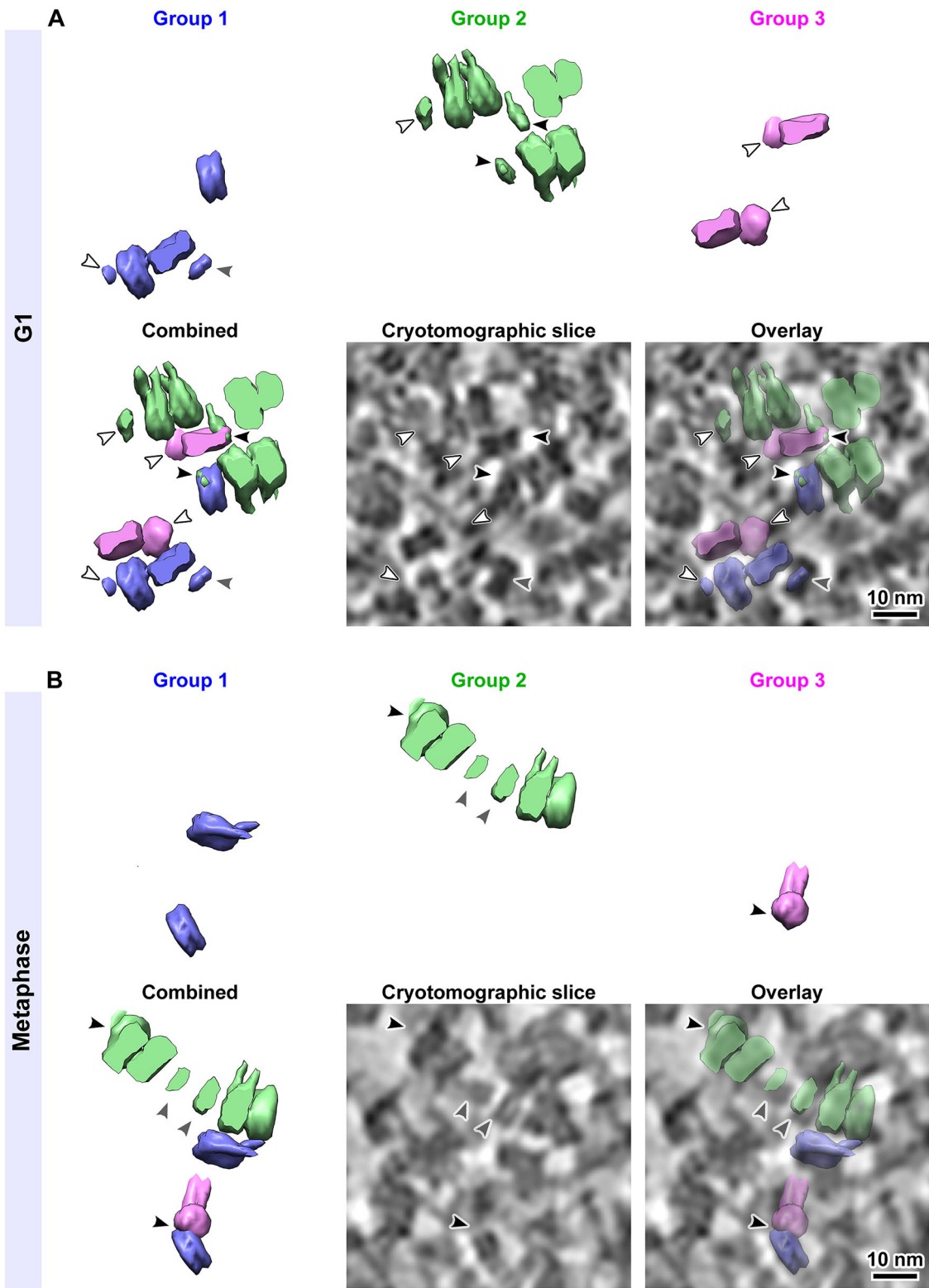

consistent with the absence of the larger ordered structures (fibers and plates; see below).

Our observation that ordered stacked dinucleosomes were not detected in glycerol-treated cells could result either from buffer-induced structural differences or insufficient signal-to-noise ratio due to the glycerol; further investigation will be needed to distinguish these possibilities. The non-cryoprotected G1 cells did not have an ordered stacked dinucleosome class, but did have a

**Figure 6. Example of partial overlaps between subtomogram class averages.**

The panels here show remapped nucleosomes and a 10-nm cryotomographic slice from a region of a (**A**) G1 and (**B**) metaphase cell cryotomogram. In the upper panels, the remapped mononucleosomes (Group 1, blue), ordered stacked dinucleosomes (Group 2, green) and nucleosomes with gyre-proximal densities (Group 3, magenta) are rendered separately; and combined in the lower left panel. The remapped models shown here were generated using the nucleosome classes shown in Figs. 3 and 5, panels (A–C). To facilitate the visualization of the remapped densities, the UCSF Chimera "hide dust" tool was used to hide the small noisy densities. Note that all the nucleosomes in the remapped models here are rendered at a lower contour level to display the extra densities. Comparison of the remapped model with the cryotomographic slice from the same region show that some of the extra densities are indeed located over nucleosomes with side views. Black arrowheads indicate extra densities that partially overlap with a neighboring remapped nucleosome. White arrowheads indicate extra densities that do not appear to overlap any remapped nucleosome. Gray arrowheads indicate an overlap between an extra density with a tomographic slice feature that resembles a nucleosome side view. Possible explanations are that these densities are not nucleosomes or that they are nucleosomes but were mis-classified. The cryotomographic slices shown here are from cryotomograms that were denoised using cryo-CARE. This figure is reproduced with non-denoised cryotomographic slices in Appendix Fig. S36.

class average that contained two nucleosome-like densities that vaguely resembles a stacking interaction. It is unclear to what extent ice-crystal artifacts contribute to the absence of ordered stacked dinucleosomes in these samples. Even though unambiguous ordered stacked nucleosomes are not present in all three cryoprotection conditions tested, it is clear that (less-ordered) stacking is abundant, as can be commonly seen in tomographic slices. Furthermore, the other nucleosome-nucleosome interaction (gyre-proximal) can be seen in a subset of remapped models. These experiments suggest that the detection of some ordered nucleosome-nucleosome assemblies may depend on the biochemical environment, quality of freezing, and image contrast, and will require further investigation.

Nucleosome stacking has been observed in multiple contexts, with varying degrees of order. Stacking is found in preparations of mononucleosomes (Bilokapic et al, 2018; Dubochet and Noll, 1978; Eltsov et al, 2018; Zhou et al, 2018); reconstituted nucleosome arrays (Dombrowski et al, 2022; Ekundayo et al, 2017; Garcia-Saez et al, 2018; Geiss et al, 2014; Jentink et al, 2023; Lewis et al, 2021; Schalch et al, 2005; Song et al, 2014; Takizawa et al, 2020; Zhou et al, 2022); isolated natural chromatin (ex vivo) from chicken erythrocytes, starfish spermatozoids, picoplankton, and budding yeast (Cai et al, 2018c; Scheffer et al, 2012); isolated chicken erythrocyte nuclei (Scheffer et al, 2011); in situ in interphase HeLa, T-lymphoblast, KE37, non-mitotic fly brain, and fly embryo cells (Cai et al, 2018a; Eltsov et al, 2018; Fatmaoui et al, 2022; Hou et al, 2023). In contrast, nucleosome stacking was not observed in either budding or fission yeast in situ (Cai et al, 2018b; Tan et al, 2023), and has not been reported for MEFS (Wang et al, 2024). In summary, these observations show that extensive nucleosome stacking is favorable both in vitro and ex vivo under some buffer conditions; limited nucleosome stacking is favorable in situ in RPE-1 cells; and that nucleosome stacking is unfavorable in situ in yeasts.

### Nucleosome DNA gyres interact non-randomly with nuclear complexes

The detection of only one group of ordered dinucleosome class averages (stacked, Type II) indicates that other kinds of inter-nucleosome interactions—for example those required for side-by-side ordered interactions—are too few to manifest as an unambiguous class average. We did, however, detect multiple mononucleosome classes that have a gyre-proximal density of unknown composition. These densities are not distributed uniformly around the nucleosome disc; instead, their centers of

masses occupy ~7 SHL positions. If these densities did occupy all possible positions around the gyres, then we probably would not be able to detect them because of their increased positional heterogeneity. Many nuclear complexes were shown to interact with the nucleosome gyres, including transcription factors (Michael et al, 2020; Nishimura et al, 2022; Tanaka et al, 2020; Wang et al, 2021a), remodelers (Armache et al, 2019; Bacic et al, 2021; Baker et al, 2021; Chittori et al, 2019; Farnung et al, 2020; Farnung et al, 2017; Liu et al, 2017; Nodelman et al, 2022; Yan et al, 2019), histone-mark interactors (Finogenova et al, 2020; Grau et al, 2021; Kasinath et al, 2021; Liu et al, 2021; Poepsel et al, 2018) and RNA polymerase (Ehara et al, 2019; Ehara et al, 2022; Farnung et al, 2022; Farnung et al, 2018; Kujirai et al, 2018). Other gyre-interacting complexes include those that either ensconce much of the nucleosome or that bind to a distorted nucleosome (Chen et al, 2022b; Dodonova et al, 2020; Eustermann et al, 2018; Farnung et al, 2021; Han et al, 2020; He et al, 2020; Liu et al, 2020; Mashtalir et al, 2020; Patel et al, 2019; Wang et al, 2023; Wang et al, 2022b; Yuan, 2022; Willhoft et al, 2018). Our inspection of remapped subtomogram averages suggests that both nucleosome and non-nucleosome complexes may contribute to these gyre-proximal densities. Because many of these complexes are related to transcription, which is repressed in metaphase and active in G1, we speculate that nucleosomes may contribute more to the gyre-proximal densities in metaphase than in G1 cells.

### Canonical nucleosomes are located in irregular domains in G1 but not metaphase

In RPE-1 G1 cell nuclei, most canonical nucleosomes are concentrated within the conspicuous chromatin domains. These domains have been seen in textbook traditional electron-microscopy images and have been studied with renewed interest using advanced light microscopies (Barth et al, 2020; Cho et al, 2022; Fang et al, 2018; Li et al, 2021; Miron et al, 2020; Nagashima et al, 2019; Nozaki et al, 2017; Otterstrom et al, 2019; Xiang et al, 2018; Xu et al, 2018). Consistent with these super-resolution studies, chromatin domains have irregular shapes throughout the G1 RPE-1 nucleus, in the perinuclear region in a cryotomogram of an interphase HeLa cell cryolamella (Cai et al, 2018a), electron spectroscopic images of mouse fibroblast cells (Strickfaden et al, 2020), and near the nuclear periphery of MEFs (Wang et al, 2024). In contrast, domain-like features are not seen in the popular model yeasts *S. cerevisiae* and *S. pombe*, which have uniform chromatin distributions throughout the interphase nucleoplasm (Cai et al, 2018b; Chen et al, 2016; Tan et al, 2023). Chromatin

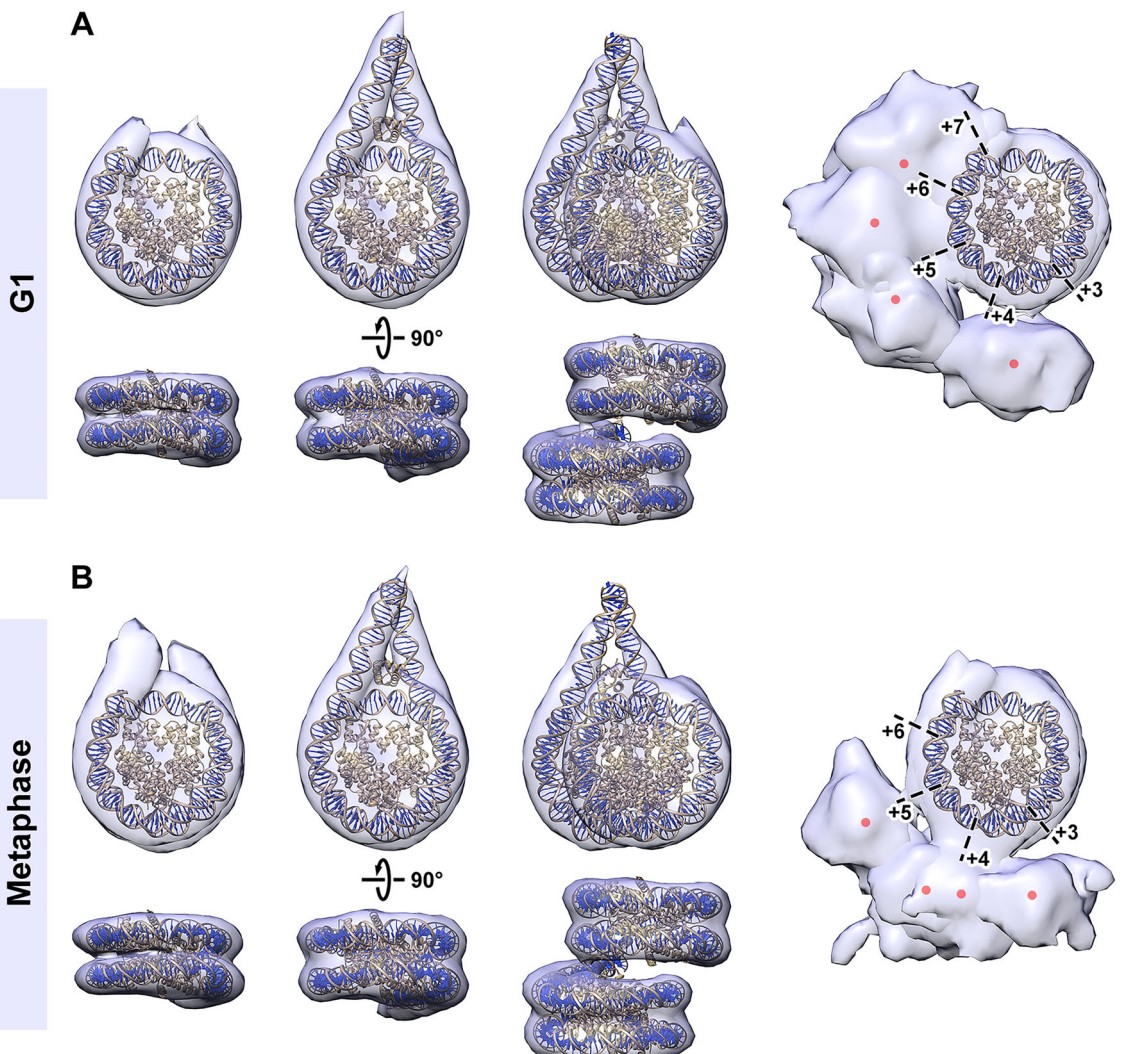

**Figure 7. Comparison of G1 and metaphase nucleosomes.**

Crystal structures of the nucleosome core particle (PDB 1AOI) and the chromatosome (PDB 5NL0) (Garcia-Saez et al, 2018) docked into the in situ nucleosome subtomogram averages (transparent blue) with shorter and longer ordered linker DNA, respectively, for (A) G1 and (B) metaphase nucleosomes. For the mononucleosome and chromatosomes averages, the subtomograms from group 1 were pooled as either the shorter or longer linker nucleosomes (separately for G1 and metaphase), then jointly refined. The DNA is rendered dark blue and gold while the histones are rendered gold. The histone N-terminal tails have been omitted for clarity. For the model containing the chromatosome, the linker histone H1 was included, though the resolution of the density map is insufficient to resolve the linker histone. The nucleosome density envelopes in the right column appear larger than in the other columns because a lower contour level is used to visualize the gyre-proximal densities. Dashed black lines and numbers indicate the super-helical locations. They also have poorer definition than in earlier figures because multiple classes are superposed here. The red dots are the approximate centers of masses of gyre-proximal densities.

**Table 1. Relative abundance of ordered nucleosome species.**

|          | NCP  | Di-NCP | Proximal |
|----------|------|--------|----------|
| G1 phase | 1.00 | 0.22   | 0.49     |
| Metaphase| 1.00 | 0.27   | 0.37     |

NCP = mononucleosomes (all linker DNA lengths), Di-NCP = stacked dinucleosomes, Proximal = nucleosomes with proximal density. The ratio is normalized to the number of NCP particles within each cell-cycle state.

domains were not reported in a cryo-ET study T-lymphoblasts (Hou et al, 2023).

Single-molecule localization studies have revealed the presence of domain-like features in mitotic cells. PALM imaging of mitotic mammalian cells revealed domains ranging from ~140 to 200 nm. (Nozaki et al, 2017). STORM imaging of histones in fixed mitotic cells (Xu et al, 2018) and of DNA in isolated metaphase chromosomes (Wang et al, 2022a) have revealed "nanodomains"—small ~50- to 90-nm granular features corresponding to locally concentrated chromatin. In our cryotomograms of metaphase cells, the canonical nucleosomes are uniformly distributed within the compacted chromosomes (with the exception of pockets; see below), i.e., we did not observe separate domain-like features within the metaphase chromatin. Human metaphase chromatin has also been studied in situ in 3-D by Chrom-EMT (Ou et al, 2017) and cryo-ET of cryo-FIB-milled cells (Zhao and Jensen, 2022). Neither of these EM-based studies reported any nanodomain-like

features in mitotic chromatin. It is unclear what factors explain the visibility of mitotic chromosome domains and nanodomains in light-microscopy but not in electron microscopy data.

The size and shapes of oligonucleosome condensates have been characterized by both light and electron microscopy (Gibson et al, 2019; Maeshima et al, 2016; Sanulli et al, 2019; Strickfaden et al, 2020; Zhang et al, 2022). Condensates of dodecameric arrays can grow to more than 4 μm (Gibson et al, 2019; Sanulli et al, 2019), which are an order of magnitude larger than G1 chromatin domains but the same order as mitotic chromosomes. Furthermore, condensates of reconstituted arrays have circular profiles, presumably because they are 2-D projections of spherical bodies (Strickfaden et al, 2020). In contrast, condensates of fractionated (natural) chromatin are irregular (Maeshima et al, 2016). Subsequent studies using defined '601' chromatin arrays confirmed that longer arrays (with 62 nucleosome repeats) generate irregular condensates (Chen et al, 2022a) whereas arrays with 17 or fewer nucleosomes form rounded condensates (Chen et al, 2022a; Gibson et al, 2023). Condensates of reconstituted tetranucleosomes appear irregular during the early, spinodal phase of in vitro condensation and then fuse into larger round ones (Zhang et al, 2022). In summary, in vitro condensates of either natural chromatin, long artificial arrays, or tetranucleosomes at the early stages of compaction more closely resemble in situ G1 chromatin domains than condensates of (short) dodecameric arrays. It remains unknown how other nuclear factors, like histone post-translational modifications, remodelers, and chaperones would affect in vitro condensates.

## Large-scale nucleosome packing is irregular in G1 and metaphase

The earliest in situ cryo-EM study of mammalian chromosomes was done on Chinese hamster ovary and HeLa cells. This study proposed that the mitotic chromatin inside these cells is irregular and "liquid-like" (McDowall et al, 1986), not highly ordered like 30-nm fibers (Finch and Klug, 1976). Our cryo-ET results here show that RPE-1 chromatin in both G1 and metaphase is also irregular. We did not observe 30-nm fibers, which have unmistakable dimensions and compact shapes in cryotomograms (Scheffer et al, 2011). The RPE-1 data reported here therefore adds to the growing list of in situ cryo-EM and cryo-ET studies of in picoalgae, budding yeast, fission yeast, HeLa, T-lymphoblast, MEFs, and fly neuron cells (Cai et al, 2018a; Cai et al, 2018b; Chen et al, 2016; Eltsov et al, 2018; Eltsov et al, 2008; Fatmaoui et al, 2022; Gan et al, 2013; Hou et al, 2023; Wang et al, 2024), which all report that frozen-hydrated eukaryotic chromatin packing is irregular in situ.

An earlier super-resolution study of both human and mouse cells reported an organizational principle called the "clutch" (Ricci et al, 2015). In the clutch, 3–300 nucleosomes are locally compacted into heterogeneous nanodomains. The mostly irregular organization or chromatin domains in our G1 RPE-1 cells is largely compatible with the clutch model. Likewise, all previous and current cryo-EM and cryo-ET studies of mammalian chromatin in situ report irregular nucleosome packing and are therefore compatible with the clutch model (Cai et al, 2018a; Eltsov et al, 2018; Eltsov et al, 2008; Fatmaoui et al, 2022; Hou et al, 2023; McDowall et al, 1986; Wang et al, 2024). In these studies, the number of nucleosomes per domain was not characterized, and

were mostly done on cell types that differ from Ricci's work, so it is unclear whether the number of nucleosomes per clutch observed by super-resolution microscopy is compatible with the number of nucleosomes per chromatin domain seen in cryotomograms.

Recent studies of purified metaphase chromosomes proposed additional models of higher-order chromatin structure: metaphase chromatin plates in frozen-hydrated samples, in which nucleosomes are packed edge-to-edge in monolayers (plates), with interdigitation between adjacent plates (Chicano et al, 2019), and 100- to 200-nm periodic structures in fixed and heavy-metal-stained samples (Hayashida et al, 2021), in which the repeat direction is parallel to the chromatid long axis. Our metaphase-cell data did not show evidence of edge-to-edge packing or 100- to 200-nm periodic structures. The simplest explanation for these differences is from structural changes induced by sample preparation (isolated vs. in situ chromosomes). Our data does not rule out the alternative possibility in which the plates are irregular, with nucleosomes having large deviations relative to the ordered edge-to-edge arrangement depicted in the earlier study. For the 100- to 200-nm periodic structures, we do not know alternative hypotheses to explain their absence in our—and others'—cryo-EM and cryo-ET data.

A cryo scanning transmission electron microscopy cryotomography study proposed another higher-order chromatin model that involves long-range ordered packing: sequential nucleosomes make a continuous stack in both interphase and mitosis (Sedat et al, 2022a; Sedat et al, 2022b). In this model, termed "slinky", the linker DNA follows a coiled path along stacks of consecutive nucleosomes akin to the hypernucleosomes of archaeal chromatin (Bowerman et al, 2021; Henneman et al, 2018), which was also seen in reconstituted telomere-like chromatin arrays (Soman et al, 2022). This slinky is proposed to coil into 100- to 300-nm gyruses, leaving a nucleosome-free tunnel in the middle. None of our class averages have the key feature needed for slinkies: linker DNA that is curved enough to connect to sequential nucleosomes in the stack. Furthermore, our cryotomograms do not show any large nucleosome-free regions in either the chromatin domains or metaphase chromosomes that would account for the hole in the middle of a slinky gyrus.

## Non-nucleosome features of G1 nuclei and metaphase chromatin

We previously observed irregular intranuclear regions that we termed "pockets" in cryotomograms of *S. pombe* cells. Pockets are conspicuous because they lack macromolecular complexes and megacomplexes. In G1 chromatin domains, we did not find any instances of pockets, but we did find a few unambiguous ones in metaphase chromatin. With the exception of pockets, the RPE-1 metaphase chromatin distribution is uniform and resembles that seen in other cryo-EM and cryo-ET studies of mitotic human cells (Eltsov et al, 2008; Zhao and Jensen, 2022). We did not detect mesh/network-like nucleosome-free regions analogous to the "reticular pattern" of negatively stained cells in a previous study (Ou et al, 2017). Nevertheless, the presence of pockets in metaphase chromosomes suggests that the compaction is not maximized.

No megacomplexes were seen in the interior of the chromatin domains or mitotic chromatin, indicating that the crowding disfavors access by very large complexes. In contrast, mitotic

budding and fission yeasts both have megacomplexes mixed with the chromatin (Cai et al, 2018b; Chen et al, 2016; Ng et al, 2019). This molecular-packing difference may be related to differences in mitotic transcription: both yeasts have high levels of mitotic transcription (Cho et al, 1998; Oliva et al, 2005; Peng et al, 2005; Rustici et al, 2004; Spellman et al, 1998) whereas mammalian chromatin undergoes genome-wide transcriptional repression (Flemming, 1880; Prescott and Bender, 1962; Taylor, 1960). Note that mammalian mitotic chromatin exhibits transient access to small complexes such as transcription factors (Teves et al, 2016) and permits very low levels of mitotic transcription (Palozola et al, 2017). High-resolution correlative light microscopy and cryo-ET analysis will be needed to relate the 3-D structure of the nucleosomes to the transcriptional state.

## The G1 nucleoplasm has many heterogeneous structures

Unlike the canonical-nucleosome-rich chromatin domains, the chromatin-poor G1 nucleoplasm is constitutionally heterogeneous. Our subtomogram analysis did not produce canonical nucleosome class averages in the nucleoplasm. There are at least two distinct possible explanations. If the nucleoplasmic nucleosomes have the canonical structure but too few, then the resultant class averages would be either too noisy to be recognizable as canonical

nucleosomes or may be mis-classified. Alternatively, if the nucleoplasmic nucleosomes are abundant but have non-canonical structures, then we would miss them because we do not yet know their structures. The linker-filament structures remain to be determined (Li et al, 2021; Miron et al, 2020).

Nuclear bodies that are larger than nucleosomes are abundant in the nucleoplasm. The most conspicuous nucleoplasmic features are the megacomplexes, which have been seen in HeLa and yeast nuclei (Cai et al, 2018a; Cai et al, 2018b; Chen et al, 2016; Ng et al, 2019). Some of the megacomplexes are preribosomes because they have the expected size (~20 to 30 nm) (Erdmann et al, 2021; Lau et al, 2021). Indeed, we have detected one class average that resembles the large ribosomal subunit. The identities of the vast majority of megacomplexes remain to be determined.

The G1 nucleoplasm also contains dense irregular bodies, which have dimensions that exceed 40 nm. Dense irregular bodies are also present in our cryotomogram of an interphase HeLa cell (Cai et al, 2018a) and in *S. cerevisiae* cells (Ng et al, 2019), though we previously did not appreciate their differences from megacomplexes. No such bodies have been observed in *S. pombe* yet. Dense irregular bodies are too big to be preribosomes and are unlikely to harbor canonical nucleosomes because they lack the characteristic double-gyre motifs. Further work is needed to determine the identity of these dense irregular bodies and their relationship to

**Table 2. Summary of chromatin and nuclear features in situ.**

| Structural feature | RPE-1 G1 | RPE-1 Meta | HeLa[a] | Yeast[a] |
|---|---|---|---|---|
| ≥100 nm | | | | |
| Mitotic chromatin plates | n/a | Not seen | n/a | Not seen |
| Arc-shaped stacks/slinkies | Not seen | Not seen | Not seen | Not seen |
| 100–200 nm periodic structure | Not seen | Not seen | Not seen | Not seen |
| Interphase domain shape | Irregular | n/a | Irregular | n/a |
| 20–80 nm | | | | |
| Chromatin packing | Irregular | Irregular | Irregular | Irregular |
| Ordered trinucleosome stacks | Not seen | Not seen | Not seen | Not seen |
| Slinky oligonucleosomes | Not seen | Not seen | Not seen | Not seen |
| Megacomplexes (in chromatin) | Rare | Rare | Rare | n/a |
| Megacomplexes (in nucleoplasm) | Abundant | n/a | Abundant | Abundant |
| Dense irregular bodies | Rare | Not seen | Rare | in *S. cere* |
| Pockets (mitotic chromatin) | n/a | Rare | n/a | in *S. pombe* |
| Pockets (G1 chromatin domain) | Ambiguous | n/a | n/a | n/a |
| 10–20 nm | | | | |
| Ordered dinucleosome stacks[b] | Abundant | Abundant | Not seen | Not seen |
| Gyre-interacting densities[b] | Abundant | Abundant | n/d | n/d |
| Canonical mononucleosome | Abundant | Abundant | Abundant | Rare |
| <10 nm | | | | |
| Ordered linker DNA length | Variable | Variable/no ultrashort | Long & short | Long & short |
| Linker DNA length symmetry | Asymmetric & symmetric | Asymmetric & symmetric | Asymmetric | Asymmetric |

The chromatin and nuclear structural features (approximate size ranges in nanometers) of RPE-1 G1 phase (G1), Metaphase (Meta), HeLa interphase, and *S. pombe* and *S. cerevisiae* in interphase and mitosis (Yeast) are listed.
[a]The HeLa phenotypes are from (Cai et al, 2018a) while the yeast phenotypes are from (Cai et al, 2018b; Chen et al, 2016; Ng et al, 2019; Tan et al, 2023); note that these earlier studies have a smaller sample size of canonical nucleosomes than here. "Not seen" means that the expected structures were absent in our dataset. We cannot rule out that they exist at low abundance. n/a: not applicable. n/d: not determined due to sample size issues.
[b]Some ordered higher-order nucleosome structures may be sensitive to cryoprotectants or image contrast effects.

megacomplexes (if any), diverse nuclear condensates that have been studied by light microscopy (Stanek and Fox, 2017; Zhu and Brangwynne, 2015), or ribonucleoprotein complexes (Pacheco-Fiallos et al, 2023).

## Metaphase chromosomes have a heterogeneous, ribosome-rich periphery

The metaphase chromosome's surface, termed the chromosome periphery, is associated with non-nucleosomal complexes such as nucleolar and ribosomal proteins and RNAs (Booth et al, 2016; Nishino et al, 2012; Ohta et al, 2010; Stenstrom et al, 2020). Ribosomes are present on the surface of metaphase chromosomes, reminiscent of those seen adjacent to HeLa and U2OS metaphase chromosomes in situ (Eltsov et al, 2008; Zhao and Jensen, 2022) and the ones that co-purify with isolated chromosomes (Nishino et al, 2012). The co-purified ribosomes generate a ~30-nm peak in small-angle X-ray scattering experiments that was previously (and erroneously) attributed to 30-nm chromatin fibers (Langmore and Paulson, 1983; Nishino et al, 2012). The chromosome periphery is organized by the proliferation marker Ki-67, which extends ~100 nm from the chromatin surface and keeps the chromosomes individualized via a surfactant function (Booth et al, 2014; Cuylen et al, 2016). Even though the Ki-67 protein (>300 kDa) is more massive than a nucleosome (200 kDa)—much of its sequence is predicted to be intrinsically disordered (Remnant et al, 2021), meaning that it has an extended structure. Long peptide-wide proteins are too thin to see in cryotomograms of cells, which would explain the absence of filamentous densities in chromosome-periphery regions of our cryotomograms. Instead, our cryotomograms show that the macromolecular contents of the nuclear periphery are both heterogeneous and lack order, consistent with the disordered nature of Ki-67.

## Conclusion

Our study has revealed in situ similarities and differences of G1 and metaphase chromatin across multiple length scales (Table 2), ranging from hundreds of nanometers (domains and chromatids) to tens of nanometers (oligonucleosomes, megacomplexes), to the nanometer scale (mononucleosomes, linker DNA). Both G1 and metaphase chromatin are enriched in canonical nucleosomes with variable linker-DNA length and the linker-DNA symmetry. The DNA gyres make non-random interactions at several super-helical locations. Both cell types have abundant nucleosome stacking, which are more ordered in DMSO cryoprotectant. Chromatin is largely devoid of megacomplexes. Metaphase canonical nucleosomes are uniformly distributed without forming chromatin domains within, which raises the question of how compaction and decompaction proceed in terms of interphase chromatin domains. Improvements in sample preparation and image analysis may enable deeper structural cell-biological characterization. For example, the identification of more nucleosomes may allow us to estimate the in situ nucleosome concentration, which was recently reported to be ~750 μM in a volume electron-microscopy study of mitotic RPE-1 cells (Cisneros-Soberanis et al, 2024), and also reveal the fold of nucleosomes, which was reported to follow an irregular zig-zag path in T-lymphoblasts (Hou et al, 2023).

## Methods

### Reagents and tools table

| Reagent/Resource | Reference or Source | Identifier or Catalog Number |
|---|---|---|
| **Experimental models** | | |
| hTERT-RPE-1 cells | ATCC | CRL-4000 |
| **Recombinant DNA** | | |
| N/A | | |
| **Antibodies** | | |
| Alexa Fluor® 647 anti-histone H3 (phospho S10) | Abcam | ab196698 |
| Rabbit polyclonal Anti-phospho-Histone H3 (Ser10) | Abcam | ab5176 |
| Mouse anti-rabbit IgG-HRP | Santa Cruz | sc-2357 |
| Rabbit polyclonal to beta Tubulin | Abcam | ab6046 |
| m-IgGκ BP-HRP | Santa Cruz | sc-516102 |
| cyclin B1 (GNS1) mouse monoclonal IgG1 | Santa Cruz | sc-245 |
| Anti-RPB1 CTD(S5P) | Abcam | ab5131 |
| Anti-RPB1 CTD(S2P) | Abcam | ab5095 |
| **Oligonucleotides and other sequence-based reagents** | | |
| N/A | | |
| **Chemicals, Enzymes and other reagents** | | |
| **Chemicals** | | |
| 4′,6-diamidino-2-phenylindole (DAPI) | TFS | D1306 |
| Dimethyl sulfoxide (DMSO) | Sigma | 472301 |
| DMEM/F-12 with GlutaMAX | Gibco/TFS | 10565-018 |
| Fetal bovine serum | Sigma | F9665 |
| Glycine | Bio-Rad | 1610718 |
| MG132 | Merck | M7449-1ML |
| Nocodazole | Merck | SML1665-1ML |
| Palbociclib | Selleckchem | S1116 |
| Paraformaldehyde | EMS | 15714 |
| Penicillin-streptomycin | Gibco | 15140-122 |
| Phosphate-buffered saline (PBS) | Vivantis | PB0344-1L |
| PBS-T (PBS + 0.1% Tween 20) | Sinopharm | T20087687 |
| Prolong Gold antifade reagent | TFS | P36930 |
| Taxol | Santa Cruz | 33069-62-4 |
| Triton X-100 | Alfa Aesar | A16046 |
| **Chromatin** | | |
| HeLa oligonucleosomes | EpiCypher | 16-0003 |

| Reagent/Resource | Reference or Source | Identifier or Catalog Number |
|---|---|---|
| **Software** | | |
| Adobe Illustrator | Adobe | |
| Adobe Photoshop | Adobe | |
| Adobe Premiere Pro | Adobe | |
| Auxiliary scripts | Gan lab, github.com/anaphaze/ot-tools | |
| BioRender | BioRender | |
| Bsoft | lsbr.niams.nih.gov/bsoft (Heymann and Belnap, 2007) | |
| cryo-CARE | github.com/juglab/cryoCARE_pip (Buchholz et al, 2019) | |
| EMAN2 | blake.bcm.edu/emanwiki/EMAN2 (Chen et al, 2017) | |
| FIJI | imagej.net/software/fiji (Schindelin et al, 2012) | |
| Google sheets | Google | |
| IMOD | bio3d.colorado.edu/imod (Mastronarde, 1997) | |
| PEET | bio3d.colorado.edu/PEET (Nicastro et al, 2006) | |
| RELION | github.com/3dem/relion (Scheres, 2012) | |
| SerialEM | bio3d.colorado.edu/SerialEM (Mastronarde, 2005) | |
| UCSF Chimera | www.cgl.ucsf.edu/chimera (Pettersen et al, 2004) | |
| **Other** | | |
| **Light microscopy** | | |
| LSM900 AiryScan microscope | Zeiss | N/A |
| 20×, 0.8 N.A. Plan-Apochromat | Zeiss | N/A |
| 63×, 1.4 N.A. oil Plan-Apochromat | Zeiss | N/A |
| ZEN 3.0 (blue edition) | Zeiss | N/A |
| Eclipse Ti microscope | Nikon | N/A |
| Plan Fluor 10x Ph1 objective | Nikon | N/A |
| 20 mm Ø coverslip | Paul Marienfeld GmbH | 0112600 |
| **Electron cryomicroscopy** | | |
| C-flat 2/4 200 mesh, gold | Protochips | CF-2/4-2Au |
| C-Clip Ring (autogrid) | TFS | 1036173 |
| CryoFIB autogrid | TFS | 1205101 |
| C-flat 2/4 200 mesh, copper | Protochips | CF-2/4-2C |
| Continuous carbon | EMS | CF200-Cu-UL |
| Whatman Grade 1 filter paper | Whatman | 1001-055 |
| Copper tubes, 0.3 mm inner diameter | Wohlwend | N/A |

| Reagent/Resource | Reference or Source | Identifier or Catalog Number |
|---|---|---|
| Copper tube loading tool | Wohlwend | Part 733-1 |
| Copper tube cutting device | Wohlwend | Part 732 |
| UC7/FC7 Microtome | Leica | N/A |
| 35° diamond knife | Diatome | Cryo35 |
| Micromanipulator, Leica | Leica | N/A |
| Micromanipulator, MN-151-S | Narishige | N/A |
| Vitrobot Mark IV | TFS | N/A |
| Helios Nanolab 650 FIB-SEM | TFS | N/A |
| PolarPrep 2000 Cryo Transfer System | Quorum | N/A |
| Titan Krios G1 cryo-TEM | TFS | N/A |
| Falcon II camera | TFS | N/A |
| Gatan K3 Summit camera | AMETEK | N/A |
| Gatan BioContinuum imaging filter | AMETEK | N/A |

## Cell culture

hTERT-RPE-1 cells (human retina epithelium cells immortalized with hTERT; CRL-4000™) were obtained from ATCC (Manassas, VA). The cells were grown in an incubator set to 37 °C and 5% $CO_2$. The cells were cultured in complete medium consisting of DMEM/F-12 with GlutaMAX™ medium (Gibco, 10565-018; Thermo Fisher Scientific (TFS), Waltham, MA) supplemented with 10% fetal bovine serum (Sigma, F9665) and 1% penicillin-streptomycin (Pen-strep) (Gibco, 15140-122). Cell washing was done with PBS+Ca+Mg, consisting of Phosphate-buffered saline (PBS, Vivantis, PB0344-1L; Vivantis Technologies Sdn. Bhd., Selangor Darul Ehsan, Malaysia), 0.1 mg/mL $CaCl_2$ (Sigma, 12095), and 0.1 mg/mL $MgCl_2$ (Sigma, M8266-100G). Details of all the reagents and equipment used are summarized in Appendix Table S1.

## Taxol-arrested cells for pilot experiments

Taxol-arrested cells were prepared following our previous work (Ruan et al, 2019). The cells were first synchronized in G1 as follows. Cells were seeded in culture flasks or dishes and incubated for 24 h. They were then subject to depleted serum culturing for 24 h. The cells were then further treated with 3 mM thymidine for another 24 h. To arrest the cells in mitosis, they were released from the thymidine block for 4 h by replacement of fresh culture medium. Next, the cells were incubated with 800 μM Taxol for 12 h. The mitotic cells were collected by shake-off.

## G1 and metaphase cell-cycle arrest for both LM and cryo-ET of lamellae

The cell-cycle synchronization was done following a published protocol (Scott et al, 2020), with modifications. Adherent cells

(destined for G1 arrest) were prepared differently for confocal microscopy versus cryo-ET. For confocal experiments, RPE-1 cells were seeded in a 35 mm μ-dish at a density of $4.8 \times 10^5$ cells/mL. After a 24-h incubation, 1000× palbociclib (Selleckchem, S1116; Selleck Chemicals, Houston, TX), dissolved in 0.22 μm filtered Milli Q water, was added to the medium to a final concentration of 1 μM and then incubated for a further 18 h. For cryo-ET experiments, gold grids were placed on the cover of a 35 mm Petri dish and exposed to UV light for 15 min in a biosafety cabinet. Then the gold grids were washed by incubating 10 min in complete medium + Pen-strep in a 35 mm dish, in a shaker at 20 RPM. The complete medium was then discarded and replaced with fresh medium. To arrest cells in G1 phase, RPE-1 cells were seeded in a 35 mm Petri dish where two gold grids were positioned in advance at a density of $2.0 \times 10^5$ cells/mL. After a 24-h incubation, 1000× palbociclib was added to the medium to a final concentration of 1 μM and then followed by an 18-h incubation. Then, the grids were picked up and immersed in DMEM/F-12 with GlutaMAX™ medium containing 9% DMSO (Sigma, 472301) for 5 s as cryoprotectant before plunge-freezing (see below).

Metaphase cells were synchronized the same way for all experiments (confocal microscopy, immunoblots, cryo-ET). RPE-1 cells were plated into a 25 cm² flask at a density of $8.0 \times 10^4$ cells/mL. After a 24-h incubation, palbociclib was added to a final concentration of 1 μM and the cells were then incubated for 18 h (as above). The medium was then pipetted out and the cells were rinsed three times with 1 mL pre-warmed PBS+Ca+Mg. The cells were then cultured in fresh complete medium for 8 h. Then nocodazole (Sigma, SML1665) (5 mg/mL stock in DMSO) was added into the medium to a final concentration of 50 ng/mL. After a 12-h incubation, the cells were harvested by both shaking the flask and tapping it horizontally against the bench. Next, the cell suspension medium was washed four times by centrifugation at $1000 \times g$ for 3 min, decanting the supernatant, and cell-pellet resuspension with 3 mL PBS+Ca+Mg. The last resuspension was done in complete culture medium with 10 μM MG132 (Sigma, M7449). After a 70-min incubation, the cells were detached by shake off and flask tapping (see above), centrifuged at $1000 \times g$ for 3 min. The cell pellet is used for the subsequent experiments.

## Immunoblots

Both nocodazole- and MG132-arrested cells were prepared as above. Nocodazole-arrested cells were equally divided into four tubes, corresponding to: no washout and 30, 60, and 120 min after nocodazole washout. In a separate experiment, MG132-arrested cells were also equally divided into four tubes, corresponding to: no washout and, 30, 60, and 120 min after MG132 washout. The cell washes were done with PBST that was pre-warmed to 37 °C in a water bath. Time zero of the nocodazole-washout experiment corresponds to when the first wash started. All cells were washed with ice-cold PBST before the addition of lysis buffer. Ice-cold protein extraction lysis buffer was composed of RIPA (Sigma, R0278), Protease inhibitor cocktail (Sigma, P8340) and Phosphatase Inhibitor Cocktail 3 (Sigma, P0044) with a volume ratio of 100:1:1. One hundred μL lysis buffer was added per $1.0 \times 10^6$ cells and mixed by vortexing for 6 s. In total, 5 vortex cycles were done with a 6 min wait between each vortex. The lysate mixture was centrifuged at $16,000 \times g$ for 20 min. The supernatant was aspirated into a clean tube and stored in −80 °C until all groups were ready

for total protein concentration determination via BCA assay (TFS, 23227). An equal volume of 2× Laemmli buffer was added into the cell lysates and mixed by vortexing briefly for less than 5 s. The mixture was boiled at 95 °C for 5 min, then centrifuged at $16,000 \times g$ for 5 min. The denatured protein mixture was divided into aliquots of about 20 μg for either immediate electrophoresis or storage in −80 °C.

Twenty μg of protein was loaded in each well of a Mini-protean TGX gel (Bio-Rad 4561083; Hercules, CA) with Precision Plus Protein Dual Color Standards (Bio-Rad 1610374) or MagicMark™ XP Western Protein Standard (TFS, LC5602) as the protein ladder. Electrophoresis was done using constant voltage at 90 V for 30 min, followed by 120 V for another 30 min until the blue dye almost reached the bottom of the gel. The ladder and proteins in the gel were transferred to the PVDF membrane (Bio-Rad 1704156) by the semi-dry method, with constant current 2.5 A for 3 min. The membrane was blocked in 1× TBST (Tris-buffered saline with 0.1% Tween 20) containing 5% non-fat milk at 23 °C for 1 h. Next, the membrane was rinsed with TBST for 5 s, then incubated with primary rabbit polyclonal antibody to beta tubulin (Abcam ab6046; Cambridge, United Kingdom), rabbit polyclonal anti-phospho-histone H3 (Ser10) antibody (Abcam, ab5176), or cyclin B1 (GNS1) mouse monoclonal IgG1 (Santa Cruz, sc-245; Dallas, TX) overnight at 4 °C. All antibodies were diluted 1:500 in blocking buffer. After three sequential 5-min washes with TBST, the membrane previously incubated with anti-Cyclin B1 antibody was incubated with m-IgGκ BP-HRP secondary antibody (Santa Cruz, sc-516102) for 1 h at 23 °C, and membranes previously incubated with anti-beta tubulin antibody or anti-phospho-histone H3 (Ser10) antibody were incubated with mouse anti-rabbit IgG-HRP secondary antibody (Santa Cruz, sc-2357) for 1 h at 23 °C. After 3 sequential 5-min washes in TBST, the membranes were incubated with Clarity Western ECL substrate solution (Bio-Rad 1705061) at 23 °C for 5 min, then imaged with ImageQuant LAS4000 system (Cytiva, Marlborough, MA). Full, uncropped blots are shown in Appendix Fig. S31. A summary of all the antibodies used is in Appendix Table S2.

## Apoptosis control experiments

RPE-1 cells were incubated in complete medium containing 1 μM staurosporine for 6 h to induce apoptosis as a positive control. To control for the carrier, another group of RPE-1 cells was incubated in complete medium containing 0.1% DMSO for 6 h. Five more flasks of RPE-1 cells were arrested in G1 phase. One flask did not have any cryoprotectant added and was used as the negative control while the other four flasks were treated with cryoprotectants. To test for apoptosis induction by cryoprotectants, flasks of G1 cells were incubated with either 4 mL 9% DMSO or 4 mL 9% glycerol in DMEM/F12 medium. These treatments lasted either 1 min or 10 min and were stopped by discarding the cryoprotectant-containing medium and rinsing with 2 mL cold PBS 3 times. For apoptosis analysis in metaphase cells, five flasks of metaphase-arrested RPE-1 cells were aliquoted into 15 mL tubes. The cells destined for cryoprotectant treatment were centrifuged at $200 \times g$ for 3 min, then the supernatant was removed via pipetting. Next the cell pellets were resuspended in 0.5 mL DMEM/F12 medium containing either 9% DMSO or 9% glycerol. These cells were incubated for either 1 min or 10 min. The cryoprotectant was

achieved by dilution with 5 mL PBS followed by three washes with 2 mL cold PBS. After the washes, the proteins were extracted and analyzed by immunoblots with rabbit monoclonal antibodies against PARP1 (Abcam, ab191217) and with immunoblots against beta tubulin as the loading control.

## Conventional fluorescence microscopy

For morphological analysis, RPE-1 cells were imaged using a Nikon Eclipse Ti microscope (Nikon Instruments Inc., Tokyo, Japan) equipped with a Plan Fluor 10× Ph1 objective lens. To obtain fields of view of multiple cells, Z-stack images were acquired on an LSM 900 AiryScan microscope controlled by ZEN 3.1 software, using a 20× 0.8 NA Plan-Apochromat under confocal mode (Zeiss, Jena, Germany). Z-projection at maximum intensity was done in FIJI (Schindelin et al, 2012).

## Immunofluorescence

All washes were done as follows. Attached cells were washed 3 times with PBST (PBS+Ca+Mg with 0.1% Tween 20 (Sinopharm, T20087687, Beijing, China)) for 5 min on a shaker at 20 RPM. Suspended cells were washed 3 times with PBST for 5 min, except the first wash after fixation (see fixation protocol below), in which PBST was replaced with PBST containing 0.5% BSA (Sigma, A9647). For each wash, the suspended cells were centrifuged at $600 \times g$ for 3 min, then supernatant was decanted, the cell pellet was resuspended with 1 mL PBST using a P1000 micropipette, and then incubated 2 min on a rotator at 20 RPM.

Palbociclib-synchronized G1 (attached) cells were washed three times with PBS+Ca+Mg after the completion of the drug treatment. The cells were then fixed with 4% paraformaldehyde (Electron Microscopy Sciences, 15714, Hatfield, PA) in PBS for 10 min at 23 °C, then washed 3 times for 5 min each in PBST, then permeabilized with 0.25% Triton X-100 (TFS, Alfa Aesar A16046) in PBS for 10 min at 23 °C. To block non-specific binding, the cells were then washed $3 \times 5$ min with PBST, then incubated with PBST containing 1% BSA and 22.5 mg/mL glycine (Bio-Rad 1610718) for 30 min at 23 °C. Next, the cells were immunostained with primary antibodies: rabbit polyclonal to RNAPII CTD repeat YSPTSPS (phospho S2) (1:300, Abcam ab5095), rabbit polyclonal to RNAPII CTD repeat YSPTSPS (phospho S5) (1:300, Abcam ab5131)), or Alexa Fluor-conjugated antibody (Alexa Fluor® 647) mouse monoclonal anti-histone H3 (phospho S10) antibody (1:250, Abcam ab196698)), in PBST with 1% BSA, on a rotator with slow rotation, overnight at 4 °C. The cells were then incubated with Alexa Fluor 488 goat anti-rabbit IgG (H + L) secondary antibody (1:500, TFS A-11008) in Milli Q purified water containing 1% BSA for 1 h at 23 °C. Finally, the cells were incubated with 300 nM 4′,6-diamidino-2-phenylindole (DAPI, TFS D1306) in PBS for 5 min. After three sequential 5-min washes with PBST, the cells were ready for imaging.

To harvest nocodazole- and MG132-arrested cells after the completion of the drug treatment, the flasks were shaken and tapped against the bench. The detached cells were collected in a 15 mL tube and centrifuged at $600 \times g$ for 3 min. Cell pellets were resuspended with 1 mL PBST and transferred to a 1.5 mL microfuge tube for all following incubations and washes, which are different from the G1 cells (attached to the dish). After the last wash before

imaging, most of the supernatant was removed, leaving about 20 µL. The cell pellets were resuspended in this remaining supernatant with a micropipette using a 200 µL tip. Finally, 5 µL of this cell suspension was mounted on a 20 mm Ø coverslip (Paul Marienfeld GmbH & Co. KG, 0112600, Lauda-Königshofen, Germany) with 15 µL Prolong Gold antifade reagent and then cured 20 to 24 h prior to imaging. All treatments between the harvesting and blocking steps were done in 15 mL conical tubes. After blocking, the washing was done in 1.5 mL microfuge tubes. All reagents including their concentrations and incubation durations were the same as above G1 cells.

To obtain a large field of view containing multiple cells, Z-stacks were acquired with a Zeiss LSM 900 confocal microscope (see above). To observe details of single cells, Z-stacks were acquired with the same microscope, but using a 63 × 1.4 NA Plan-Apochromat oil-immersion objective and operated in Airyscan super-resolution mode. Z-stacks (3-D) were projected using Fiji's "3D Project" brightest point method. All the confocal microscopy details are summarized in Appendix Table S3.

## Self-pressurized freezing

Taxol-treated RPE-1 cells were scraped or shaken off in 1× PBS. The cells were further pelleted by centrifugation at $600 \times g$ for 10 min. For 100 µL cell pellets, 25 µL 50% w/v dextran 6000 (Sigma 31388) was added and mixed gently as the extracellular cryoprotectant, yielding a final concentration of 12.5% dextran. The cells were loaded into copper tubes, which were sealed and then dropped into liquid ethane for self-pressurized freezing.

## Vitreous sectioning

Cryomicrotomy was done in an UC7/FC7 cryomicrotome (Leica Microsystems, Vienna, Austria). RPE-1 cells were cut into 100-nm-thick (nominal) cryosections using a 35° diamond knife (Diatome Cryo35, Nidau, Switzerland). We used the dual-micromanipulator method (Ladinsky et al, 2006; Ng et al, 2020; Studer et al, 2014). The cryosections were attached to a continuous carbon grid (EMS CF200-Cu-UL, Hatfield, PA) using a Crion device (Leica) in "charge" mode.

## Plunge-freezing of oligonucleosomes

HeLa oligonucleosomes (EpiCypher 16-0003, Durham, NC; sold as "polynucleosomes") in storage buffer (20 mM HEPES, pH 7.5, 1 mM EDTA) was diluted 1:1 in a buffer containing 20 mM HEPES-KOH, pH 7.8, 2 mM $MgCl_2$. The additional $MgCl_2$ was added to increase the local oligonucleosome concentration to facilitate targeting but was of insufficient concentration; since the same $MgCl_2$-containing buffer was used in all the cryoprotectant test samples, the conclusions were independent of $MgCl_2$. For preparations containing cryoprotectant, the sample was also supplemented with DMSO to a final concentration of 9% (v/v). Each glow-discharged Quantifoil R2/4 200 mesh copper grid (Quantifoil Micro Tools GmbH, Jena, Germany) was then deposited with 4 µL sample and plunge-frozen in liquid ethane using a Vitrobot Mark IV (TFS) operated at 22 °C and 100% humidity.

## Plunge-freezing of RPE-1 cells

RPE-1 cells were plunge-frozen in 63/37 propane-ethane mixture (Tivol et al, 2008) using a Vitrobot Mark IV (TFS) operated at 100% humidity and 37 °C. For G1-arrested RPE-1 cells that were grown on C-flat 2/4 200 mesh gold grids (Protochips, Morrisville, NC), the grid was held with a Vitrobot tweezer and then immersed for 5 s in DMEM/F12 with GlutaMAX™ medium that was supplemented with either 9% DMSO or 9% glycerol as cryoprotectant. For the no-cryoprotection control, the grid was immersed in fresh DMEM/F12 with GlutaMAX™ only. The grid was then loaded on the Vitrobot and manually blotted from the back using Whatman® Grade 1 filter paper before being plunge-frozen. Note that in the earlier screening experiments to find the optimal cryoprotectant concentration (Appendix Fig. S4), the grids were immersed in PBS+Ca+Mg that were supplemented with 0–9% DMSO. Subsequent sample preparation was performed using DMEM/F12 with GlutaMAX™ instead (see below).

Metaphase-arrested RPE-1 cells were first harvested as a suspension and concentrated to $1.5 \times 10^6$ cells/mL. Immediately before plunge freezing, the cell suspension was mixed with an equal volume of fresh DMEM/F12 with GlutaMAX™ medium containing either 18% DMSO or 18% glycerol, bringing the final DMSO (or glycerol) concentration to 9%. For the no-cryoprotection control, the cells were resuspended in fresh DMEM/F12 with GlutaMAX™ only. 4 μL of the cell suspension was then deposited onto C-flat 2/4 200 mesh gold grids (without any coatings) that were glow discharged at 15 mA for 15 s, before being manually back-blotted and plunge-frozen as described above.

## Cryo-FIB milling

Cryolamellae of plunge-frozen RPE-1 cells were prepared using a Helios NanoLab 600 DualBeam (TFS) equipped with a PolarPrep 2000 Cryo Transfer System (Quorum Technologies, Laughton, UK) (Medeiros et al, 2018). The samples were first coated with a layer of organometallic platinum using an in-chamber gas injection system that was positioned 9 mm from the grid center and operated for approximately 10 to 18 s flow time at 28 °C (Hayles et al, 2007). Cryo-FIB milling was then performed with the Helios stage tilted to 12°. The bulk of the cellular material was first removed using the FIB at 30 kV 2.8 nA, followed by further thinning of the cryolamellae at a lower current of 0.28 nA. The cryolamellae were then polished to a final nominal thickness of ~150 nm at 48 pA.

## Electron cryotomography

A summary of the cryo-ET sample preparation, imaging, and processing parameters are reported in Appendix Table S4. Additional details of each tilt series are reported in Appendix Table S5. Cryo-ET data of both RPE-1 cryolamellae and purified HeLa oligonucleosome samples were collected using SerialEM (Mastronarde, 2005) on a Titan Krios (TFS) equipped with a K3 camera (AMETEK, Berwyn, PA) and a BioContinuum energy filter (AMETEK) operating in zero-loss mode with a 20 eV slit width. Tilt series were acquired in super-resolution mode with a bin ×1 pixel size of 3.4 Å at the specimen level. These data were collected in the dose-symmetric tilt scheme and using either defocus phase contrast at −5 μm defocus or Volta phase contrast at −0.5 μm

defocus (both nominal values). The K3 data was collected after TFS changed their Volta phase plate design, requiring that the phase-plate heater be operated at ~370 mW power.

To speed up the setup of the tilt series collection on SerialEM, we used the Python scripts described by Weis et al (Weis et al, 2021) to generate virtual maps of regions of interest from low-magnification montage images of cryolamellae or grid squares. HeLa oligonucleosome samples were imaged using a tilt range of ±60° and a tilt increment of 2°, for a nominal total dose of 120 e⁻/Å² per tilt series. For cryolamellae samples, a stage pre-tilt of −14° was applied so that the cryolamellae are approximately perpendicular to the electron-optical axis when the first image is recorded. This pre-tilt angle was determined by trial-and-error. Cryolamellae tilt series were collected using a tilt range of −70° to +42° and a 2° tilt increment. The nominal total dose per tilt series for cryolamellae samples was 110 e⁻/Å².

To more accurately target regions of interest for cryolamellae imaging, we first generated longer exposure (~0.1 e⁻/Å² per tile) low-magnification montage images of our samples. Montage tiles were acquired at −90 μm defocus using an unbinned pixel size of 19.2 Å (4800× magnification) and a 2-s exposure time. In this imaging condition, chromosome regions and cytoplasm of metaphase-arrested RPE-1 cells show different density features (Appendix Fig. S24). Microtubules are also frequently seen surrounding the chromosome regions. For G1-arrested cells, nuclear regions were identified based on cytological cues. The nucleus periphery regions were found based on the presence of the nuclear envelope. If the nuclear envelope was not visible, imaging positions were selected based on the absence of the cytoplasmic components listed above.

Super-resolution movie frames were aligned and Fourier cropped to 3.4 Å pixel size using MotionCor2 (Zheng et al, 2017). Tilt series stacks were aligned by patch tracking and reconstructed into cryotomograms at 6.8 Å voxel size using the IMOD program *Etomo* (Kremer et al, 1996; Mastronarde, 1997; Mastronarde and Held, 2017). All subsequent template matching and subtomogram analyses were performed using cryotomograms at 6.8 Å voxel size.

Automated segmentation of chromatin regions in the cryotomograms was performed using EMAN2 (Chen et al, 2017). For the convolutional neural network training, 64 × 64 tiles containing either G1 chromatin domains or metaphase chromosome regions were extracted from the cryotomograms (binned to a voxel size of 13.5 Å) and used as positive training examples. Tiles from the cytoplasm, nucleoplasm and lamella surface contaminants were used as negative examples.

For the cryosection pilot experiments, both defocus phase-contrast and Volta phase contrast tilt series were collected using the same Titan Krios as above, but with a Falcon II camera (TFS) in integration mode at a 12.4 or 7.3 Å pixel size as the specimen level, all controlled with Tomo4 (TFS). A nominal defocus of −10 μm (defocus phase contrast) or −0.5 μm (VPP) was applied. Bi-directional tilt series, starting at 0°, were collected with a ± 60° tilt range and 2° tilt increment. These data were collected before TFS implemented a change in Volta phase plate design, so the phase-plate heater was operated at lower than 150 mW power.

## Denoising for visualization

A subset of cryotomograms were denoised for visualization purposes; the subtomogram analysis steps in the following sections were done on

non-denoised data. MotionCor2 was used to generate tilt series images where either the odd- or even-numbered raw movie frames were excluded. The odd- and even-frames only tilt series stacks were then reconstructed at 6.8 Å voxel size in IMOD and used as the input for denoising in cryo-CARE (Buchholz et al, 2019). The default parameters were used for cryo-CARE, with the exception of the "n_tiles" value in the prediction settings which was changed to 5. All cryotomogram slices shown in Figs. 2–6 were created from denoised data. A copy of these figures was also created using non-denoised data and can be found as Appendix Figs. S32–S36. Appendix Figs. S14 and S17 were also created from denoised data.

## Template matching

Candidate nucleosome subtomograms were picked from bin ×2 cryotomograms (6.8 Å pixel size; low-pass filtered using *mtffilter* with mean and sigma values of 0.17 and 0.05, respectively, with the roll-off starting at 40 Å resolution) by template matching using the PEET package (Heumann, 2023; Heumann et al, 2011; Nicastro et al, 2006). A featureless soft-edged cylinder ~10 nm diameter and ~5.5 nm tall was generated as the template-matching reference using the Bsoft program *beditimg* (Heymann, 2022). Cryotomograms of interest were seeded with a grid of points (from which translational searches initialize) that were spaced ~7 nm apart using the PEET program *gridInit*. For efficiency, we excluded seeded points that were located outside chromatin domains or metaphase chromosome regions by using the segmented version of the cryotomograms as masks. At the end of the PEET template matching run, duplicate removal was performed on the candidate points, with a distance threshold of ~5.5 nm.

Globular complexes such as ribosomes and preribosomes were picked using PEET, as described above with three differences: (1) A featureless sphere of ~24 nm diameter was used as the reference. (2) The initial grid of points seeded were ~21 nm apart. (3) Duplicate removal was performed with a 13.5 nm distance threshold.

The coordinates of the template matching subtomograms (candidate nucleosomes/ribosomes/preribosomes) were then exported to RELION for subtomogram averaging, classification, and refinement. The remaining "junk" subtomograms were excluded at the end of each classification run.

## Subtomogram analysis of mononucleosomes

Classification and subtomogram averaging were performed using RELION 3 (Kimanius et al, 2016; Zivanov et al, 2018). A summary of the workflow is shown in Appendix Figs. S12 and S26. $64 \times 64 \times 64$ voxel subtomograms corresponding to the template-matching coordinates in PEET were extracted in RELION and subjected to 3-D classification. To analyze for mononucleosomes, the subtomograms were classified using a featureless cylinder as the reference and a 110 Å-diameter spherical mask (Appendix Figs. S12A and S26).

To analyze for ordered stacked dinucleosomes, the same set of extracted subtomograms from the mononucleosome PEET run were subjected to 3-D classification, using a double-stacked featureless cylinder as the reference and a larger 240 Å-diameter spherical mask. The resulting nucleosome classes from the double-cylinder 3-D classification run were selected and subjected to a second round of 3-D classification, where an ordered stacked dinucleosome class was observed (Appendix Figs. S12 and S26B,C).

In an attempt to pick up more nucleosome particles, the original set of extracted subtomograms were again subjected to 3-D classification, this time using the ordered stacked dinucleosome average as the reference, low-pass filtered to 60 Å in RELION (Appendix Figs. S12D and S26D). Nucleosome averages from this classification run were manually placed into three groups: (1) mononucleosomes; (2) ordered stacked dinucleosomes; and (3) mononucleosomes with an additional proximal density. Subtomograms from each of these three groups were subjected to a second round of 3-D classification separately (Appendix Figs. S12E and S26E). For Group 1, the resulting mononucleosome classes were manually separated into two further groups corresponding to the presence of either short or long linker DNA densities and then subjected to 3-D refinement separately. 3-D classification for Group 2 resulted in several ordered stacked dinucleosome classes, which were refined as a single group. For Group 3, we detected several mononucleosome classes with an additional density at different positions around the DNA gyres. Each of these Group 3 classes were subjected to separate 3-D refinements (Appendix Figs. S13 and S27). The details are summarized in Appendix Tables S6 and S7. For the details corresponding to the control G1 cells in 9% glycerol or without cryoprotectant, see Appendix Table S8.

## Subtomogram analysis of other nucleosome and multi-nucleosome structures

To detect the presence of ordered side-by-side dinucleosomes and stacked tri-/tetra-nucleosomes, further 3-D classification was performed on the nucleosome particles in Groups 2 and 3 (Appendix Figs. S13, S27, panels B and C), respectively. Custom masks were generated to include the nucleosome densities seen in the subtomogram averages; and to enclose additional volume where we anticipate more nucleosomes may be found. Figures EV3 and EV5 show a rendering of the reference and mask combination used for each of the 3-D classification runs. These custom masks were created in UCSF Chimera (Pettersen et al, 2004) by manually positioning multiple featureless cylinders of 15 nm-diameter and 10 nm-height, and then combining them into a single volume using the following Chimera commands:

```
vop new empty size 64 gridSpacing 6.766
vop maximum [cyl #1] [cyl #2] [cyl #3] onGrid [empty #]
```

The multi-cylinder masks' edges were then smoothed by Gaussian filtering with the command:

```
vop gaussian [combined cylinder #] sDev 67.66
```

References used during these 3-D classification runs were subtomogram averages generated during the refinement step from either Group 2 or 3, all low-pass filtered to 60 Å in RELION. The angular search range was restricted to ±15° for these 3-D classification runs.

## Subtomogram analysis of ribosomes and preribosomes

Template matching hits for candidate ribosomes in the cytoplasm were subjected to direct 3-D classification using a 24-nm-diameter sphere as a reference. The unambiguous ribosome class averages were pooled

and then subjected to refinement. For preribosome subtomogram averaging, template matching hits from both the nuclear region and the cytoplasm was included during RELION classification. Candidate preribosomes were first 2-D classified, whereupon classes that did not have large complexes were removed. Next, 3-D classification was performed on the remaining particles, using the refined ribosome class average generated in the previous step as a reference. An initial low-pass filter of 60 Å was applied to the ribosome reference in RELION, during the 3-D classification run. See details in Appendix Table S9.

### Docking of nucleosome crystal structures

The docking of nucleosome crystal structures into our refined subtomogram averages was performed using UCSF Chimera. The nucleosome crystal structure (PDB 1AOI) (Luger et al, 1997) was docked into nucleosome subtomogram averages that have short linker DNAs. The chromatosome crystal structure (PDB 5NL0) (Garcia-Saez et al, 2018) was docked into subtomogram averages with longer linker DNAs. The histone tails were removed from the atomic models prior to docking into our subtomogram averages. Briefly, the corresponding atomic model was manually aligned into the subtomogram average, and then the fit-in-map function was executed. For the fit-in-map function, the option to simulate a density map from the atomic model at 20 Å was used.

For the stacked dinucleosome averages, the Chimera "hide dust" function was used to hide the nucleosome density with the shorter linker DNA. PDB 5NL0 was then fitted to the remaining visible nucleosome. The hidden nucleosome density was next rendered visible using the command "sop invertShown", and then docked by PDB 1AOI. For the mononucleosome subtomogram averages that have an additional proximal density, PDB 1AOI was fitted to the nucleosome region, with the proximal density still rendered.

The fitting correlation coefficients for the G1 nucleosomes subtomogram averages were as follows: mononucleosome, short linker = 0.78; mononucleosome, long linker = 0.78; stacked dinucleosome, nucleosome 1 = 0.71; stacked dinucleosome, nucleosome 2 = 0.67; nucleosome with proximal densities = 0.65–0.73. For the metaphase nucleosome subtomogram averages: mononucleosome, short linker = 0.77; mononucleosome, long linker = 0.77; stacked dinucleosome, nucleosome 1 = 0.68; stacked dinucleosome, nucleosome 2 = 0.66; nucleosome with proximal densities = 0.70–0.72.

### Simulation of ribosome densities

Coordinates for the single-particle reconstruction of human ribosomes (PDB 4UG0) (Khatter et al, 2015) were used as input for density map simulations. The EMAN2 program *e2pdb2mrc.py* was used to create an .mrc file using the following command:

```
e2pdb2mrc.py --apix 6.766 --box 64 --res [res]
--center 4UG0.pdb 4UG0.mrc
```

The option [res] corresponds to the resolution of the output map. Three maps were created, covering resolutions (25–35 Å) typical of averages of the order of ~1000 subtomograms, such as the one obtained from the G1 cell cryotomograms.

### Remapping

Remapping of nucleosome, ribosome, and preribosome particles was performed using a custom python script *ot_remap_v2.py*

(available at https://github.com/anaphaze/ot-tools). The coordinate and orientation information for each particle from the RELION refined .star files was imposed on the corresponding subtomogram average, which was then written into a new .mrc file. Remapped models were visualized using UCSF Chimera.

## Data availability

A copy of all the tilt series raw data and cryotomograms presented in this paper (Table S5) was deposited in EMPIAR (Iudin et al, 2023) as EMPIAR-12425. Subtomogram averages of the RPE-1 nucleosomes, dinucleosomes, nucleosomes with gyre-proximal densities (Figs. 3A–C and 5A–C), cytoplasmic ribosomes (Appendix Fig. S10B), and preribosomes (Fig. EV2B) have been deposited at EMDB. The EMD serial numbers are shown in Appendix Tables S6–S9. Raw confocal and immunoblot are available on the BioImage Archive as entry S-BIAD1466.

The source data of this paper are collected in the following database record: biostudies:S-SCDT-10_1038-S44318-025-00407-2.

## Peer review information

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

## Acknowledgements

We thank the CBIS microscopy staff for support and training. We thank Paulo D'Avino for advice on RPE-1 synchronization. JC, TL, and LG were supported by a Singapore Ministry of Education grant MOE2019-T2-1-140. LG and JC were supported in part by startup funds from the University of Virginia Strategic Investment Fund. WR and US were funded by the Biomedical Research Council of A*STAR (Agency for Science, Technology and Research), Singapore.

## Author contributions

**Jon Ken Chen**: Conceptualization; Investigation; Visualization; Writing—original draft; Writing—review and editing. **Tingsheng Liu**: Conceptualization; Investigation; Visualization; Writing—original draft; Writing—review and editing. **Shujun Cai**: Conceptualization; Investigation; Writing—original draft; Writing—review and editing. **Weimei Ruan**: Conceptualization; Investigation; Writing—original draft; Writing—review and editing. **Cai Tong Ng**: Conceptualization; Investigation; Writing—original draft; Writing—review and editing. **Jian Shi**: Supervision. **Uttam Surana**: Conceptualization; Writing—original draft; Writing—review and editing. **Lu Gan**: Conceptualization; Supervision; Funding acquisition; Visualization; Writing—original draft; Writing—review and editing.

Source data underlying figure panels in this paper may have individual authorship assigned. Where available, figure panel/source data authorship is listed in the following database record: biostudies:S-SCDT-10_1038-S44318-025-00407-2.

## Disclosure and competing interests statement

The authors declare no competing interests.

# Expanded View Figures

**Figure EV1. Metaphase chromosomes are depleted of elongating RNA polymerase II.**

Differential interference contrast (DIC) images of representative G1 and metaphase cells that were stained to detect DNA (stained with DAPI) and immunofluorescent detection of elongating RNAPII phosphorylated at serine 2 of the RPB1 subunit's C-terminal tail heptad repeats (RPB1-S2P). (**A**) G1 and metaphase cells were incubated in DMEM/F12 medium for 1 min before fixation. (**B**) G1 and metaphase cells were incubated in DMEM/F12 medium containing 9% DMSO for 1 min before fixation. (**C**) G1 and metaphase cells were incubated in DMEM/F12 medium containing 9% glycerol for 1 min before fixation. In the middle two columns, the DAPI and immunofluorescence signals are shown in inverted contrast for clarity.

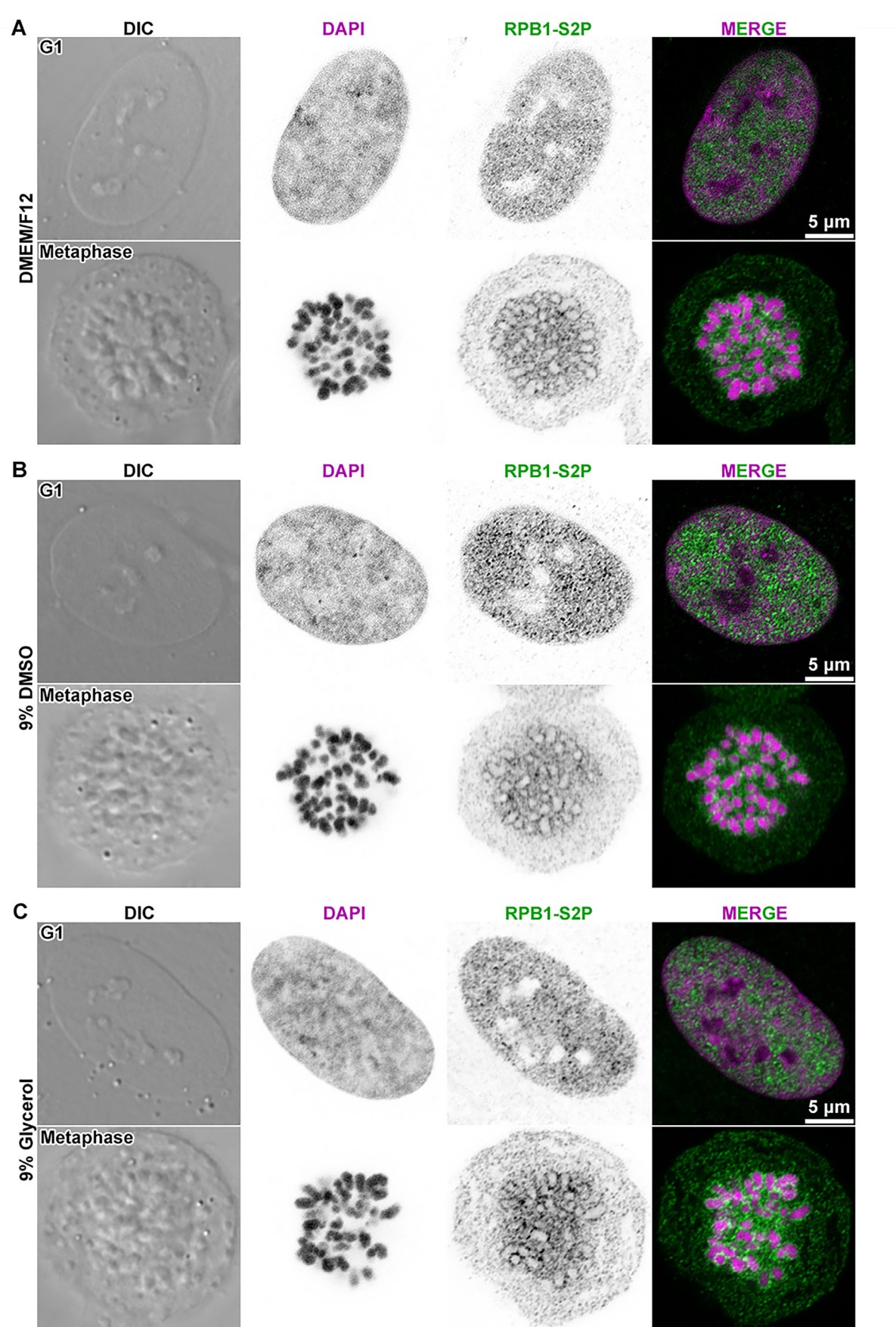

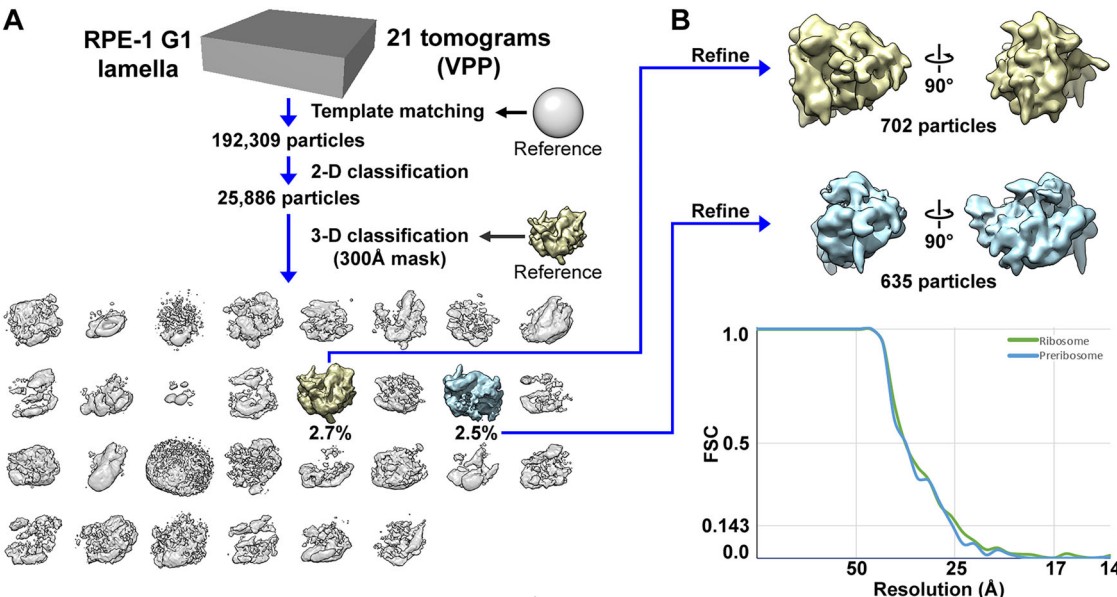

**Figure EV2. Subtomogram analysis of preribosomes.**

(A) Template matching for preribosomes was performed on all cryotomograms containing nuclear regions, using a spherical reference. The candidate hits were then subjected to 2-D classification; classes that contain subtomograms that do not correspond to large complexes were removed. The remaining subtomograms were then subjected to 3-D classification, using the ribosome refined class average shown in Appendix Fig. S10B as the reference, but low-pass filtered to 60 Å resolution. (B) The preribosome (blue) class average was refined to ~32 Å resolution, based on the FSC = 0.5 criterion. The 80S ribosome average (yellow) is also shown for comparison purposes. The preribosome resembles the 60S subunit of the mature ribosome and is oriented to better show their similarities.

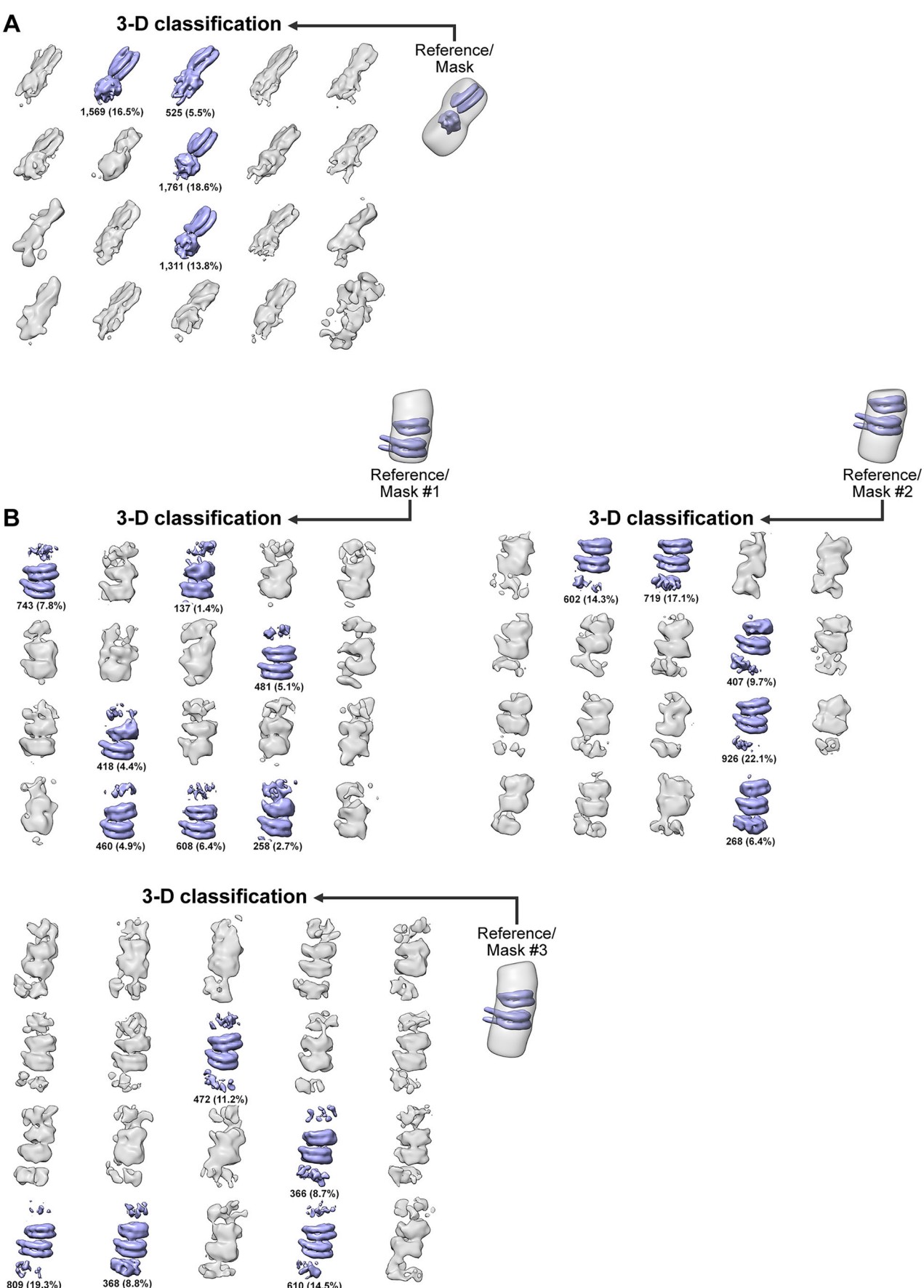

**Figure EV3.  Subtomogram analysis of alternative ordered nucleosome packing motifs in G1 cells.**

Nucleosome particles from Groups 2 and 3 (Appendix Fig. S13B, C, respectively) were subjected to an additional round of 3-D classification, using custom masks that enclose volumes where an additional complex may reside. Since the particles were already aligned from the previous refinement step, a restricted angular search range was imposed for these runs. The "reference/mask" models in the figure depicts the location of the volume masked-in (gray), relative to the reference (blue) used for each classification run. The masks used for these 3-D classification runs were optimized for (**A**) side-by-side nucleosomes and (**B**) ordered trinucleosomes and tetranucleosomes. Class averages that contain at least one ordered nucleosome are shaded blue.

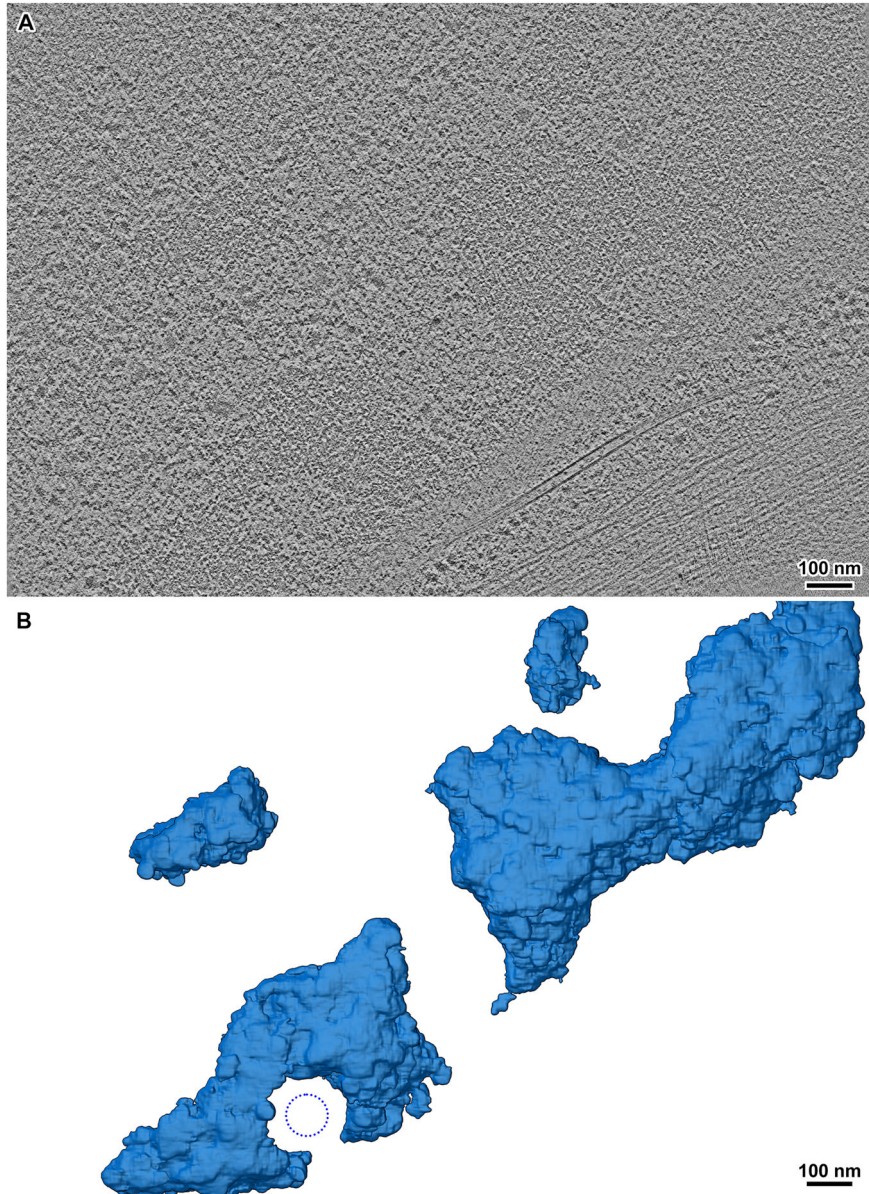

**Figure EV4. G1 chromatin domains are irregular.**

(**A**) Cryotomographic slice a region near the nuclear envelope. (**B**) Convolutional neural network (CNN) based segmentation of the region in (**A**). For clarity, the CNN segmentation rendering shows the entire ~90 nm thickness of the lamella whereas the cryotomographic slice shows the central 10 nm. Because of this difference, the CNN segmentation shows more chromatin than is visible in the cryotomographic slice. The dotted blue circle indicates the approximate position of the nuclear pore complex (not segmented).

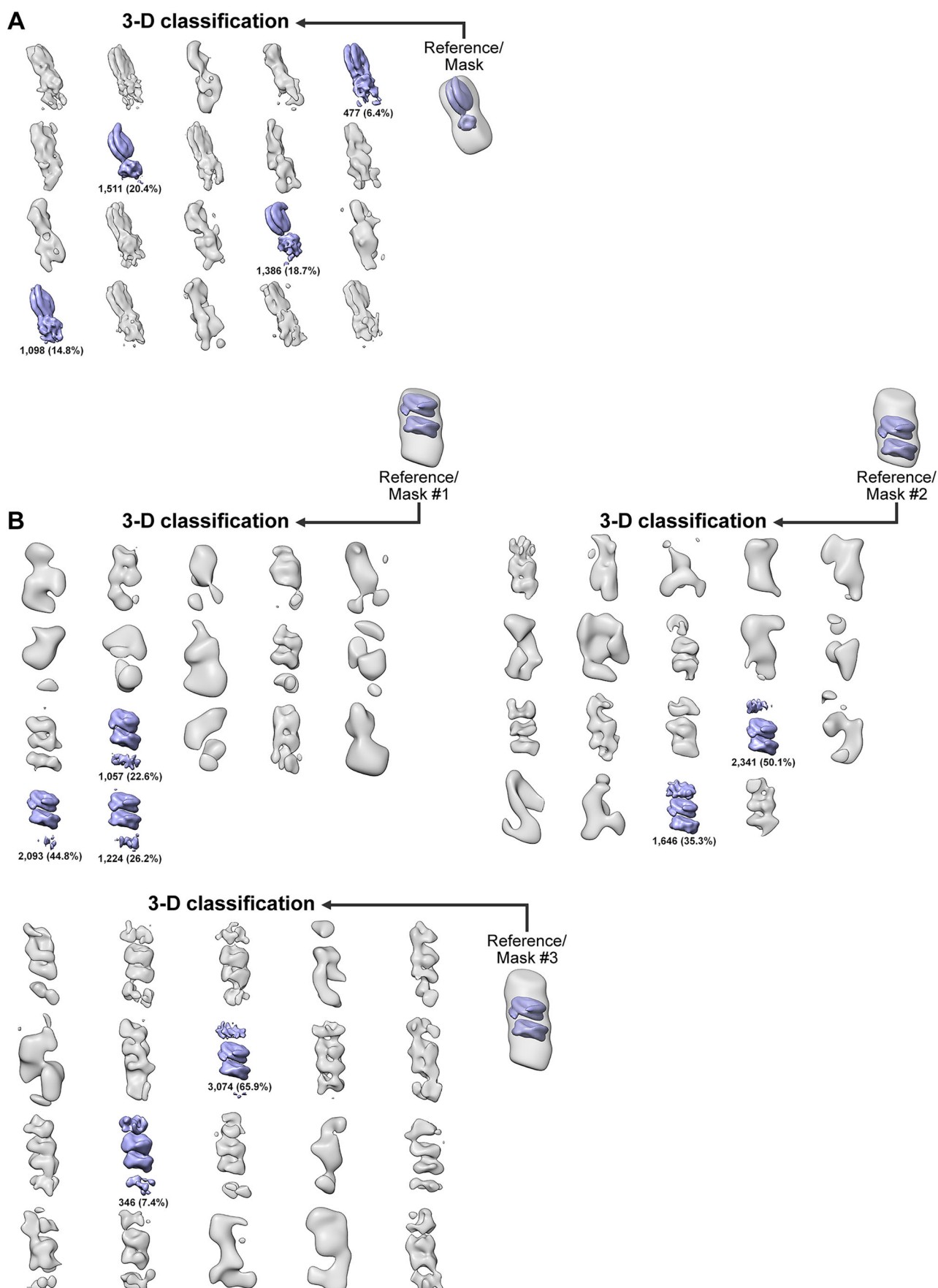

◀ **Figure EV5. Subtomogram analysis of alternative ordered nucleosome packing motifs in metaphase cells.**

Similar to Fig. EV3, nucleosome particles from Groups 2 and 3 (Appendix Fig. S27B, C, respectively) were subjected to an additional round of 3-D classification, using custom masks that enclose volumes where an additional complex may reside. Since the particles were already aligned from the previous refinement step, a restricted angular search range was imposed for these runs. The "reference/mask" models in the figure depicts the location of the volume masked-in (gray), relative to the reference (blue) used for each classification run. The masks used for these 3-D classification runs were optimized for (**A**) side-by-side nucleosomes and (**B**) ordered trinucleosomes. Class averages that contain at least one ordered nucleosome are shaded blue.

