## [Peer Review File · The EMBO Journal]

Nanoscale analysis of human G1 and metaphase chromatin in situ

Jon Ken Chen, Tingsheng Liu, Shujun Cai, Weimei Ruan, Cai Ng, Jian Shi, Uttam Surana, and Lu Gan

Corresponding author(s): Lu Gan (lu@anaphase.org)

Review Timeline:

Submission Date:	26th May 24
Editorial Decision:	5th Aug 24
Revision Received:	27th Nov 24
Editorial Decision:	13th Jan 25
Revision Received:	11th Feb 25
Accepted:	21st Feb 25

Editor: Cornelius Schneider

Transaction Report:

Please note that the manuscript was transferred from another journal where it was originally reviewed.

To help keep track of the large number of revised figures and new figures, please see the list below. Also note that **in the manuscript files, revised text and figures have been highlighted yellow.**

REVISED FIGURES:

- Figure 1. Metaphase chromosomes are depleted of RNA polymerase II – Immunofluorescence revised to include treatment with 9% DMSO or 9% glycerol.
- Figure 3. Subtomogram analysis of G1 nucleosomes – Classification revised to show numbers of each particle class.
- Figure 5. Subtomogram analysis of metaphase nucleosomes – Classification revised to show numbers of each particle class.
- Figure S14. Classification flowchart of G1 chromatin domains – Classification revised to show numbers of each particle class.
- Figure S15. Refinement of G1 mononucleosomes and dinucleosomes – Classification revised to show numbers of each particle class.
- Figure S16. Subtomogram analysis of alternative ordered nucleosome packing motifs in G1 cells – Classification revised to show numbers of each particle class.
- Figure S30. Classification flowchart of M cells – Classification revised to show numbers of each particle class.
- Figure S31. Refinement of metaphase mononucleosomes and dinucleosomes – Classification revised to show numbers of each particle class.
- Figure S32. Subtomogram analysis of alternative ordered nucleosome packing motifs in metaphase cells – Classification revised to show numbers of each particle class.

NEW FIGURES:

- Figure S8. Metaphase chromosomes are depleted of RNA polymerase II.
- Figure S9. Cryoprotection treatments do not induce apoptosis markers.
- Figure S17. Cryo-ET of G1 chromatin in situ of cells with glycerol cryoprotection.
- Figure S18. Classification flowchart of G1 chromatin domains in 9% glycerol-cryoprotectant cells.
- Figure S19. Refinement of mononucleosomes from glycerol-cryoprotected G1 cells.
- Figure S20. Cryo-ET of G1 chromatin in situ of cells without cryoprotection.
- Figure S1. Classification flowchart of G1 chromatin domains for cells without cryoprotection.
- Figure S22. Refinement of Group 1 nucleosome class averages from G1 cells without cryoprotection.
- Figure S25. Remapped models of G1 nucleosome groups.
- Figure S26. Remapped models of G1 nucleosome individual classes.
- Figure S33. Remapped models of metaphase nucleosome groups.
- Figure S34. Remapped models of metaphase nucleosome individual classes.

Reviewer: 1

Eukaryotic chromatin undergoes dynamic changes in its structure during cellular differentiation and in each stage of the cell cycle. The specific local chromatin structures in each type of chromatin region are thought to play an instrumental role in exerting their specific functions. However, it remains unclear what local chromatin structures are formed in these regions and how these chromatin structures control the events on chromatin. The author's group is one of the pioneers in the chromatin biology field in investigating chromatin structure *in situ*. In this study, the authors reported their updated cryo-ET method for analyzing the cellular nucleosome structures and determined nucleosome structures formed in interphase (G1 phase) and metaphase RPE-1 cells (hTERT-immortalized human retinal pigment epithelial cells) *in situ*.

The authors optimized the cryoprotectant solution and determined that a buffer containing 9% DMSO is suitable for vitrifying specimens uniformly, preserving nucleosome structures without damage. Subsequently, they employed this cryoprotectant solution for cryo-focused ion beam (cryo-FIB) milling of G1 phase and metaphase cells, enabling the determination of subtomogram averaged structures for mono-nucleosomes and stacked di-nucleosomes. While mono-nucleosomes from the G1 phase displayed variable linker DNA lengths, including a class with very short linker DNA, metaphase nucleosomes exhibited uniform structures with closed and extended linker DNA. Regarding di-nucleosomes, both G1 phase and metaphase chromatin contained "Type II" di-nucleosomes formed through interactions between the H4 N-terminal tail and the H2A-H2B acidic patch. Furthermore, the study elucidated the structures of mono-nucleosomes with additional densities.

Overall, this study constitutes pioneering work that provides fundamental knowledge for chromatin biology. The manuscript is well-written and effectively highlights its novel findings through references to prior research. The analysis pipeline is designed well, and the results clearly support the principal conclusion of the manuscript, which is the prevalence of canonical octameric nucleosomes in interphase and metaphase RPE1 chromatin. However, it is unclear if the result reflects the entire picture of the cellular chromatin structures. Additionally, while the manuscript has substantial potential to address numerous unresolved questions in chromatin biology, its primary focus lies in reporting the observed nucleosome structures, with limited exploration of the biological significance of these structures. To address these issues, the following points should be considered:

Thank you for the detailed, constructive comments. We agree with most of the critiques and have done more than 4 new experiments, made 12 new figures, revised 9 existing figures and multiple text sections. We believe our new controls address both Reviewer 1 and Reviewer 2's concerns about the effects of DMSO, namely that the amount of ordered stacked dinucleosomes and nucleosomes with gyre-proximal densities may be affected by buffer and contrast, but the overall conclusions don't change. Regarding the exploration of biological significance, both Reviewers 1 and 2 have made suggestions that would require us to overinterpret our data. Given that *in situ* cryo-ET is a nascent field, we would rather interpret our data much more cautiously (we explain in detail

below). We prefer a more conservative interpretation than to make measurements and claims that would have increased the novelty or earth shattering-ness of our work. Perhaps in the future, when *in situ* cryo-ET is more mature, we and others will be able to revisit this work (all data will be deposited to EMPIAR) and more confidently draw new conclusions, which may be more novel and earth-shattering.

Major points

1. Effect of 9% DMSO on chromatin structure and nucleosome interaction:

The use of cryoprotectants for preparing cryo-FIB-milled cell lamellas holds significant potential for enabling reproducible cellular chromatin structural analyses and can be applied to various cellular targets. However, given that DMSO is known to potentially weaken protein folding and inter-protein interactions (PMID: 28803470), it is essential to thoroughly assess the effects of the 9% DMSO used in this study. While the cryo-ET of extracted HeLa oligonucleosomes with 9% DMSO (as shown in Fig. S6 and S7) suggests that nucleosome structure remains intact in the cryoprotectant buffer, I recommend investigating other interactions, such as nucleosome-nucleosome and nucleosome-chromatin protein interactions. Additionally, while the authors examined the effect of cryoprotectants using G1 cells, the images presented in Fig. S3 are too small to evaluate the effects of 9% DMSO on cellular chromatin. To assess the impact of 9% DMSO on nucleosome-nucleosome and nucleosome-chromatin protein interactions in cellular environments, it is crucial to rigorously investigate chromosome volume and the localization of typical chromatin binding proteins (e.g., linker histone H1, HP1, transcription factors) in both G1 and metaphase cells. Assessing the effect on linker histone H1 binding is particularly important since the cryo-ET structures of nucleosomes in this manuscript (Fig. 3A & 5A) lack densities corresponding to H1, the most abundant chromatin protein except for core histones, despite the presence of additional densities for other chromatin-bound proteins (Fig. 3C & 5C).

We are unable to rigorously assess the effects of linker histone H1 binding by cryo-ET because we lack the resolution to identify H1 (~6 Å resolution is needed to unambiguously identify the H1 fold) and correlative cryo light and electron microscopy lacks the resolution as well. We have instead used four approaches available to us to study the potentially detrimental effects of DMSO:

1) Immunofluorescence of cells treated with 9% DMSO and 9% Glycerol. These experiments are now presented as Figures 1 (RPB1-S5P) and S8 (RPB1-S2P) and discussed in lines 233 – 237:

“DMSO – and also glycerol – may also have unexpected effects on cellular health. Immunofluorescence of G1 and metaphase cells treated with 9% DMSO or 9% glycerol show a similar CTD-S5P (Figure 1, B and C) and CTD-S2P (Figure S8) distributions compared to untreated cells. These confocal microscopy data suggest that large-scale chromatin structure is not disrupted by these cryoprotectant treatments.”

2) Immunoblots to show that there is no induction of apoptosis, which was raised by Reviewer 2. These immunoblots are presented as Figure S9 and discussed in lines 237 – 248:

“To control for the induction of apoptosis, we performed immunoblot analysis for the apoptotic marker, cleaved poly (ADP-ribose) polymerase-1 (PARP-1) {Kaufmann, 1989; Kaufmann, 1993}. Untreated negative control G1 and metaphase cells did not show cleaved PARP-1 while positive-control cells treated with the apoptosis inducer staurosporine {Bertrand, 1994} showed cleaved PARP-1 (Figure S9). Neither DMSO nor glycerol treatment induced cleaved PARP-1 (Figure S9), indicating that brief cryoprotectant treatments with these cryoprotectants did not induce apoptosis. In summary, brief immersion in medium containing 9% DMSO both cryoprotects plunge-frozen cells without killing them and preserves nucleosome structure. In the cryo-ET experiments described below, most samples were cryoprotected in medium with 9% DMSO, with additional control samples prepared in medium with either no cryoprotectant or 9% glycerol.”

3) Cryo-ET of G1 cells using the alternative cryoprotectant glycerol, also at 9% concentration. These cryotomograms of cells in 9% glycerol are presented as Figures S17 – S19 and discussed in lines 413 – 427:

“To control for the effects of DMSO treatment in situ, we also performed cryo-ET on G1 cells that were cryoprotected in 9% glycerol (Figure S17, A and B). The nucleosomes in these cells were also enriched in the domains (Figure S17, C – E), but the DNA gyres were less-defined than in the DMSO-treated cells. Stacked nucleosomes (Figure S17, C – E) and dense irregular bodies were also found in the nucleoplasm (Figure S17, F). By visual inspection, we could not find any notable differences between the glycerol- and DMSO-cryoprotectant nuclei cryotomogram densities.

To determine if glycerol and DMSO treatment lead to differences in chromatin structure that is too fine to directly visualize, we performed 3-D classification and averaging analysis. Template-matching and 3-D classification analysis using the same strategy as for DMSO-treated cells produced class averages of mononucleosomes and nucleosomes with gyre-proximal densities (Figure S18). No ordered stacked dinucleosome classes were found. Group 1’s class averages refined to 29–34 Å resolution while group 3’s class averages refined to 31 Å resolution (Figure S19).”

4) Cryo-ET of G1 cells without any cryoprotectant. These cryotomograms of cells in the absence of cryoprotectant are presented as Figures S20 – S22 and discussed in lines 429 – 451:

“We also performed Volta cryo-ET analysis of G1 cells that were not treated with any cryoprotectants. Note that these cells were damaged by crystalline ice, whose presence could be detected in the tilt-series diffraction-contrast features exemplified in our DMSO concentration screen (Figure S4). Volta cryotomographic slices of perinuclear regions had greater contrast (Figure S20, A and B) than similar regions imaged in either DMSO- or glycerol-cryoprotected cells. Enlargements show that the density difference between

nucleosomes and nucleoplasm is more pronounced (Figure S20, C – E) than in cells cryoprotected with DMSO or glycerol. The nucleoplasm between the macromolecular complexes appears emptier than in both the DMSO- and glycerol-cryoprotected cells. This empty-nucleoplasm phenotype, like the string-like chromatin phenotype of the cryosections, may have resulted from the aggregation artifact that occurs in the absence of vitreous freezing {Kellenberger, 1987; Dubochet, 2001}.

To characterize the changes at higher resolution, we did classification analysis using the workflow described above (Figure S21, A – D), but only detected convincing Group 1 (mononucleosome) class averages. We did not detect any convincing group 2 (ordered stacked dinucleosome) and group 3 (nucleosomes with gyre-proximal densities) class averages. Group 1's class averages refined to 29 – 56 Å resolution (Figure S22); note that two of the class averages (2 and 9) have only a few hundred subtomograms and are therefore at much-lower resolution. Given the lower contrast cryotomograms of the glycerol-cryoprotected cells and the ice-crystal artifacts in the non-cryoprotected cells, the analysis below reports only on the DMSO-treated cells, with caveats where appropriate.”

We summarize our current understanding of cryoprotectant effects in lines 705 – 718:

“Our observation that ordered stacked dinucleosomes were not detected in glycerol-treated cells could result either from insufficient signal-to-noise ratio due to the glycerol or buffer-induced differences in abundance; further investigation will be needed to distinguish these possibilities. The non-cryoprotected G1 cells did not have an ordered stacked dinucleosome class, but did have a class average that contained two nucleosome-like densities that vaguely resembles a stacking interaction. It is unclear to what extent ice-crystal artifacts contribute to the absence of ordered stacked dinucleosomes in these samples. Even though unambiguous ordered stacked nucleosomes are not present in all three cryoprotection conditions tested, it is clear that stacking is abundant, as can be commonly seen in tomographic slices. Furthermore, the other nucleosome-nucleosome interaction (gyre-proximal) can be seen in a subset of remapped models. These experiments suggest that the detection of some ordered nucleosome-nucleosome assemblies may depend on the biochemical environment, quality of freezing, and image contrast, and will require further investigation.”

We also updated Table S2 to acknowledge that ordered stacked dinucleosomes and nucleosomes with gyre-proximal densities may be sensitive to cryoprotectants or image contrast.

2. The linker DNA angle and Linker DNA length in reconstructed 3D tomographic maps

The subtomogram averaging of G1 phase and metaphase nucleosomes resulted in nucleosome structures with closed linker DNA (Fig. 3A & 5A), suggesting that linker DNA angles tend to be closed in cellular environments. However, subtomogram averaging pipelines are often biased

toward stable structures. Indeed, many picked nucleosome- like particles were removed during the process (Fig S11, S12, S20). Therefore, it remains uncertain whether this result fully represents the diversity of cellular nucleosome structures. This issue can be addressed by directly measuring the linker DNA length and angle within the 3D tomographic map of chromatin, as has been demonstrated in previous studies by Beel et al (PMID: 34520722), Zhang et al (PMID: 35907400), and Zhang et al (PMID: 37217501). This approach may also help identify whether the various linker DNA lengths observed in G1 phase nucleosome classes are due to actual variations in linker DNA length or differences in linker DNA structural stability in G1 phase chromatin.

Presumably the “many picked nucleosome- like particles were removed during the process (Fig S13, S14, S30; formerly Fig S11, S12, S20)” refers to the gray densities since the blue densities were kept for further analysis. The gray densities are not nucleosome- like. Instead, they are averages of either non-nucleosome or non-canonical-nucleosome densities. The non-nucleosome nature of these gray densities can most easily be seen by comparison to the natural negative control: the subtomogram analysis of the cytoplasm from G1 cells (Figure S13B; formerly Fig S11B), which shows ambiguous densities that are also seen in the nucleus. Our subtomogram workflow uses a featureless cylinder as a template and a low cross-correlation cutoff, which reduces model bias but results in large numbers of false positives, which may result in readers thinking that we have rejected large numbers of nucleosome- like particles. To make it clearer that the excluded densities are not nucleosome- like, we have added the following text to the legends of Figures S13, S14, and S30:

“The ambiguous densities (gray) are not canonical nucleosomes; they are abundant because the template matching process uses a featureless cylinder reference and a low cross-correlation cutoff. As a result, large numbers of false positives are included in the classification analysis.”

We are unable to analyze linker DNA length using the method of Beel et al (PMID: 34520722), Zhang et al (PMID: 35907400), and Zhang et al (PMID: 37217501). Beel et al (PMID: 34520722) used purified chromosomes and Zhang et al (PMID: 35907400) used reconstitute tetranucleosomes. The densities in these studies are unambiguous because they lack other nuclear constituents that are present in our *in situ* data. Zhang et al (PMID: 37217501) denoised their data using simulated training data as ground truth. Their denoised nucleosome subtomograms were only used to study the variability in linker- DNA conformation – no other forms of variability was presented, which shows the limitation of using simulated data for ground truth. While denoising increases the visual appeal of subtomograms, it is unclear what false negatives and false positives may arise because there are no methods to validate the *in situ* ground truths. To explain these limitations, we have added the following text:

“Nucleosome linker DNA orientation and length variability may both contribute to the nucleosome class averages that have short linker- DNA densities. Previous work on

isolated chromosomes and oligonucleosomes *in vitro* produced tomograms of unambiguous nucleosomes {Beel, 2021 #145; Zhang, 2022 #170; Zhang, 2023 #266}. While it is possible to directly visualize DNA in purified chromatin, which contains nucleosome and DNA densities, such a task is not feasible *in situ*, which contains many non-chromatin densities. Denoising may facilitate the visualization of chromatin densities, but may introduce errors if the training data is biased, for example with simulated densities. Further improvements in image quality, sample preparation, and denoising algorithms may facilitate the annotation and measurement of DNA in unaveraged subtomograms *in situ*.”

3. Correlation of chromatin compaction levels and linker DNA structures

The manuscript focuses on reporting the observed nucleosome structures with their updated cryo-ET method, and the current manuscript may suit more in a specific journal. However, I believe this manuscript has the potential to address numerous unresolved questions in the field of chromatin biology with minor modifications. For instance, classical textbook models of chromatin suggest that nucleosomes with open and closed linker DNA induce less and more tightly compacted chromatin fibers, respectively. Simulation studies also support the idea that linker DNA angle and length play a pivotal role in controlling chromatin compaction (PMID: 29628212). The subtomogram averaging of G1 phase chromatin reveals nucleosomes with varying linker DNA lengths (Fig. 3A), implying the presence of nucleosomes with flexible (or shorter) linker DNA in the G1 phase. By re-mapping each class and assessing the correlation between nucleosome density and linker DNA angle, the authors could potentially provide crucial insights into how chromatin compaction is regulated.

We had indeed remapped the nucleosome class averages, which showed us that the distribution of each class was random. However, we did not show these results in the original manuscript. We now include the remapped nucleosomes (of each group and of each individual class average) for representative G1 and metaphase cell cryotomograms as Figures S25, S25, S33, and S34. The fact that nucleosomes with each length of ordered (or longer) linker DNA have random distributions are now reported in lines 469 – 475:

“To determine if nucleosomes belonging to each group had a biased distribution, we remapped the mononucleosomes, ordered stacked dinucleosomes, or nucleosomes with gyre-proximal densities into three separate maps (Figure S25), but did not find any bias or clustering of any of these classes. Furthermore, separate remaps of each class (7 for mononucleosomes, 1 for stacked dinucleosomes, 5 for nucleosomes with gyre-proximal densities) did not reveal any biased distributions (Figure S26).”

and lines 578 – 583:

“As with the G1 cryotomograms, we also remapped the mononucleosomes, ordered stacked dinucleosomes, and nucleosomes with gyre-proximal densities as three separate groups (Figure S33), but did not find any distribution bias. Likewise, we remapped each class (8 for mononucleosomes, 1 for ordered stacked dinucleosomes, and 4 for nucleosomes with gyre-proximal densities) and did not find biased distributions of these class averages either (Figure S34).”

Minor points

1. [Fig. 1] The current Figure 1 does not align with the primary focus of this manuscript and could be misleading, as the cells used in Fig 1 were not treated with the cryoprotectant (9% DMSO) and were subjected to different conditions than those used in the cryo-ET experiments.

The new control immunofluorescence experiments that show the G1 cells treated with 9% glycerol or 9% DMSO were added to Figure 1 and the new Figure S8. The revised Figure 1 now aligns with the cryo-ET experiments.

2. [Fig 3 and 5] The populations of each 3D class should be provided.

We now report the populations of each 3D class in the revised Figures 3 and 5.

3. [Page 11, line 20] Please provide data supporting the claim, “We found modest differences in the class-average populations: compared to G1 cells, metaphase cells have ~23% more stacked dinucleosomes and ~24% fewer nucleosomes with gyre-proximal densities relative to mononucleosomes.”

The data were summarized in Table 1 and were calculated from the raw values in Tables S6 and S7. These are now reported at the end of the quoted sentence in line 619:

“(Table 1; raw values are in Tables S6 and S7)”

4. [Page 15, line 11] The claim “the combined cryo-ET and fluorescence microscopy results confirm that eukaryotic transcription is inversely correlated with macromolecular crowding” lacks clear support from the data in this manuscript. Mitotic transcription occurs in specific gene regions, which were not targeted during cryo-FIB-milling. Also, since the effect of 9% DMSO on nucleosome-nucleosome interactions is unclear, it may not be appropriate to compare the previous results without the use of cryoprotectants with those of this study.

Agreed – this text is inaccurate. It could also be misinterpreted to mean that all mitotic transcription is repressed, which is incorrect. We have revised the text to tone down the strength of our contribution and to acknowledge the complexity of mitotic transcriptional repression, in the lines 877 – 879:

“High-resolution correlative light microscopy and cryo-ET analysis will be needed to relate the 3-D structure of the nucleosomes to the transcriptional state.”

As for the effects of DMSO, we have now done the control experiment (immunofluorescence of transcriptionally active RNAPII markers; see above) and added this to the revised sentences above.

5. [Page 15, line 37] While the authors claimed that the dense irregular bodies are too big to be preribosomes and are unlikely to harbor canonical nucleosomes because they lack the characteristic double-gyre motifs, Zhang et al (PMID: 35907400) revealed that the LLPS droplet of tetra-nucleosome forms the spherical condensates, in which double-gyre motifs of the nucleosome are not observed. The authors should consider the possibility that dense irregular bodies are caused by the phase separation of the chromatin-related complexes, such as nucleosome, HP1, and PRC1.

The absence of double-gyre motifs in Zhang et al (PMID: 35907400) is probably from issues with the samples quality, data quality, or JPEG/Image conversion issues because even the uncondensed chromatin in their data lack the double-gyre motif. In fact, their reconstructions should have much higher signal-to-noise ratio than ours because their samples are reconstituted tetranucleosomes in non-physiological buffer (non-physiological because there are no solutes, small proteins, and other molecules only found *in situ* that lower image contrast). The potential pathologies of their data are also evident in the subtomogram average (screenshot below), which shows jagged DNA features near the entry/exit site. So we do not find that Zhang et al (PMID: 35907400) informs on the appearance of chromatin condensates *in situ*. Furthermore, to claim that the dense irregular bodies consist of specific components, we would have to use super-resolution cryo-correlative light and electron microscopy with X, Y, and Z resolution on the order of 10 to 40 nm, which is not available. We believe the identities of these interesting structures should be explored in future studies, once super-resolution correlative methods are both available and reproducible.

6. [Materials and Methods] The sentence, "For visualization purposes, cryotomographic slices shown in Figs. 2, 3, 4, and 5 were denoised using cryo-CARE (161)," appears redundant with the following section titled "Denoising for Visualization."

Thank you for catching this redundancy. The first sentence has been removed.

7. [Fig S12, S13, S14, S20, S21, S22] The populations of each 3D class should be provided.

Thanks for noticing this omission. We have added the populations of each 3-D class average to the figures (the numbers of subtomograms per class and the percentage of each class within a nucleosome group). The revised manuscript corresponds to Figures 3, 5, S14, S15, S16, S30, S31, and S32.

Reviewer: 2

The authors use Cryo-electron tomography and class averaging to analyze the G1 and metaphase chromatin in RPE1 cells. They detect canonical mononucleosomes with asymmetric and variable linker DNA lengths, particularly in G1, as well as some stacked dinucleosomes. No higher order oligomeric nucleosome assemblies are detected either in G1 or in metaphase. Nucleosomes form heterogeneous domains as also seen in light microscopy studies. The authors can also detect additional densities close to nucleosomes although the nature of these additional densities is unclear. Finally, they detect "megacomplexes" as well as ribosomes.

While the ability to detect mononucleosomes in situ is impressive, this is not new, the authors as well as others have done this in other cell types including HeLa cells. The findings do not seem to offer earthshattering new insights about chromatin structure beyond what is already known from super-resolution light microscopy, ChromEMT and other Cryo-ET studies. The significance of doing this analysis in RPE1 cells is not clear. Beyond this lack of a conceptual advance, I also have the following major technical concerns:

None of the super-resolution LM, ChromEMT, or other cryo-ET studies have revealed the packing of ordered stacked dinucleosomes, the diversity of linker-DNA lengths, nor the presence of gyre-proximal densities. We understand that not all readers will believe these to be earth shattering. On the matter of differences with these previous studies, aside from resolution of nucleosomes (not seen in super-resolution LM), our study has also shown that two of the ChromEMT-visible phenotypes (24-nm-thick fibers and nucleosome-free reticular patterns) are not present in RPE-1 cells, along with explanations for the technical basis of these differences.

1. The authors use cryoprotection in 9% DMSO as this provides higher contrast. They justify that this does not disrupt nucleosome structure by testing it on in vitro reconstituted chromatin fibers. I do not find this experiment to be a sufficient control. While 9% DMSO may not impact in vitro nucleosome structure, the authors provide no evidence that this is not leading to a stress or apoptotic response or activation of some pathway in the cell context that can disrupt chromatin structure.

Thank you for voicing this concern about the potential impact of DMSO, which was also raised by Reviewer 1. We have performed 4 new experiments to test for potential adverse effects of DMSO treatment. Please see our detailed response to Reviewer 1's main point 1 above. Specifically, treatment with 9% DMSO for short periods did not activate the apoptotic response. Regarding the effect on chromatin structure, we collected new data on G1 cells that were either cryoprotected with 9% glycerol or without any cryoprotectants. The glycerol-cryoprotected cells had similar distributions of nucleosomes, but lacked the ordered stacked dinucleosomes and nucleosomes with gyre-proximal densities. This result does not necessarily mean that glycerol disrupted a higher-order interaction because the lower contrast may have prevented our classification from detecting these ordered multi-nucleosome assemblies. The non-

cryoprotected cells had ice-crystals and an empty-appearing nucleoplasm. While non-cryoprotected cells also lacked the ordered stacked dinucleosome and a subset of nucleosomes with gyre-proximal densities, these have clearly suffered macromolecular rearrangements as indicated by the crystalline-ice artifacts just described.

2. The raw images look very grainy and low contrast to the eyes of this reviewer. The authors do several steps of template matching, remapping etc... to extract nucleosome structure. It is not clear to this reviewer if these many steps of refinement can introduce artefacts or miss structures that exist.

The raw images were recorded with a dose of ~2 electrons per square Ångstrom ($e^{-}/\text{Å}^2$). This is a much lower dose than what we believe Reviewer 2 is making a comparison to: single-particle cryo-EM images. For single-particle cryo-EM, the typical dose used nowadays is approximately two orders of magnitude higher (20 – 100 $e^{-}/\text{Å}^2$). However, cryotomograms are reconstructed from ~60 images, meaning that the cumulative dose is ~100 to 120 $e^{-}/\text{Å}^2$.

The reviewer is concerned about our use of image processing to extract (canonical) nucleosomes. Presumably the alternate approach is to manually annotate tomographic slices, as Reviewer 1 suggested we attempt to characterize linker-DNA heterogeneity. The manual annotation is feasible when the sample has undergone purification, i.e., when the cryo-ET densities arise from DNA and nucleosomes. *In situ* cryo-ET densities include so many non-chromatin densities that drawing conclusions from manual annotations would risk the inclusion of non-chromatin features in the analysis. As for the introduction of artifacts due to image processing, our workflow would have missed structures that we do not recognize, for example non-canonical nucleosomes.

3. Along similar lines, simply because the authors do not detect higher order oligomeric structures is not proof that such structures do not exist. They maybe missed by the analysis pipeline employed or the contrast may not be sufficient to capture them. While it is reasonable to conclude that nucleosome oligomers are heterogenous beyond the stacked dinucleosome given what we have learned over the years about chromatin structure from many other studies, this conclusion is neither novel nor directly and explicitly supported by the data in this study.

Yes, it is not possible to “prove” the absence of something. We have already acknowledged that the analysis pipeline focuses on ordered assemblies, which we explicitly define for the stacked dinucleosome (lines 323 – 325): “Here, the term “ordered” implies that there are sufficient numbers of canonical nucleosome pairs that are oriented and positioned with very low heterogeneity, resulting in a class average with two unambiguous canonical nucleosome densities.”

This definition allows for the existence of structures beyond stacked dinucleosomes, which we indeed observed (lines 696 – 699): “Furthermore, while stacks with three nucleosomes are seen in cryotomographic slices (Figure 4) and stacks with up to four

non-ordered nucleosomes were seen in remapped models (Figures 3D, 5D, 6), such arrangements are rare compared to ordered stacked dinucleosomes.”

Regarding the presence and “structure” of oligomers beyond stacked dinucleosomes, we do not see any conflict between what the Reviewer has written and what we have reported. Given this absence of conflict, it would seem that Reviewer 2 would consider the detection of, e.g., ordered stacked tri, tetra, pentanucleosomes etc., to be a “novel conclusion”. Unfortunately, such structures are too rare to detect as a class average in our dataset, so we are sorry that our conclusion is not novel enough.

Reviewer: 3

This manuscript describes cryo-ET analysis of in situ chromatin in both the G1 and metaphase stages of the mammalian cell cycle. It uses state-of-the-art sample preparation methods, imaging, and data processing to reveal nucleosome structure and orientation within the cryo-preserved nuclear environment. While the study describes, overall, modest effects of the chemically induced cell cycle stages, it does reveal a trove of details about chromatin structure/architecture in situ that are significant. There are also many technical aspects that will be of interest to the whole field of cellular tomography. Barring fundamental issues from other reviewers, I would suggest publishing mostly as is.

We thank the Reviewer for their positive assessment of our study.

Issue to address:

Figure 6 panel A seems to be missing a label for “Group 3”.

Thank you, we have added the missing “Group 3” label to Figure 6A.

Dear Dr. Gan,

Thank you for submitting your manuscript for consideration by the EMBO Journal. It has now been seen by two referees whose comments are shown below.

Given the referees' positive recommendations, I would like to invite you to submit a revised version of the manuscript, addressing the comments of both the reviewers. As you can see from the reports the referees find the manuscript interesting but find that the manuscript would very much benefit from restructuring, further discussion of related publications and an increased focus on the comparison between G1 and metaphase chromatin.

I should add that it is EMBO Journal policy to allow only a single round of revision, and acceptance of your manuscript will therefore depend on the completeness of your responses in this revised version.

Thank you for the opportunity to consider your work for publication. I look forward to your revision.

Yours sincerely,

Cornelius Schneider, PhD
Editor
The EMBO Journal
c.schneider@embojournal.org

We realize that it is difficult to revise to a specific deadline. In the interest of protecting the conceptual advance provided by the work, we recommend a revision within 3 months (3rd Nov 2024). Please discuss the revision progress ahead of this time with the editor if you require more time to complete the revisions. Use the link below to submit your revision:

Referee #1:

Although in situ analysis of eukaryotic cell chromatin structure has recently been the subject of intensive work worldwide, structural information on chromatin related to the cell cycle is still limited. In this article, the authors analyzed the chromatin structure of RPE-1 cells in G1 phase and metaphase using Cryo-FIB-SEM and Cryo-ET techniques. This analysis reveals a certain organization of chromatin in G1 phase and metaphase at nucleosomal level, as well as detailed differences in nucleosome structure within these 2 phases. The results also suggest the presence of nucleosomes with disordered gyre-proximal densities and ordered stacked dinucleosomes in the chromatin of RPE-1 cells. This study therefore contributes to the detailed elucidation of in situ chromatin structure in G1 phase and metaphase and constitutes a pioneering study in this field. On the other hand, the paper itself is highly specialized, even overly technical, and it is questionable whether it will appeal to a wide range of readers. The volume of the article is also very large (143 pages), including many results that are not directly related to the conclusions. The manuscript should therefore be simplified in view of its length. In particular, it describes at some length the cryosection, cryoprotectant and VPP conditions used to prepare cryogenic samples, covering 13 pages, but this is not the main subject of the article. This part should be moved to the methods section (or consider publishing it in a separate article, as it includes information useful to other researchers.). The authors should also consider removing the glycerol experiments from the manuscript.

In my opinion, given that the content includes many interesting points, I strongly recommend that the authors first significantly revise the overall composition of their manuscript and simplify unnecessary parts to present only the important results, focusing on the main topic, which is the in situ analysis of chromatin structure in G1 and metaphase. They can then resubmit their manuscript, making it easier to read and understand, and above all in the form of an article with greater impact.

Authors are encouraged to respond to the following comments to improve the quality of the manuscript.

Comment 1: The discussion is long. I would like to see a concise conclusion rather than a list of bullet points of results obtained.

Comment 2:

I don't think technical contents are necessary in the main text. Below are just a few examples:

-Pages 5-8: The passage " Optimization of cryoprotectants for cellular cryo-ET " should be considerably shortened or deleted. The description should be reduced to the strict minimum needed to understand the figures, so the part on experimental conditions could be omitted.

-Pages 8-10: The detailed description of the process of subtomogram averaging should be moved to the "method" section.

Comment 3:

The authors claim that nucleosome stacks are an artefact of DMSO. However, it seems to me that this assertion would cast doubt on all the stack structures obtained in this study, considering them all to be an artefact. The authors should be more cautious about this statement.

Comment 4:

The conclusion that nucleosomes could not be observed is consistent throughout the study, but from a biological point of view, I don't think it is a meaningful conclusion. It is possible that the nucleosomes simply weren't included in the tomogram, or that they couldn't be reconstructed in 3D. Their analysis therefore does not prove that nucleosomes do not exist.

Comment 5: Page 12

" When rendered at a lower contour level, an additional noisy density becomes visible proximal to the DNA gyres ".
The map corresponding to this description should be shown in the figure.

Comment 6: Figures S15, 19, 22, 31, 35

FSC curve threshold and map resolution should be added.

Comment 7: Figures S37-41

The analysis of the tomogram before denoising can be omitted as it is not a priority within this manuscript.

Referee #2:

Interphase chromatin organization plays critical roles in various genome functions, including transcription and DNA replication/repair. Mitotic chromosome assembly is an essential process for properly transmitting a replicated genome. Using state-of-the-art cryo-FIB and cryo-ET, Chen et al. determined the in-situ structures of nucleosomes in the G1 phase and metaphase of human RPE-1 cells. The authors observed that (1) G1 nucleosomes are concentrated in globular chromatin domains, while metaphase nucleosomes are uniformly distributed (no chromatin domains). (2) G1 and metaphase chromatin have ordered stacked dinucleosome classes, but no oligomers with more than two ordered nucleosomes. (3) Both G1 chromatin domains and metaphase chromosomes are densely packed with nucleosomes and are largely devoid of megacomplexes.

The study was carefully performed, and the obtained image data are very impressive and analyzed in detail. This paper is informative in the field of chromatin/chromosome research. On the other hand, the novelty of the work, especially in understanding chromatin/chromosome organization, is unclear. My specific comments are as follows:

- 1) I am still concerned about the structural preservation of the samples with cryoprotectant because I assume the cells with 9% DMSO are resistant for a short time but will eventually die.
- 2) I wonder whether the authors observed "clutch" structures proposed by Ricci et al. (PMID: 25768910), as their samples are much more "native" than those in Ricci et al., which underwent harsh manipulation and treatment.
- 3) Could the authors estimate the nucleosome densities (concentrations) in G1 chromatin domains and metaphase chromosomes and compare them with those in ChromEMT (PMID: 28751582)? Such data (and data on Comment 2) would be good additions to the paper regarding biological significance.
- 4) The authors might want to discuss data from Hou et al. (PMID: 37816746). It would be helpful for general readers.

Dear Dr. Schneider,

We thank the referees for their constructive comments. We have addressed all of their comments by doing major rearrangements of the text and figures and adding more discussion. The revised text focuses more on biology. Most of the technical text and figures have been moved to the Appendix.

Below, we have copied the original referee comments in black text along with point-by-point rebuttal in **bold blue text**. We have also made minor corrections throughout

Lu Gan

Referee #1:

Although in situ analysis of eukaryotic cell chromatin structure has recently been the subject of intensive work worldwide, structural information on chromatin related to the cell cycle is still limited. In this article, the authors analyzed the chromatin structure of RPE-1 cells in G1 phase and metaphase using Cryo-FIB-SEM and Cryo-ET techniques. This analysis reveals a certain organization of chromatin in G1 phase and metaphase at nucleosomal level, as well as detailed differences in nucleosome structure within these 2 phases. The results also suggest the presence of nucleosomes with disordered gyre-proximal densities and ordered stacked dinucleosomes in the chromatin of RPE-1 cells. This study therefore contributes to the detailed elucidation of in situ chromatin structure in G1 phase and metaphase and constitutes a pioneering study in this field.

On the other hand, the paper itself is highly specialized, even overly technical, and it is questionable whether it will appeal to a wide range of readers. The volume of the article is also very large (143 pages), including many results that are not directly related to the conclusions. The manuscript should therefore be simplified in view of its length. In particular, it describes at some length the cryosection, cryoprotectant and VPP conditions used to prepare cryogenic samples, covering 13 pages, but this is not the main subject of the article. This part should be moved to the methods section (or consider publishing it in a separate article, as it includes information useful to other researchers.). The authors should also consider removing the glycerol experiments from the manuscript.

In my opinion, given that the content includes many interesting points, I strongly recommend that the authors first significantly revise the overall composition of their manuscript and simplify unnecessary parts to present only the important results, focusing on the main topic, which is the in situ analysis of chromatin structure in G1 and metaphase. They can then resubmit their manuscript, making it easier to read and understand, and above all in the form of an article with greater impact.

Authors are encouraged to respond to the following comments to improve the quality of the manuscript.

We acknowledge that the manuscript is both technical and long. These features are necessary because the field (structural cell biology) is young and requires many more controls. While it is possible to substantially shorten the paper by removing figures/data or labeling them as “data not shown”, we believe it is in the best interests of the readership to present all the experiments and thought processes needed to make the conclusions here. As a compromise, we have shortened the manuscript body as much as possible by moving large blocks of text and figures to the Appendix. These changes should improve the readability while leaving the experimental record intact.

Comment 1: The discussion is long. I would like to see a concise conclusion rather than a list of bullet points of results obtained.

The first paragraph has been merged with the Conclusion paragraph. Note that some text was added to address Referee #2’s suggestions, though we tried to keep the new text

short. We have modified the subheadings to summarize the associated section rather than reading as bullet points.

Comment 2:

I don't think technical contents are necessary in the main text. Below are just a few examples:

-Pages 5-8: The passage " Optimization of cryoprotectants for cellular cryo-ET " should be considerably shortened or deleted. The description should be reduced to the strict minimum needed to understand the figures, so the part on experimental conditions could be omitted.

-Pages 8-10: The detailed description of the process of subtomogram averaging should be moved to the "method" section.

We have moved most of the technical details to the Appendix Discussion while keeping the essentials in the main text.

Comment 3:

The authors claim that nucleosome stacks are an artefact of DMSO. However, it seems to me that this assertion would cast doubt on all the stack structures obtained in this study, considering them all to be an artefact. The authors should be more cautious about this statement.

Thank you for the caveat on dinucleosome stacking. Stacked dinucleosomes are not artifacts of DMSO, which are visible even without averaging in the other two conditions. Instead, it is the ordered stacked dinucleosomes that may be artifacts. We have added additional reminders to readers that there is a difference between ordered stacked dinucleosomes and stacked dinucleosomes at key points throughout the text.

Comment 4:

The conclusion that nucleosomes could not be observed is consistent throughout the study, but from a biological point of view, I don't think it is a meaningful conclusion. It is possible that the nucleosomes simply weren't included in the tomogram, or that they couldn't be reconstructed in 3D. Their analysis therefore does not prove that nucleosomes do not exist.

We are not sure what is meant by "nucleosomes could not be observed" because nucleosomes are abundant in both the interphase and mitotic cells, regardless of the presence of cryoprotectants. Perhaps this comment refers to the various larger ordered structures such as arc-shaped stacks / slinkies, 100 – 200 nm periodic structure, ordered trinucleosome stacks, and slinky oligonucleosomes? If these larger ordered structures are what comment 4 refers to, then our conclusion that they were "not seen" is accurate because they were indeed not seen in any of our data. Such structures, if they exist, are rare, or else they would have been seen in one of our large dataset's tomograms. Given that the ordered structures are much-larger than ribosomes (which are easy to see in our data), and that the papers where they were proposed gave very detailed models, it is unlikely we would have missed them if they exist in any of our data. Also, please note we have not used the term "prove" in the manuscript.

Comment 5: Page 12

" When rendered at a lower contour level, an additional noisy density becomes visible proximal to the DNA gyres ".

The map corresponding to this description should be shown in the figure.

Thank you for catching this mistake. We regretfully omitted the figure callout here, which is now in the revised sentence (lines 287 – 289):

“When rendered at a lower contour level, an additional noisy density becomes visible proximal to the DNA gyres (Appendix Fig S13C, lower row)”

We have also added the figure callout to a similar description of the metaphase nucleosome class averages (lines 469 – 472):

“When rendered at a lower contour level, the metaphase stacked dinucleosome and some of the mononucleosome class averages show an extra density near the nucleosome face (Appendix Fig S27A and S27B, lower rows)”

Comment 6: Figures S15, 19, 22, 31, 35

FSC curve threshold and map resolution should be added.

We have now added the FSC = 0.5 and 0.143 threshold lines to all the FSC plots.

Comment 7: Figures S37-41

The analysis of the tomogram before denoising can be omitted as it is not a priority within this manuscript.

Because this study is one of the few to use denoising on Volta phase plate data, we feel it is valuable for the broader cellular cryo-ET community to see the before-and-after images. As with the other technically oriented material, we have moved the un-denoised images to the Appendix.

Referee #2:

Interphase chromatin organization plays critical roles in various genome functions, including transcription and DNA replication/repair. Mitotic chromosome assembly is an essential process for properly transmitting a replicated genome. Using state-of-the-art cryo-FIB and cryo-ET, Chen et al. determined the in-situ structures of nucleosomes in the G1 phase and metaphase of human RPE-1 cells. The authors observed that (1) G1 nucleosomes are concentrated in globular chromatin domains, while metaphase nucleosomes are uniformly distributed (no chromatin domains). (2) G1 and metaphase chromatin have ordered stacked dinucleosome classes, but no oligomers with more than two ordered nucleosomes. (3) Both G1 chromatin domains and metaphase chromosomes are densely packed with nucleosomes and are largely devoid of megacomplexes.

The study was carefully performed, and the obtained image data are very impressive and analyzed in detail. This paper is informative in the field of chromatin/chromosome research. On the other hand, the novelty of the work, especially in understanding chromatin/chromosome organization, is unclear. My specific comments are as follows:

1) I am still concerned about the structural preservation of the samples with cryoprotectant because I assume the cells with 9% DMSO are resistant for a short time but will eventually die.

We are unable to detect any structural differences between the nucleosomes in cryoprotectant (glycerol or DMSO) versus no cryoprotectant, but we do detect ordered stacked dinucleosomes in the presence of DMSO. More *in situ* cryo-ET studies of chromatin need to be done in the presence and absence of DMSO (and other cryoprotectants) to reveal the detrimental effects of these additives. We hope that the deposition of our raw cryo-ET data of a popular cell line (RPE-1) will facilitate such comparative analyses.

2) I wonder whether the authors observed "clutch" structures proposed by Ricci et al. (PMID: 25768910), as their samples are much more "native" than those in Ricci et al., which underwent harsh manipulation and treatment.

We have added a new paragraph to compare the clutch model with our study and other lab's cryo-ET studies in lines 736 – 748:

“An earlier super-resolution study of both human and mouse cells reported an organizational principle called the “clutch” (Ricci et al., 2015). In the clutch, 3 – 300 nucleosomes are locally compacted into heterogeneous nanodomains. The mostly irregular organization or chromatin domains in our G1 RPE-1 cells is largely compatible with the clutch model. Likewise, all previous and current cryo-EM and cryo-ET studies of mammalian chromatin in situ report irregular nucleosome packing and are therefore compatible with the clutch model (Cai et al., 2018a; Eltsov et al., 2018; Eltsov et al., 2008; Fatmaoui et al., 2022; Hou et al., 2023; McDowall et al., 1986; Wang et al., 2024). In these studies, the number of nucleosomes per domain was not characterized, and were mostly done on cell types that differ from Ricci's work, so it is unclear whether the number of

nucleosomes per clutch observed by super-resolution microscopy is compatible with the number of nucleosomes per chromatin domain seen in cryotomograms.”

3) Could the authors estimate the nucleosome densities (concentrations) in G1 chromatin domains and metaphase chromosomes and compare them with those in ChromEMT (PMID: 28751582)? Such data (and data on Comment 2) would be good additions to the paper regarding biological significance.

Our study is unable to identify enough nucleosomes to allow us to estimate nucleosome concentrations, which we now acknowledge in new lines 869 – 874:

“Improvements in sample preparation and image analysis may enable deeper structural cell-biological characterization. For example, the identification of more nucleosomes may allow us to estimate the in situ nucleosome concentration, which was recently reported to be ~750 μM in a volume electron-microscopy study of mitotic RPE-1 cells (Cisneros-Soberanis et al, 2024)”

The ChromEMT paper defined “concentration” based on a parameter they term “chromatin volume concentration” (CVC), based on the number of voxels within a 120 x 120 x 120 nm cube whose densities pass a threshold that is deemed to be chromatin. While the CVC is reported to be higher in mitosis than interphase chromatin, it is unclear what this number means in terms of nucleosome concentration in μM . Therefore, we don’t think it is possible to compare chromatin concentrations yet.

4) The authors might want to discuss data from Hou et al. (PMID: 37816746). It would be helpful for general readers.

The discussion of Hou et al. (PMID: 37816746) has been spread through different sections along with related papers. We have added additional discussions of this paper that were missing before. Here are all the instances.

Lines 219 – 222:

“Next, we performed subtomogram classification and averaging analysis of nucleosomes in the G1 chromatin domains, which revealed canonical nucleosome class averages (Appendix Fig S11A) like we saw in HeLa (Cai et al., 2018a) and what others have seen in T-lymphoblast cells (Hou et al., 2023), and mouse cells (Wang et al., 2024).”

Lines 559 – 561:

“Nucleosome class averages that have short and long ordered linker DNA were also seen in HeLa, T-lymphoblast, and budding yeast cells (Cai et al., 2018a; Hou et al., 2023; Tan et al., 2023), but not in MEFS (Wang et al., 2024)”

Lines 677 – 685:

“Consistent with these super-resolution studies, chromatin domains have irregular shapes throughout the G1 RPE-1 nucleus, in the perinuclear region in a cryotomogram of

an interphase HeLa cell cryolamella (Cai et al., 2018a), electron spectroscopic images of mouse fibroblast cells (Strickfaden et al., 2020), and near the nuclear periphery of MEFs (Wang et al., 2024). In contrast, domain-like features are not seen in the popular model yeasts *S. cerevisiae* and *S. pombe*, which have uniform chromatin distributions throughout the interphase nucleoplasm (Cai et al., 2018b; Chen et al., 2016; Tan et al., 2023). Chromatin domains were not reported in a cryo-ET study T-lymphoblasts (Hou et al., 2023).”

Lines 729 – 734:

“The RPE-1 data reported here therefore adds to the growing list of in situ cryo-EM and cryo-ET studies of in picoalgae, budding yeast, fission yeast, HeLa, T-lymphoblast, MEFs, and fly neuron cells (Cai et al., 2018a; Cai et al., 2018b; Chen et al., 2016; Eltsov et al., 2018; Eltsov et al., 2008; Fatmaoui et al., 2022; Gan et al., 2013; Hou et al., 2023; Wang et al., 2024), which all report that frozen-hydrated eukaryotic chromatin packing is irregular in situ.”

Lines 740 – 748:

“Likewise, all previous and current cryo-EM and cryo-ET studies of mammalian chromatin in situ report irregular nucleosome packing and are therefore compatible with the clutch model (Cai et al., 2018a; Eltsov et al., 2018; Eltsov et al., 2008; Fatmaoui et al., 2022; Hou et al., 2023; McDowall et al., 1986; Wang et al., 2024). In these studies, the number of nucleosomes per domain was not characterized, and were mostly done on cell types that differ from Ricci’s work, so it is unclear whether the number of nucleosomes per clutch observed by super-resolution microscopy is compatible with the number of nucleosomes per chromatin domain seen in cryotomograms.”

Lines 874 – 875:

“and also reveal the fold of nucleosomes, which was reported to follow an irregular zig-zag path in T-lymphoblasts (Hou et al., 2023).”

Dear Dr. Gan,

Thank you for submitting a revised version of your manuscript. Your study has now been seen by all original referees, who find that their previous concerns have been addressed and now recommend publication of the manuscript. There remain only a few mainly editorial points that have to be addressed before I can extend formal acceptance of the manuscript:

- Please remove "Abbreviations used" and "New terms" from the title page
- Please reduce the number of keywords on the abstract page to five (ideally choosing broad general terms).
- Please rename the "data sharing" section to "Data Availability"
- As we are switching from a free-text author contribution statement towards a more formal statement based on Contributor Role Taxonomy (CRediT) terms, please remove the present Author Contribution section and instead specify each author's contribution(s) directly in the Author Information page of our submission system during upload of the final manuscript. See <https://casrai.org/credit/> for more information.
- There is a reference to "data not shown" on page 11, "Canonical nucleosome classes were not obtained from the nucleoplasm" section. According to our policy, which does not permit references to "data not shown", please include this information in the Appendix. Please see also <https://www.embopress.org/page/journal/14602075/authorguide#unpublisheddata>.
- Please correct the author checklist: only section names should be included in the third column (the pink one), not explanations, and if the response is "Not Applicable" the pink box should be blank
- Please convert the Appendix into PDF format; page numbers should be added to included items in ToC
- Please provide source data and fill out the blank SD checklist which you can find on your manuscript page
- Please provide suggestions for a short 'blurb' text prefacing and summing up the conceptual aspect of the study in two sentences (max. 250 characters), followed by 3-5 one-sentence 'bullet points' with brief factual statements of key results of the paper; they will form the basis of an editor-written 'Synopsis' accompanying the online version of the article. Please also provide an altered synopsis image, making sure that the aspect ratio conforms to our website's format - it should be exactly 550 pixels wide and between 300-600 pixels high.
- Figure Legends (main + EV):
 1. Please note that the scale bar is missing for figure 2B
 2. Please note that scale bar and its definition are missing for figures 2, 4B, D, F; 6B
 3. Please note that the black arrows are not defined in the legend of figure 3D, 5D.
- Please remove the legends from ms file and zip together with each movie file.
- Please adjust the order of the manuscript sections: Title page with complete author information, Abstract, Keywords, Introduction, Results, Discussion, Methods, Data Availability Section, Acknowledgements, Disclosure and Competing Interests Statement, References, Main figure legends, Tables, Expanded Figure Legends.

With best regards,

Cornelius Schneider

Cornelius Schneider, PhD
Editor
The EMBO Journal
c.schneider@embojournal.org

We realize that it is difficult to revise to a specific deadline. In the interest of protecting the conceptual advance provided by the work, we recommend a revision within 3 months (13th Apr 2025). Please discuss the revision progress ahead of this time with the editor if you require more time to complete the revisions. Use the link below to submit your revision:

Referee #1:

All comments have been addressed, and the main concern regarding the excessive length of the manuscript has been resolved by moving a large number of figures to the Appendix. The overall text length has also been reduced to a satisfactory level. These adjustments have significantly streamlined the manuscript.

I have no further comments or suggestions.

Referee #2:

The authors adequately addressed my comments. The revised manuscript has been much improved and is ready for publication in EMBO J.

Dear Dr. Gan,

Thank you for submitting a revised version of your manuscript. Your study has now been seen by all original referees, who find that their previous concerns have been addressed and now recommend publication of the manuscript. There remain only a few mainly editorial points that have to be addressed before I can extend formal acceptance of the manuscript:

Dear Editor,

Please see our responses below, all of which are highlighted and bolded. The changes made to the manuscript files are also highlighted.

Note that two of the authors have moved to the University of Virginia, so we have added our new affiliation to the title page as "Present Address"; this may be deleted if not allowed by EMBO Journal. We have added an acknowledgement for the University of Virginia funding for these two authors.

- Please remove "Abbreviations used" and "New terms" from the title page

Both sections are removed in _revise3.

- Please reduce the number of keywords on the abstract page to five (ideally choosing broad general terms).

Removed "compaction" and "human cells" keywords.

- Please rename the "data sharing" section to "Data Availability"

Renamed "Data sharing" section to "Data availability".

- As we are switching from a free-text author contribution statement towards a more formal statement based on Contributor Role Taxonomy (CRediT) terms, please remove the present Author Contribution section and instead specify each author's contribution(s) directly in the Author Information page of our submission system during upload of the final manuscript. See <https://casrai.org/credit/> for more information.

Removed "Contributions" section from text.

- There is a reference to "data not shown" on page 11, "Canonical nucleosome classes were not obtained from the nucleoplasm" section. According to our policy, which does not permit references to "data not shown", please include this information in the Appendix. Please see also

<https://www.embopress.org/page/journal/14602075/authorguide#unpublisheddata>.

The negative result here is an empty text file (no nucleosome positions recorded) so there is nothing to show. We have removed the text "not shown" to avoid confusion.

- Please correct the author checklist: only section names should be included in the third column (the pink one), not explanations, and if the response is "Not Applicable" the pink box should be blank

We have corrected the checklist as instructed. For some of rows, we entered "Appendix Table 1". If these should be less specific, i.e., "Appendix", let us know.

- Please convert the Appendix into PDF format; page numbers should be added to included items in ToC

Converted Appendix docx to pdf format. Added page numbers to items in ToC.

- Please provide source data and fill out the blank SD checklist which you can find on your manuscript page

As discussed in our email, there are no changes to the SD checklist. The source data is either accessible (Bioimage Archive, EMDDB) or accessible in the next release cycle (EMPIAR).

- Please provide suggestions for a short 'blurb' text prefacing and summing up the conceptual aspect of the study in two sentences (max. 250 characters), followed by 3-5 one-sentence 'bullet points' with brief factual statements of key results of the paper; they will form the basis of an editor-written 'Synopsis' accompanying the online version of the article. Please also provide an altered synopsis image, making sure that the aspect ratio conforms to our website's format - it should be exactly 550 pixels wide and between 300-600 pixels high.

G1 chromatin domains

Blurb:

Advances in electron tomography reveal nucleosome linker-DNA variability and the irregular packing of groups of more than two nucleosomes in both human RPE-1 cells at G1 and metaphase.

Bullet points:

- In situ electron cryotomography of chromatin in RPE-1 cells arrested at G1 and metaphase.
- Nucleosomes are concentrated in chromatin domains at G1 and chromatids at metaphase.
- Subtomogram classification and averaging analysis reveal multiple species of nucleosomes.
- The most-ordered motif detected as a class average is the stacked dinucleosome, suggesting that nucleosome packing is heterogeneous overall.

- Figure Legends (main + EV):

1. Please note that the scale bar is missing for figure 2B

Scalebar and its definition added for Figure 2B.

2. Please note that scale bar and its definition are missing for figures 2, 4B, D, F; 6B

Scalebars and their definition added for Figure 2D, E, F; Figures 4B, D, E, F; and 6B.

3. Please note that the black arrows are not defined in the legend of figure 3D, 5D.

The definition of the black arrowheads and dashed outlines in the insets of Figure 3D and 5D have been added to the figure legend (highlighted yellow).

- Please remove the legends from ms file and zip together with each movie file.

Movie legends have been removed from the ms file. Movie files are now zipped with their respective legend docx and re-uploaded.

- Please adjust the order of the manuscript sections: Title page with complete author information, Abstract, Keywords, Introduction, Results, Discussion, Methods, Data Availability Section, Acknowledgements, Disclosure and Competing Interests Statement, References, Main figure legends, Tables, Expanded Figure Legends.

The sections have been reordered as listed above.

With best regards,

Cornelius Schneider

Cornelius Schneider, PhD
Editor
The EMBO Journal
c.schneider@embojournal.org

Further information is available in our Guide For Authors:

We realize that it is difficult to revise to a specific deadline. In the interest of protecting the conceptual advance provided by the work, we recommend a revision within 3 months (13th Apr 2025). Please discuss the revision progress ahead of this time with the editor if you require more time to complete the revisions. Use the link below to submit your revision:

Dear Dr. Gan,

I am pleased to inform you that your manuscript has been accepted for publication in the EMBO Journal.

Yours sincerely,

Cornelius Schneider, PhD
Editor
The EMBO Journal
c.schneider@embojournal.org
